# On the limits of fitting complex models of population history to *f*-statistics

Robert Maier[1]*[†], Pavel Flegontov[1,2]*[†], Olga Flegontova[2], Ulaş Işıldak[2], Piya Changmai[2], David Reich[1,3,4,5]*

[1]Department of Human Evolutionary Biology, Harvard University, Cambridge, United States; [2]Department of Biology and Ecology, Faculty of Science, University of Ostrava, Ostrava, Czech Republic; [3]Broad Institute of Harvard and MIT, Cambridge, United States; [4]Howard Hughes Medical Institute, Harvard Medical School, Boston, United States; [5]Department of Genetics, Harvard Medical School, Boston, United States

**\*For correspondence:**
robertmaier@gmx.net (RM);
Pavel_Flegontov@hms.harvard.
edu (PF);
reich@genetics.med.harvard.
edu (DR)

[†]These authors contributed
equally to this work

**Competing interest:** The authors
declare that no competing
interests exist.

**Reviewing Editor:** Magnus
Nordborg, Gregor Mendel
Institute, Austria

**Abstract** Our understanding of population history in deep time has been assisted by fitting admixture graphs (AGs) to data: models that specify the ordering of population splits and mixtures, which along with the amount of genetic drift and the proportions of mixture, is the only information needed to predict the patterns of allele frequency correlation among populations. The space of possible AGs relating populations is vast, and thus most published studies have identified fitting AGs through a manual process driven by prior hypotheses, leaving the majority of alternative models unexplored. Here, we develop a method for systematically searching the space of all AGs that can incorporate non-genetic information in the form of topology constraints. We implement this *findGraphs* tool within a software package, *ADMIXTOOLS 2*, which is a reimplementation of the *ADMIXTOOLS* software with new features and large performance gains. We apply this methodology to identify alternative models to AGs that played key roles in eight publications and find that in nearly all cases many alternative models fit nominally or significantly better than the published one. Our results suggest that strong claims about population history from AGs should only be made when all well-fitting and temporally plausible models share common topological features. Our re-evaluation of published data also provides insight into the population histories of humans, dogs, and horses, identifying features that are stable across the models we explored, as well as scenarios of populations relationships that differ in important ways from models that have been highlighted in the literature.

## Editor's evaluation

This is a rigorous and critical analysis of the performance of a popular suite of methods for inferring population history, accompanied by improvements. Should be of broad interest to anyone interested in human history.

## Introduction

Admixture graph models provide a powerful intellectual framework for describing the relationships among populations that allows not only branching of populations from a common ancestor but also mixture events. An admixture graph (abbreviated below as AG), as fit in the widely used software packages *ADMIXTOOLS* (*Patterson et al., 2012*) and *TreeMix* (*Pickrell and Pritchard, 2012*; *Molloy et al., 2021*), is a directed acyclic bifurcating graph with two types of edges: those representing genetic drift, and those representing gene flow. Each admixture event is represented as a confluence

of two gene flow edges. Nodes of such a graph represent unsampled intermediate populations, and terminal nodes (leaves) represent sampled present-day or ancient groups (see a mathematical definition in *Soraggi and Wiuf, 2019*). An attractive feature of AGs is that they can summarize important features of population history without requiring specification of all parameters such as population sizes, split times, mixture times, and distinguishing between sudden splits or drawn-out separations. All these parameters describe important features of demographic history and are fit by many methods for fitting demographic models (*Gutenkunst et al., 2009*; *Gronau et al., 2011*; *Schiffels et al., 2016*; *Flegontov et al., 2019*; *Kamm et al., 2020*, *Rogers, 2019*; *Hubisz et al., 2020*). However, the fact that it is possible to factor this difficult problem by first inferring important aspects of the topology (AGs fitted to allele frequency correlation statistics), and then fitting additional demographic parameters to data such as site frequency spectra, simplifies demographic inference (*Patterson et al., 2012*; *Pickrell and Pritchard, 2012*; *Lipson et al., 2013*; *Leppälä et al., 2017*; *Lipson et al., 2020b*; *Molloy et al., 2021*; *Yan et al., 2021*). AGs thus serve both as conceptual frameworks that allow us to think about the relationships of populations deep in time, and as mathematical models we can fit to genetic data.

AGs are fitted to *f*-statistics (*Reich et al., 2009*; *Patterson et al., 2012*; *Peter, 2016*; *Soraggi and Wiuf, 2019*). For convenience, below we use a concise definition of *f*-statistics by *Lipson, 2020a*: 'The most general definition is that of the $f_4$-statistic $f_4(A, B; C, D)$, which measures the average correlation in allele frequency differences between (1) populations $A$ and $B$ and (2) populations $C$ and $D$ that is, $(p_A - p_B) * (p_C - p_D)$, for allele frequencies $p$, typically averaged over many biallelic single-nucleotide polymorphisms. This $f_4$-statistic is the same as the *D*-statistic up to a normalization factor.' The other *f*-statistics ($f_2$ and $f_3$) can be defined as special cases of $f_4$-statistics: $f_2(A, B) = f_4(A, B; A, B)$ and $f_3(A; B, C) = f_4(A, B; A, C)$. $f_4$-Statistics can be written as linear combinations of $f_3$- or $f_2$-statistics, and $f_3$-statistics can be written as linear combinations of $f_4$- and $f_2$-statistics. $f_2$-, $f_3$-, and $f_4$-statistics have straightforward interpretations in terms of drift edges along the tree, see Figure 2 in *Patterson et al., 2012* and *Appendix 1—figure 1b*. A challenge for fitting AG models is that they are often not uniquely constrained by the data, with many providing equally good fits to the $f_2$-, $f_3$-, and $f_4$-statistics used to constrain them within the limits of statistical resolution. Previously published methods for finding fitting AGs (mainly *qpGraph*, *Patterson et al., 2012* and *TreeMix*, *Pickrell and Pritchard, 2012*; *Molloy et al., 2021*) were not well equipped to handle the large range of equally well-fitting models for three reasons: (1) They did not reliably provide information on whether there is a uniquely fitting parsimonious model or alternatively whether there are many models that fit equally well to the limits of statistical resolution, (2) they did not provide formal goodness-of-fit tests, and related to this, (3) they did not provide tests for whether the difference between the fits of any two models is statistically significant. As a consequence, and as we demonstrate in what follows, many published AG models have been interpreted as providing more confidence than is merited about the extent to which genetic data allows us to disentangle ancestral relationships.

To appreciate these problems, we first need to consider two main approaches that were utilized to study demographic history with AGs.

The first approach is to identify AGs automatically, either without human intervention or with guidance. It is possible in theory to exhaustively test all possible graphs for a given set of populations and pre-specified number of admixture events, as implemented, for example, in the *admixturegraph* R package (*Leppälä et al., 2017*). An exhaustive approach can provide a complete view of the range of models that are consistent with the data for a specified level of parsimony (total number of admixture events allowed in the graph), which is not biased by the algorithm used to explore the space of possible AGs. However, this approach is limited to small graphs (typically up to six groups, two admixture events) due to the rapid increase in the number of possible AGs as the number of populations and admixture events grows. As we show in our discussion of case studies, the simple models explored with an exhaustive approach can lead to misleading conclusions about population history because not including additional populations can blind users to additional mixture events that occurred (and whose existence is revealed by examining data from additional populations). Furthermore, models with additional admixture events that are qualitatively different to the best-fitting parsimonious graph and that capture the true history, will sometimes be completely missed when constraining the number of gene flows. Alternatively, the programs *TreeMix* (*Pickrell and Pritchard, 2012*; *Molloy et al., 2021*), *MixMapper* (*Lipson et al., 2013*), *miqoGraph* (*Yan et al., 2021*), and *AdmixtureBayes* (*Nielsen*

*et al., 2023* preprint) all address the problem of how to rapidly explore the vast space of AGs relating a set of populations by applying algorithmic ideas or heuristics; all of these methods speed up model search by orders of magnitude.

The second approach to fitting AGs is to manually build them up by grafting additional populations onto simpler smaller graphs that fit the data. This approach involves stepwise addition of populations in an order that is chosen based on the best judgment of the user, and for each newly added population involves adding admixture events or tweaks in the graph until a fit is obtained; the user then moves on to adding the next population (see *Reich et al., 2009*; *Reich et al., 2011*; *Reich et al., 2012*; *Lazaridis et al., 2014*; *Seguin-Orlando et al., 2014*; *Fu et al., 2016*; *Skoglund et al., 2016*; *Yang et al., 2017*; *McColl et al., 2018*; *Moreno-Mayar et al., 2018*; *Tambets et al., 2018*; *van de Loosdrecht et al., 2018*; *Flegontov et al., 2019*; *Sikora et al., 2019*; *Wang et al., 2019*; *Lipson et al., 2020b*; *Shinde et al., 2019*; *Yang et al., 2020*; *Hajdinjak et al., 2021*; *Wang et al., 2021*; *Bergström et al., 2022* for examples). The program *qpGraph* in the *ADMIXTOOLS* package (*Patterson et al., 2012*) has been the most common computational method used for testing fits of individual AGs. Most AGs in the literature have been constructed manually in this way, often acknowledging the existence of alternative models by presenting plausible models side-by-side, and this approach has been the basis for many claims about population history (*Lazaridis et al., 2014*; *Yang et al., 2017*; *Posth et al., 2018*; *Sikora et al., 2019*; *Shinde et al., 2019*; *Bergström et al., 2020*; *Lipson et al., 2020b*; *Hajdinjak et al., 2021*; *Wang et al., 2021*; *Bergström et al., 2022*). A strength of this approach is that it takes advantage of human judgment and outside knowledge about what graphs best fit the history of the human or animal populations being analyzed. This external information is powerful as it can incorporate nongenetic evidence such as geographic plausibility and temporal ordering of populations or linguistic similarity, or other genetic data such as estimates of population split times, or shared Y chromosomes, or rejection of proposed scenarios based on joint analysis of much larger numbers of populations than can reasonably be analyzed within a single AG. Thus, while manual approaches explore many orders of magnitude fewer topologies than automatic approaches often do, they still may provide inferences about population history that are more useful than those provided by automatic approaches. These methods' strength is also their weakness: by relying on intuition, following a manual approach has the potential to validate the biases users have as to what types of histories are most plausible (these may be the only types of histories that will be carefully explored). This can blind users to surprises: to profoundly different topologies that may correspond more closely to the true history, and we discuss examples of this in the Results section.

In this study, we introduce a new method, *findGraphs*, that belongs to the first class of algorithms (those for automated AG topology inference). Algorithmic innovations and speedups in *findGraphs* enable us to explore a much larger proportion of plausible AG space than many other methods reported to date. The *findGraphs* method combines the advantages of automated and manual topology exploration by allowing users to encode various sources of information as constraints on the space of AGs, which is then explored automatically. However, the main innovations in *findGraphs* are not computational, but instead conceptual. Instead of finding one or a few AGs fitting the data well, we use *findGraphs* for exploring AG spaces and assessing if any reliable information on population history can be extracted from a given AG space (defined by a population set and parsimony constraints) in the first place.

## Results

Regardless of the approach used to search through the space of possibly fitting AGs, a challenge in the effort to find a uniquely well-fitting AG (or group of topologically similar AGs) is that it has been difficult to quantify the absolute goodness of fit of a model to date. We have not been entirely successful with this and are not aware of other work that has been successful. It is also difficult to assess the relative fits of multiple models, especially if they differ in complexity. Performance gains relative to the original implementation of *qpGraph* allow us to address this problem by obtaining bootstrap confidence intervals and p-values for estimated parameters of single models, as well as for the difference in fit quality of two models (see Appendix 1, Sections 1.B.3 and 2.E). In combination with the approach to automating the search of well-fitting AGs, this leads to a situation where we are able to find and test a large number of models, many of which fit equally well despite often having very different topological features. Published approaches to comparing the fits of AG models based

on Akaike information criterion (AIC) or Bayesian information criterion (BIC), see *Flegontov et al., 2019*; *Shinde et al., 2019* have the problem that it is often not clear what the effective number of degrees of freedom is in the two models being compared since in the case of AGs it depends not only on the number of graph edges, but also on graph topology.

The methods for automated graph topology inference and model comparison relying on bootstrap resampling are implemented in *ADMIXTOOLS 2*, a comprehensive platform for learning about population history from *f*-statistics. It is built to provide a stand-alone workspace for research in this area and is implemented as an R package. For all computations, *ADMIXTOOLS 2* exhibits large speedups relative to previously published platforms for *f*-statistic analysis (e.g., *popstats* and *ADMIXTOOLS* version 6.0 which we call 'Classic *ADMIXTOOLS*' in what follows to distinguish it from updated *ADMIXTOOLS* version 7.0.2 which implements some of the speedup ideas also implemented in *ADMIXTOOLS 2*). This is achieved by deploying a series of algorithmic improvements, most notably storage of precomputed *f*-statistics in random access memory, which avoids having to rely on reading in extremely large genotype matrices to perform most computations. In addition to the new algorithmic ideas allowing efficient searching through the space of AGs and comparing the fits of two AGs, *ADMIXTOOLS 2* also provides a solution to the question of which parameters of an AG are identifiable in the limit of infinite data. Methodological details are presented in Appendix 1, and below we focus on documenting problems of AG inference on simulated data and revisiting AGs from the literature to understand the extent to which methodological challenges with AG fitting biased previous studies.

## Topological diversity of well-fitting models and effects of parsimony constraints on simulated data

First, we explored the performance of the *findGraphs* method for automated topology inference on simulated AGs of random topology, focusing on the following questions: (1) among *findGraphs* results, how common are AGs fitting nominally or significantly better than the true one but different topologically; (2) what is the degree of topological diversity among these models fitting the data better than the true one? For this purpose, we simulated AGs of four complexity classes using *msprime v.1.1.1*: eight or nine non-outgroup populations, and four or five admixture events. Only simulations where pairwise $F_{ST}$ for groups were in the range characteristic for anatomically modern and archaic humans were selected for further analysis, resulting in 20 random topologies per complexity class, each including a distant outgroup that facilitates automated exploration of the topology space.

We ran *findGraphs* on each simulated dataset starting from random graphs and pre-specifying the true number of admixture events ($n$), or $n - 1$, or $n + 1$ events. For each of these graph complexity levels, we performed 100 independent *findGraphs* runs and recorded 5 AGs from each run having the best log-likelihood (LL) scores. Topologically redundant AGs were discarded, and for the remaining AGs we calculated worst $f_4$-statistic residuals (WR) and tested if the newly found models fit significantly better than the true model, using the bootstrap model comparison method developed in this study (see Appendix 1, Sections 1.B.3 and 2.E). In *Figure 1a–c*, we show the following statistics for each simulated AG, summarized across simulated complexity classes and parsimony levels allowed at the stage of topology exploration: fraction of topologies found with *findGraphs* that fit better than the true AG (according to LL score), or that fit significantly better than the true AG, or those with plausible absolute fits (WR < 3 SE). It is clear that for the great majority of simulated datasets, even a shallow exploration of the topology space with *findGraph* (100 independent runs) uncovers AGs that fit nominally better than the true topology (*Figure 1a and d*) and are topologically diverse (see *Figure 1e* for examples). When allowing for $n$ admixture events, at least one AG fitting significantly better than the true one was found for 60% of simulated datasets. When $n + 1$ admixture events were allowed, this grew to 100% (all 80 datasets). It should be noted that some admixture events are indistinguishable with *f*-statistics; for instance, successive gene flows between two lineages, with no other edges branching off between the gene flows. If such gene flows were included in the random topologies we simulated, AGs with $n$ events were overly complex for representing the true history. Thus, if we are dealing with random histories, choosing an optimal complexity class for topology search is not straightforward.

These results on simulated data raise concerns about the extent to which fitting AG topologies provide reliable information about population history. Even for histories including eight or nine groups, an outgroup, and four or five pulse-like admixture events, perfect diploid data, and groups as

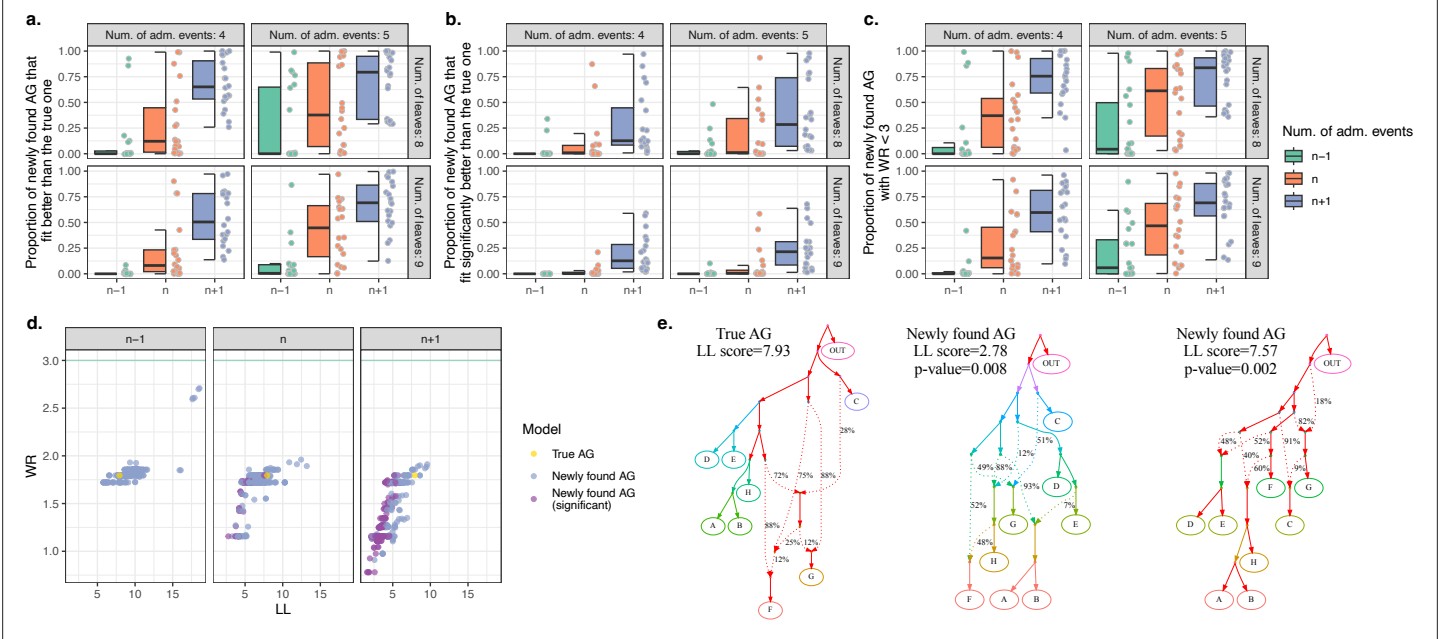

**Figure 1.** Computer simulations show that when the true admixture graph (AG) topology is complex, *findGraphs* frequently finds AGs fitting the data better than the true AG. (**a**) Fractions of distinct AGs found with *findGraphs* that fit the data nominally better than the true AG (according to log-likelihood [LL] scores). The simulated datasets are grouped by complexity class (eight or nine leaves, four or five admixture events) and by the number of admixture events allowed at the topology search stage (n − 1 on the left, n in the middle, and n + 1 on the right, where n is the true number of simulated admixture events). Each dot represents a simulated random history, and 20 such histories were simulated for each complexity class. (**b**) Fractions of distinct AGs found with *findGraphs* that fit the data significantly better than the true AG (two-tailed empirical p-value of the bootstrap model comparison method <0.05). (**c**) Fractions of distinct AGs found with *findGraphs* that fit the data well in absolute terms (WR < 3 SE). (**d**) Distinct AGs found for a particular simulated history (eight groups and four admixture events) in the LL and WR coordinates. Only best-fitting graphs with WR < 3 SE are shown. The fit of the true topology is shown in yellow, and topologies that fit the data significantly better than the true one are in purple. The true topology was not recovered by our *findGraphs* searches. (**e**) The true model from panel (**d**) and two alternative models found with *findGraphs*, both fitting significantly better than the true one (based on the bootstrap p-value) and very different topologically. This is presented as an example of very high topological diversity seen among well-fitting models. Model parameters (graph edges) that were inferred to be unidentifiable (see Appendix 1, Section 2.F) are plotted in red.

differentiated as Neanderthals and anatomically modern humans—a complexity class that is simpler than many models fitted to real genetic data in published papers—models fitting the data as well as or better than the true one are common, and their topological diversity is in most cases so high that it precludes consensus inference of topology by analysis of multiple topologies. As we demonstrate on another set of simulated data in Appendix 1 (Section 1.B.1), the probability of finding a 'wrong' model that fits better than the true one grows with increasing graph complexity, and that effect is reproduced with both *findGraphs* and *TreeMix* (*Appendix 1—figure 2*). We expect this problem to be even more acute when researchers are dealing with realistic complex histories. However, geneticists often rely on external constraints on AG topologies (such as temporal plausibility of a topology, results from *qpAdm* modelling, $f_4$-statistics, PCA, *ADMIXTURE*, geographical, archaeological, and linguistic considerations) that were not used for filtering the results of our topology searches on simulated data. Thus, it is possible in principle that published AG models are more robust than our results on simulated data, and we explore this issue in depth in the next section and in Appendix 2.

## Revisiting published AGs

We studied AGs from eight publications (*Lazaridis et al., 2014*; *Shinde et al., 2019*; *Sikora et al., 2019*; *Bergström et al., 2020*; *Lipson et al., 2020b*; *Hajdinjak et al., 2021Librado et al., 2021*; *Wang et al., 2021*) with the goal of comparing published models to models identified by our algorithm for automatically inferring optimal (best-fitting) AGs (*Table 1*). In all but one study, *qpGraph* or its automated reimplementation (*admixturegraph*, *Leppälä et al., 2017*) was used for fitting topologies to genetic data, while *Librado et al., 2021* relied on the automated *OrientAGraph* method (*Molloy*

**Table 1.** Published graphs in the context of automatically found graphs.
We compared graphs from eight publications to alternative graphs inferred on the same or very similar data (see *Supplementary file 1* for details).

| Publication | Figure in the original publication | Groups (populations) | Admixture events | SNPs used | Publ. model: worst residual, SE | Distinct alternative topologies found | Significantly better fitting topologies, % | Non-significantly better fitting topologies, % | Non-significantly worse fitting topologies, % | Significantly worse fitting topologies, % |
|---|---|---|---|---|---|---|---|---|---|---|
| *Bergström et al., 2020* | 1e | 7 | 3 | 312,282 | 2.1 | 221 | 0.5 | 2.3 | 16.7 | 80.5 |
| *Lazaridis et al., 2014* | 3 | 7 | 4 | 642,247 | 2.2 | 306 | 1.0 | 12.1 | 80.7 | 6.2 |
| *Shinde et al., 2019* | 3 | 8* | 3 | 2,49,009 | 2.6 | 143 | 0.0 | 2.8 | 3.5 | 93.7 |
| | 3b | | 3 | | 23.9 | 324 | 6.8 | 15.7 | 24.1 | 53.4 |
| *Librado et al., 2021* | Ext 5d | | 4 | | 14.1 | 535 | 0.0 | 0.0 | 4.5 | 95.5 |
| | Ext 5e | 10* | 5 | 1,767,419 | 6.9 | 784 | 0.0 | 0.3 | 28.4 | 71.3 |
| *Hajdinjak et al., 2021* | 2d | 12 | 8 | 263,698 | 4.8 | 1988 | 15.7 | 55.7 | 6.6 | 22.0 |
| *Lipson et al., 2020b* | Ext 4 | 12 | 11† | 211,738 | 2.3 | 2000 | 0.0 | 11.9 | 77.1 | 10.4 |
| *Wang et al., 2021* | Ext 6 | 12 | 8† | 203,753 | 3.8 | 1778 | 12.6 | 84.3 | 3.1 | 0.0 |
| *Sikora et al., 2019* | 3f (left) | 13 | 6† | 344,903 | 3.8 | 894 | 0.3 | 17.1 | 34.6 | 48.0 |
| | 3f (right) | 14 | 6† | 613,509 | 4.2 | 2785 | 0.1 | 0.9 | 9.8 | 89.2 |

**Publication:** Last name of the first author and year of the relevant publication.

**Figure in the original publication:** Figure number in the original paper where the AG is presented.

**Groups (populations):** The number of populations in each graph.

**Admixture events:** The number of admixture events in each graph.

**SNPs used:** The number of single-nucleotide polymorphisms (SNPs; with no missing data at the group level) used for fitting the AGs. For all case studies, we tested the original data (SNPs, population composition, and the published graph topology) and obtained model fits very similar to the published ones. However, for the purpose of efficient topology search, we in some cases adjusted settings for $f_3$-statistic calculation, population composition, or graph complexity as noted in the footnotes, in *Supplementary file 1*, and discussed in the text.

**Publ. model:** Worst residual, SE: The worst $f$-statistic residual of the published graph fitted to the SNP set shown in the 'SNPs used' column, measured in standard errors (SE).

**Distinct alternative topologies found:** The number of distinct newly found topologies differing from the published one.

**Significantly better fitting topologies, %:** The percentage of distinct alternative topologies that fit significantly better than the published graph according to the bootstrap model comparison test (two-tailed empirical p-value <0.05). If the number of distinct topologies was very large, a representative sample of models (1/20 to 1/3 of models evenly distributed along the log-likelihood spectrum) was compared to the published one instead, and the percentages in this and following columns were calculated on this sample.

**Non-significantly better fitting topologies, %:** The percentage of distinct topologies that fit non-significantly (nominally) better than the published graph according to the bootstrap model comparison test (two-tailed empirical p-value ≥0.05).

**Non-significantly worse fitting topologies, %:** The percentage of distinct topologies that fit non-significantly (nominally) worse than the published graph according to the bootstrap model comparison test (two-tailed empirical p-value ≥0.05).

**Significantly worse fitting topologies, %:** The percentage of distinct topologies that fit significantly worse than the published graph according to the bootstrap model comparison test (two-tailed empirical p-value <0.05).

*The population composition was modified, see *Supplementary file 1* and the text.

†Certain gene flows were removed from the published model for simplicity, see *Supplementary file 1* and the text.

*et al., 2021*). The main question we were interested in is whether we can find alternative models which (1) fit as well as, or better than the published graph, (2) differ in important ways from the published graph, and (3) cannot immediately be rejected based on other evidence such as temporal plausibility. The studies were selected according to the criterion that an AG model inferred in the study is used as primary evidence for at least one statement about population history in the main text of the study. In other words, the AG method was used in the original studies to support new conclusions about population history, and not simply to show that there is a model that exists that does not contradict results of other genetic analyses, an approach that is a valid use of AGs and has been taken in some studies (e.g., *Seguin-Orlando et al., 2014*; *Narasimhan et al., 2019*; *Wang et al., 2019*). There are many published studies that could have been included in our re-evaluation exercise as they meet our key criterion (e.g., *Yang et al., 2017*; *McColl et al., 2018*; *Posth et al., 2018*; *Flegontov et al., 2019*; *Carlhoff et al., 2021*, *Kutanan et al., 2021*; *Bergström et al., 2022*; *Lipson et al., 2022*; *Vallini et al., 2022*). However, critical re-evaluation of each published graph is an intensive process, and the sample of studies we revisited is diverse enough to identify some general patterns.

Here we present a high-level summary of these analyses. Discussion of individual case studies follows below, and for details see the exposition in Appendix 2.

For 19 out of 22 published graphs we examined, we were able to find at least one, but usually many, graphs of the same complexity (number of groups and admixture events), with an LL score that was nominally better than that of the published graph (see results for 11 selected graphs in *Table 1* and full results for all 22 graphs in *Supplementary file 1*). The 22 graphs were drawn from the 8 publications as there were multiple final graphs presented in some of the publications (*Shinde et al., 2019*; *Sikora et al., 2019*; *Librado et al., 2021*), or we examined selected intermediates in the model construction process (*Bergström et al., 2020*; *Lazaridis et al., 2014*; *Lipson et al., 2020b*; *Wang et al., 2021*), or we introduced an outgroup not used in the original study (*Hajdinjak et al., 2021*; *Sikora et al., 2019*), or we tested additional graph complexity classes dropping 'unnecessary' admixture events (*Lipson et al., 2020b*; *Sikora et al., 2019*).

These alternative graphs often fit not significantly better than the published one after taking into account variability across single-nucleotide polymorphisms (SNPs) via bootstrapping. In the following cases, at least one model that fits significantly better than the published one according to our bootstrap model comparison method was found: the Bergström et al. and Lazaridis et al. 7-population graphs; the *Librado et al., 2021* graph with 3 admixture events; the Hajdinjak et al. graphs with or without adding a chimpanzee outgroup; the *Lipson et al., 2020b*. intermediate graphs with 7 groups and 4 admixture events and with 10 groups and 8 admixture events; the Wang et al. 12-population graph; and the Sikora et al. graphs for West Eurasians and for East Eurasians with 10 or 6 admixture events (*Supplementary file 1*). In nearly all cases (except for the Lazaridis et al. six-population graph, Shinde et al. graph with eight populations and three admixture events, and the *Librado et al., 2021*. graph with four admixture events), we also identified many additional graphs that fit the data not significantly worse than the published ones. In every example, some of these graphs have topologies that are qualitatively different in important ways from those of the published graphs. Features such as which populations are admixed or unadmixed, direction of gene flow, or the order of split events, if not constrained a priori, are generally not the same between alternative fitting models for the same populations. This result agrees with the expectation from our exploration of simulated AGs (*Figure 1*). While some of these graphs can be rejected since their topologies appear highly unlikely because of non-genetic or unrelated genetic evidence, for all of the publications except one (*Shinde et al., 2019*), there are alternative equally-well-or-better-fitting graphs we identified and examined manually that differ in qualitatively important ways with regard to the implications about history, are temporally plausible (for instance, very ancient populations do not receive gene flows from sources closely related to much less ancient groups), and not obviously wrong based on other lines of evidence. These findings and the results on simulated AGs suggest that complex AG models, even with a very good fit to the data, often differ in important ways from true population histories.

The previous statements are valid if the original parsimony constraints are applied, that is, if the graph complexity (the number of admixture events) is not altered. Below in selected case studies (Shinde et al., *Librado et al., 2021*.) we also explore the effect of relaxing the parsimony constraint. *Table 1* and *Figure 2* summarize these results for one or a few graphs from each publication, while

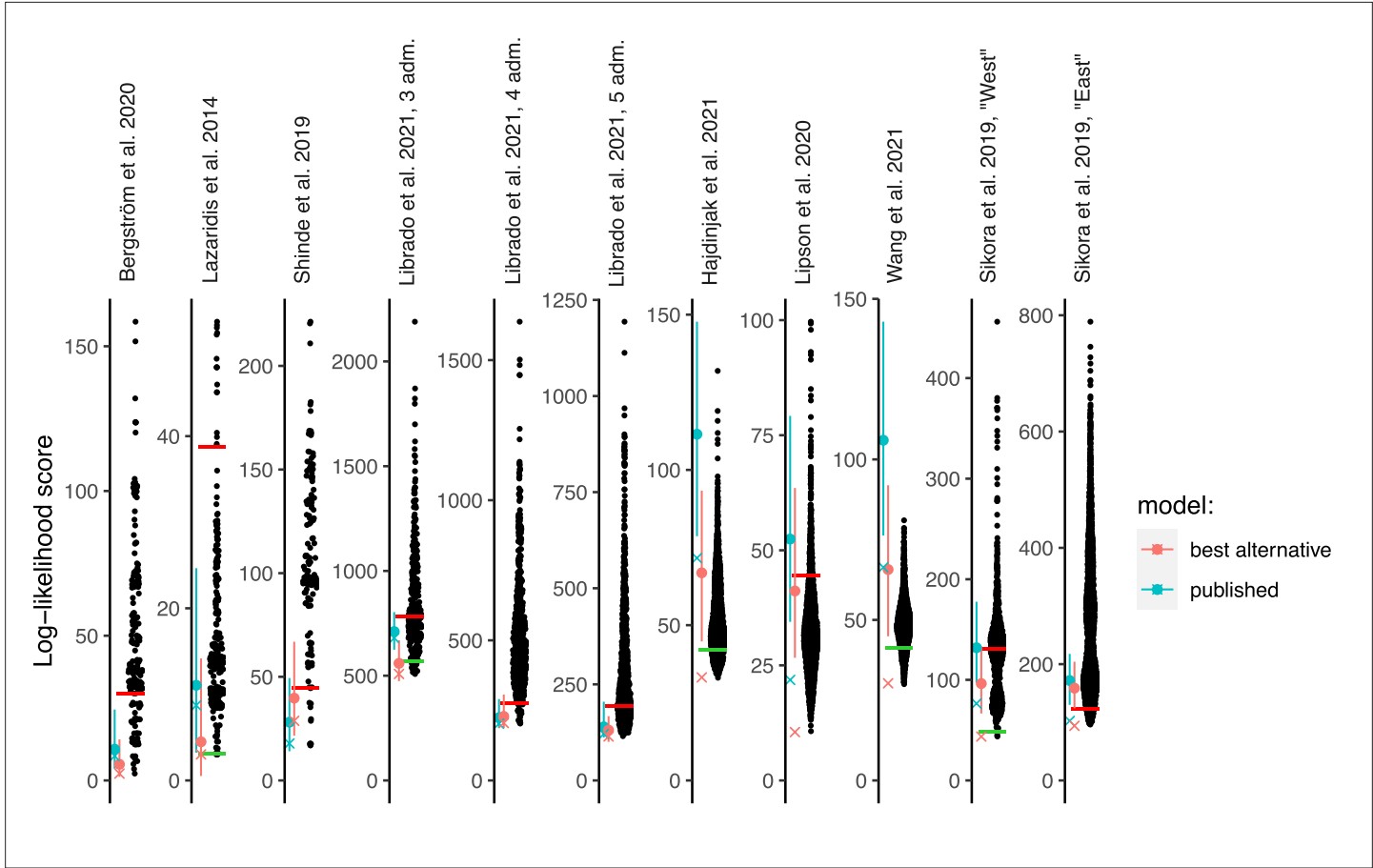

**Figure 2.** Log-likelihood (LL) scores of published graphs (those shown in *Table 1*) and automatically inferred graphs. Each dot represents the LL score of a best-fitting graph from one *findGraphs* iteration (low values of the score indicate a better fit); only topologically distinct graphs are shown. LL scores for the published models and best-fitting alternative models found are shown by blue and pink x's, respectively. Bootstrap distributions of LL scores for these models (vertical lines, 90% CI) and their medians (solid dots) are also shown. Lower scores of the fits obtained using all single-nucleotide polymorphisms (SNPs), relative to the bootstrap distribution, indicate overfitting. Green and red horizontal lines show the approximate locations where newly found models consistently have fits significantly better or worse, respectively, than those of the published model. In the case of the Bergström et al., Lazaridis et al., and Hajdinjak et al. studies, one or more worst-fitting models were removed for improving the visualization. The setups shown here (population composition, number of groups and admixture events, topology search constraints) match those shown in *Table 1*.

*Supplementary file 1* contains the full results for all studied graphs and setups. *Table 2* summarizes our assessment of inferences in the original publications that were supported by the published graphs.

To identify alternative models, we ran many iterations of *findGraphs* for each set of input populations, constraints, and the number of admixture events being fit to the data, and we selected the best-fitting graph in each iteration, that is, a graph with the lowest LL score. Each iteration was initiated from a random graph. The algorithm is non-deterministic so that in each iteration it takes a different trajectory through graph space, possibly terminating in a different final best graph. The number of admixture events in the initial random graphs and in the output graphs was always kept equal to that of the published graph. For each example, we counted how many distinct topologies were found with significantly or non-significantly better or worse LL scores than that of the published graph (*Table 1*, *Supplementary file 1*). To obtain a formally correct comparison of model fit, the published graph and each alternative model were fitted to resampled replicates of the dataset and the resulting LL score distributions were compared (see Appendix 1, Sections 1.B.3 and 2.E). As shown in *Figure 2*, for four of the eight publications we re-analyzed, the LL score of the published graph run on the full data is better than almost all the bootstrap replicates on the same data (it falls below the fifth percentile), which is a sign of overfitting, and underscores the importance of applying bootstrap to assess the robustness of fitted models and conclusions drawn from them.

**Table 2.** Features of the published admixture graphs (AGs) that support inferences in the original studies and the level of their support in our re-analysis. The table lists key features whose support we assessed in sets of alternative well-fitting and temporally plausible models generated by *findGraphs*. Since this assessment had to be performed manually, only in two cases (marked by asterisks) all models fitting better and non-significantly worse than the published one were scrutinized; in other cases only a subset of best-fitting models was examined (see the respective sections in Appendix 2 for details).

| Study | Groups/ admixture events | Features of the published model supported by all temporally plausible alternative models generated by *findGraphs* that we scrutinized | Features of the published model lacking universal support among temporally plausible alternative models generated by *findGraphs* |
|---|---|---|---|
| **Bergström et al., 2020** | 7/3* | Early divergence of domesticated dog lineages (prior to the date of the Karelian dog, 10,900 ya). | Siberian (Baikal), American, and East Mediterranean dog lineages are unadmixed, and the West European (Germany Early Neolithic), East European (Karelia), and dogs of Southeast Asian origin (New Guinea singing dog) are admixed. |
| **Lazaridis et al., 2014** | 7/4 | Present-day Europeans represent a mixture of three ancestral sources related to the following groups: Mal'ta (MA1), West European hunter-gatherers, and early European farmers. | N/A |
| | 8/3* | (1) Iranian farmer-related ancestry in the Indus Periphery group is not derived from the Hajji Firuz Neolithic or Tepe Hissar Chalcolithic groups. (2)There is Asian-related ancestry in the Indus Periphery group. | N/A |
| **Shinde et al., 2019** | 8/4 | (2) There is Asian ancestry in the Indus Periphery group. | (1) Iranian farmer-related ancestry in the Indus Periphery group is not derived from the Hajji Firuz Neolithic or Tepe Hissar Chalcolithic groups. |
| **Librado et al., 2021** | 10/8 or 9 | (2) DOM2 and C-PONT are sister groups (they form a clade). (4) There is gene flow from a deep-branching ghost group to the NEO-ANA group. | (1) NEO-ANA-related admixture is absent in the DOM2 group. (3) There is no gene flow connecting the CWC group and the cluster associated with Yamnaya horses and horses of the later Sintashta culture whose ancestry is maximized in the Western Steppe (DOM2, C-PONT, TURG). (5) Tarpan is a mixture of a CWC-related and a DOM2-related lineage. |
| **Hajdinjak et al., 2021** | 12/8 | (3) The Vestonice16 lineage is a mixture of a Sunghir-related and a BK1653-related lineage. | (1) There are gene flows from the lineage found in the ~45,000- to 43,000-year-old Bacho Kiro Initial Upper Paleolithic (IUP)-associated lineage to the Ust'-Ishim, Tianyuan, and GoyetQ116-1 lineages. (2) The ~35,000-year-old Bacho Kiro Cave individual BK1653 belonged to a population that was related, but not identical, to that of the GoyetQ116-1 individual. |
| **Lipson et al., 2020b** | 12/11 | N/A | (1) A lineage maximized in present-day West African groups (Lemande, Mende, and Yoruba) also contributed some ancestry to the ancient Shum Laka individual and to present-day Biaka and Mbuti. (2) Another ancestry component in Shum Laka is a deep-branching lineage maximized in the rainforest hunter-gatherers Biaka and Mbuti. (3) 'Super-archaic' ancestry (i.e., diverging at the modern human/Neanderthal split point or deeper) contributed to Biaka, Mbuti, Shum Laka, Lemande, Mende, and Yoruba. (4) A ghost modern human lineage (or lineages) contributed to Agaw, Mota, Biaka, Mbuti, Shum Laka, Lemande, Mende, and Yoruba. |
| **Wang et al., 2021** | 12/8 | N/A | Admixture from a source related to Andamanese hunter-gatherers is almost universal in East Asians, occurring in the Jomon, Tibetan, Upper Yellow River Late Neolithic, West Liao River Late Neolithic, Taiwan Iron Age, and China Island Early Neolithic (Liangdao) groups. |
| **Sikora et al., 2019 'West'** | 13/6 | N/A | The Mal'ta (MA1_ANE) lineage received gene flow from the Caucasus hunter-gatherer (CaucasusHG_LP or CHG) lineage. |

*Table 2 continued on next page*

Table 2 continued

| Study | Groups/ admixture events | Features of the published model supported by all temporally plausible alternative models generated by *findGraphs* that we scrutinized | Features of the published model lacking universal support among temporally plausible alternative models generated by *findGraphs* |
|---|---|---|---|
| *Sikora et al., 2019* 'East' | 14/6 | (2) European-related ancestry in the Kolyma, USR1, and Clovis lineages is closer to Mal'ta than to Yana. | (1)The Mal'ta (MA1_ANE) and Yana (Yana_UP) lineages received gene flow from a common East Asian-associated source diverging before the ones contributing to the Devil's Cave (DevilsCave_N), Kolyma (Kolyma_M), USR1 (Alaska_LP), and Clovis (Clovis_LP) lineages. (3)The Devil's Cave lineage received no European-related gene flows, and Kolyma has less European-related ancestry than ancient Americans (USR1 and Clovis). |

The fraction of graphs with scores better than the score of the published graph should not be overinterpreted, as it is influenced by the *findGraphs* algorithm, which does not guarantee ergodic sampling from the space of well-fitting AGs. In particular, it is possible that despite *findGraph*'s strategies for efficiently identifying classes of well-fitting AGs (see Appendix 1, Sections 1.B.1 and 2.C), it has a bias toward missing particular classes of graph topologies. However, even one alternative graph which is not significantly worse-fitting than the published graph suggests that we are not able to identify a single best-fitting model. Many of these alternatives, despite providing a good fit to the data, appear unlikely, for example, because they suggest that Paleolithic-era humans are mixed between different lineages closely related to present-day humans. We were mainly interested in alternative models which are also plausible, and so we constrained the space of allowed topologies in *findGraphs* to those we considered plausible a priori, in cases where this was necessary for reducing the search space size. Constraints were either integrated into the topology search itself, or were applied to outcomes of unconstrained searches, as detailed below.

Below we summarize our key findings and the methodological implications from our re-analysis of the eight published datasets. For more detailed discussions see Appendix 2.

## 1. *Bergström et al., 2020*

The AG for ancient and present-day dogs in Figure 1e of *Bergström et al., 2020* includes an outgroup, six other groups and three admixture events (*Figure 3a*, *Figure 3—source data 1*). A best-fitting newly found graph fits the data nominally better than the published one (two-tailed empirical p-value = 0.332), and it bears a closer resemblance to the human population history (*Figure 3—source data 2*). In this new seven-population model (*Figure 3a*), both American and Siberian dog lineages represent a mixture between groups related to the Asian and East European dog lineages, and robust genetic results suggest that in the time horizon investigated in the original publication (after ca. 10,900 years ago) nearly all Siberian (*Jeong et al., 2019*; *Sikora et al., 2019*) and all American (*Raghavan et al., 2014*; *Raghavan et al., 2015*; *Moreno-Mayar et al., 2018*) human populations were admixed between groups most closely related to Europeans and East Asians. According to this model, East Mediterranean dogs are modeled as a mixture of a basal branch (splitting deeper than the divergence of the Asian and European dogs) and West European dogs, again in agreement with current models of genetic history of West Asian human populations who are modeled as a mixture of 'basal Eurasians' and West European hunter–gatherers (*Lazaridis et al., 2016*; *Lipson et al., 2017*). Although greater congruence with human history increases the plausibility of *findGraph*'s newly identified model relative to the published model, to make unbiased comparisons between the history of the two species, model selection should be done strictly independently for each species, and so the genetic data alone does not favor one model more than another.

To explain why the original paper on the population history of dogs missed the model that *findGraphs* identified, we observe that the *Bergström et al., 2020* AG search was exhaustive under the parsimony constraint (no more than two admixture events for six populations, with the seventh group added at a later stage without an exhaustive topology search), and thus missed the potentially true topology including three admixture events for these six populations. This case study also illustrates that even in a relatively low complexity context (seven groups and three admixture events) applying manual approaches for finding optimal models is risky. When any new group such as an Early Neolithic dog from Germany is added to the model, it may introduce crucial new constraints into the system, and re-exploring the whole graph space in an automated way is necessary to avoid missing the true model. In contrast, mapping a newly added group on a simple skeleton graph (even when that skeleton is a uniquely best-fitting model like in *Bergström et al., 2020*) may yield a topology that is at odds with the true history.

## 2. *Lazaridis et al., 2014*

The graph in Figure 3 (in *Lazaridis et al., 2014*) suggested that present-day Europeans are derived from at least three populations that are very much differentiated genetically: West European hunter–gatherers (WHG), early European farmers (EEF), and Siberian hunter–gatherers from the same lineage as that of the Mal'ta boy who lived about 24,000 years ago (MA1). For seven-population graphs with four admixture events, we found 40 out of 306 distinct graphs with a score better than that of the published graph (10 of those graphs are shown in *Figure 3—source data 3*). The best-fitting newly

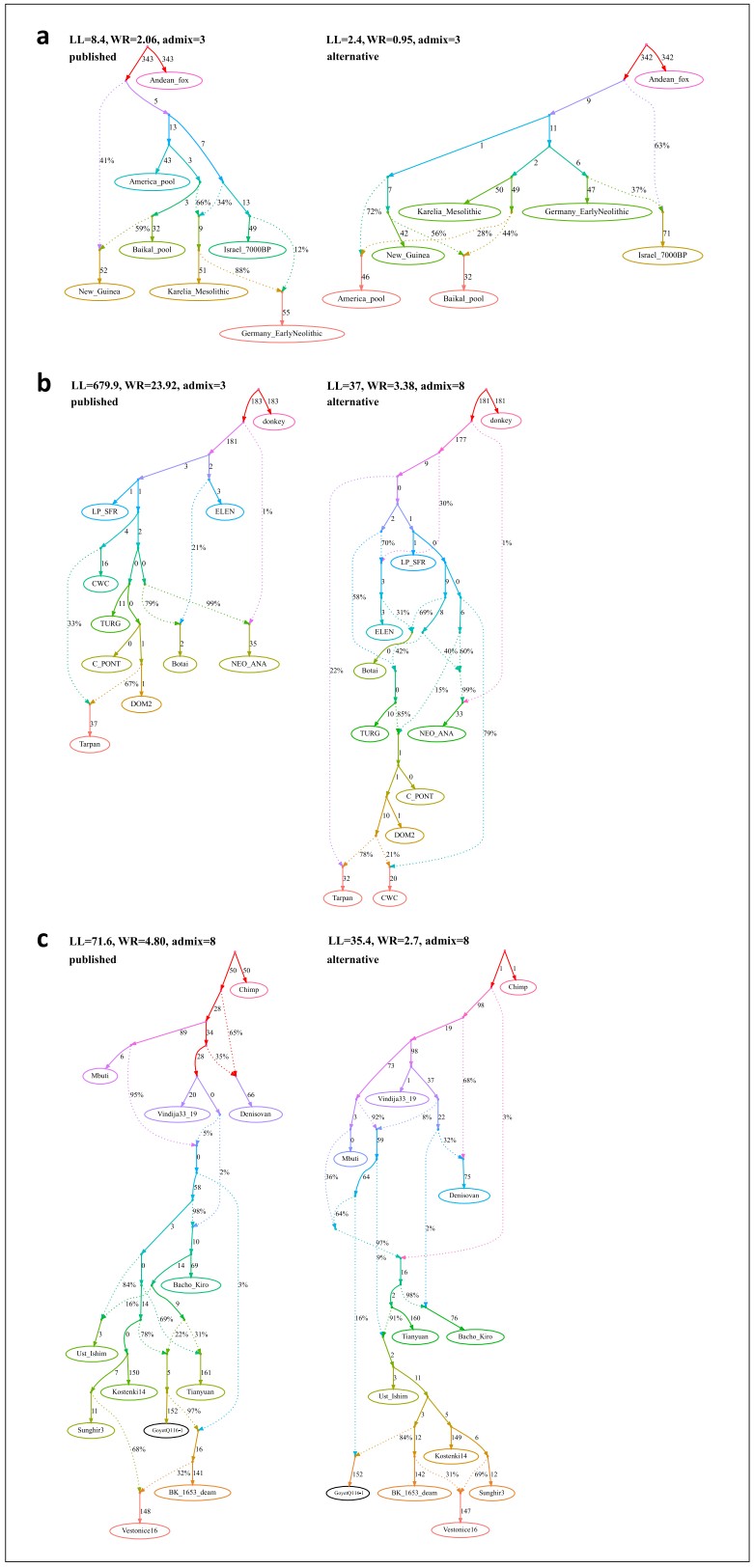

**Figure 3.** Published graphs and selected alternative models from three studies for which we explored alternative admixture graph (AG) fits. In all cases, we selected a temporally plausible alternative model that fits nominally or significantly better than the published model and has important qualitative differences compared to the published model with respect to the interpretation about population relationships. In all but one case, the model has the

*Figure 3 continued on next page*

*Figure 3 continued*

same complexity as the published model shown on the left with respect to the number of admixture events; the exception is the re-analysis of the *Librado et al., 2021* horse dataset since the published model with three admixture events is a poor fit (worst *Z*-score comparing the observed and expected *f*-statistics has an absolute value of 23.9 even when changing the composition of the population groups to increase their homogeneity and improve the fit relative to the composition used in the published study). For this case, we show an alternative model with 8 admixture events that fits well and has important qualitative differences from the point of view of population history interpretation. The existence of well-fitting AG models does not mean that the alternative models are the correct models; however, their identification is important because they prove that alternative reasonable scenarios exist that are qualitatively different from published models. Model parameters (graph edges) that were inferred to be unidentifiable (see Appendix 1, Section 2.F) are plotted in red. (**a**) The graph published by *Bergström et al., 2020* (on the left) and a nominally better fitting graph for dogs that is more congruent to human history (on the right). For both species, Baikal and Native American groups are mixed between European- and East Asian-related lineages, and a 'Basal Eurasian' lineage contributes to West Asian groups; these features are all characteristic of human history but absent in the published dog graph. (**b**) The graph published by *Librado et al., 2021* (modified population composition, on the left) and a significantly better fitting AG that is temporally and geographically plausible (on the right). In contrast to the published graph, in this graph with eight mixture events (the minimum necessary to obtain an acceptable statistical fit to the data), a lineage maximized in horses associated with Yamnaya steppe pastoralists or their Sintashta descendants (C-PONT, TURG, or DOM2) contributes a substantial proportion of ancestry to the horses from the Corded Ware Complex (CWC). Thus, in this model both CWC humans and horses are mixtures of Yamnaya and European farmer-associated lineages. This is qualitatively different from the suggestion that there was no Yamnaya-associated contribution to CWC horses which was a possibility raised in the paper. The AG with eight admixture events is also different from the published model in that it shows a fitting model where the Tarpan horse does not have the history claimed in the study (as an admixture of the CWC and DOM2 horses). (**c**) The graph published by *Hajdinjak et al., 2021* (on the left) and a significantly better fitting AG, but without a specific lineage shared between the Bacho Kiro Initial Upper Paleolithic group and East Asians (on the right). In this model, all the lineages shared between Bacho Kiro IUP and East Asians contributed a large fraction of the ancestry of later European hunter–gatherers as well, and thus this graph does not imply distinctive shared ancestry between the earliest modern humans in Europe and later people in East Asia, and instead could be explained by a quite different and also archaeologically plausible scenario of a primary modern human expansion out of West Asia contributing serially to the major lineages leading to Bacho Kiro, then later East Asians, then Ust'-Ishim, then the primary ancestry in later European hunter–gatherers.

The online version of this article includes the following source data for figure 3:

**Source data 1.** The published (*Bergström et al., 2020*) and alternative admixture graphs for dogs found with *findGraphs*.

**Source data 2.** Alternative admixture graphs for humans found with *findGraphs* for the dataset from *Bergström et al., 2020*.

**Source data 3.** The published admixture graph from *Lazaridis et al., 2014* and alternative graphs found with *findGraphs* (seven populations, four admixture events).

**Source data 4.** The published admixture graph from *Shinde et al., 2019* and alternative graphs found with findGraphs (8 pops., 3 adm. events) relying on the original set of SNPs and group composition, and original (incorrect) algorithm for calculating f-statistics.

**Source data 5.** The published admixture graph from *Shinde et al., 2019* and alternative graphs found with *findGraphs* (eight populations, three admixture events) for the modified group composition and using the updated algorithm for calculating *f*-statistics.

**Source data 6.** Alternative graphs allowing for an additional admixture event found with *findGraphs* for the dataset from *Shinde et al., 2019*: 8 populations, 4 admixture events, the modified group composition, and the updated algorithm for calculating *f*-statistics.

**Source data 7.** The published admixture graphs from *Librado et al., 2021* and alternative graphs found with *findGraphs* (10 populations, 3–5 admixture events) for the modified group composition and using the updated algorithm for calculating *f*-statistics.

**Source data 8.** Published admixture graph from *Hajdinjak et al., 2021* and alternative graphs found with *findGraphs* (12 populations, 8 admixture events).

found model and two other models fit the data significantly better than the published model (*Supplementary file 1*), but their topology is qualitatively very similar to that of the published graph (*Figure 3—source data 3*). In the best-fitting newly found model, French and Karitiana share some drift to the exclusion of MA1, while in the published model the source of MA1-related ancestry in French is closer to MA1 than to Karitiana.

It is important to point out that not all of the 40 alternative graphs that fit nominally or significantly better than the published one are consistent with the conclusion that modern European populations are admixed between three different ancestral populations (*Figure 3—source data 3*). According to the fifth alternative graph in *Figure 3—source data 3* that fits nominally better than the published model (p-value = 0.464), the present-day European population was formed by admixture of an MA1-related lineage and a European Neolithic-related lineage, with no WHG contribution. Of course, other lines of evidence make it clear that LBK Stuttgart is a mixture of Anatolian farmer- and WHG Loschbour-related ancestry (e.g., *Lazaridis et al., 2016*; *Lipson et al., 2017*), thus providing external information in favor of the *Lazaridis et al., 2014* model. The use of such ancillary information in concert with graph exploration is important in order to obtain more confident inferences about population history taking advantage of AGs. We also note that a large group of newly found models (247 graphs) fits not significantly worse than the published one (*Supplementary file 1*), and those are topologically diverse. Thus, strictly speaking, the AG method on the given dataset cannot be used to prove that the published model is the only one fitting the data.

## 3. *Shinde et al., 2019*

The skeleton AG in the original study (*Shinde et al., 2019*) was constructed manually, and subsequently all possible branching orders (105) within the five-population Iranian farmer/herder-related clade were tested. The published model (*Figure 3* in that study) included nine groups and three admixture events, but one group (Belt Cave Mesolithic) had a very high missing data rate. Following the approach of the published paper, we repeated *findGraphs* analysis both with and without the Belt Cave individual. Thus, we initially explored the following topology classes: 9 groups with 3 admixture events on ca. 19,000 polymorphic sites and 8 groups with 3 admixture events on ca. 470,000 sites (*Figure 3—source data 4*, *Supplementary file 1*). The finding that the predominant ancestry component of the Indus Periphery group was the most basal branch in the Iranian farmer clade was a prominent claim of the original study (*Shinde et al., 2019*). This finding if correct is important, since it implies that the Iranian-related ancestry in the Indus Valley Civilization genetic grouping (which is the same group as Indus Periphery or IP) split from the Iranian-related ancestry in the first Iranian plateau farmers before the date of the Hajji Firuz farmers, who at ~8000 years ago are among the earliest people living on the Iranian plateau known to have grown West Asian crops. The ancient DNA record combined with radiocarbon dating evidence suggests that beginning around the time of the Hajji Firuz farmers, both West Asian domesticated plants such as wheat and barley, and Anatolian farmer-related admixture, began spreading eastward across the Iranian plateau. If the Iranian-related ancestry in IP was spread eastward into the Indus Valley across the Iranian plateau as part of the same agriculturally associated expansion—perhaps brought by people speaking Indo-European languages as well as introducing West Asian crops—then we would expect to see at least some of the Iranian-related ancestry in IP being a clade with that in Hajji Firuz relative to Ganj Dareh.

The following groups were admixed by default in the graph models compared in the original study: Hajji Firuz Neolithic (labeled 'Chalcolithic' in that study but the dates are Neolithic) and Tepe Hissar Chalcolithic were considered as mixtures of an Anatolian farmer-related lineage and an Iranian farmer-related lineage. Indus Periphery was assumed to be a mixture of an Andamanese-related lineage representing ancient South Indians (ASI) and an Iranian farmer-related lineage. The original study differed from ours since these constraints were introduced manually, but we wanted our topology search to be automatic and to explore a wider range of parameter space.

To provide power to detect negative $f_3$-statistics useful for constraining the model search, we introduced several modifications to the original group composition (described in Appendix 2) and a new algorithm that makes it possible to compute negative $f_3$-statistics on pseudo-haploid data, but at a cost of removing sites with only one chromosome genotyped in any non-singleton population (see Appendix 1, Section 1.A). We repeated topology search with this set of $f$-statistics providing additional constraints, performing 4,000 runs of the *findGraphs* algorithm. The Mota ancient African

individual was set as an outgroup and three admixture events were allowed in the eight-population graph. Among 4,000 resulting graphs (one from each *findGraphs* run), 144 were distinct topologically, and the published model was recovered in 13 runs of 4,000 (*Supplementary file 1*). Only four distinct topologies fitting nominally better than the published one were found, and those had LL scores almost identical to that of the published AG. These four alternative models (*Figure 3—source data 5b*) shared all topologically important features of the published model (*Figure 3—source data 5*). Five other topologies differed in important ways from the published one and emerged as fitting the data worse, but not significantly worse, than the published one (*Figure 3—source data 5c*).

But in fact, the AG analysis reported above may not be an adequate exploration of the problem. Although absolute fits of the best models found are good (WR = 2.5 SE), the parsimony constraint allowing only three admixture events precluded correct modeling of basal Eurasian ancestry shared by all West Asian groups (*Lazaridis et al., 2016*) or of the Indus Periphery group itself, for which a more complex 3-component admixture model was proposed elsewhere (*Narasimhan et al., 2019*). Concerned that this oversimplification could be causing our search to miss important classes of models, we explored *qpAdm* models for the Indus Periphery group, following the protocol by *Narasimhan et al., 2019* (see the dataset composition in *Supplementary file 4*). Our *qpAdm* results (*Supplementary files 5 and 6*, Appendix 2) show that the parsimony assumption that was made when constructing the AG analysis in *Shinde et al., 2019* is contradicted by *f*-statistic evidence since the simplest fitting *qpAdm* model for the IP group includes four ancestry sources, not two (Indus Periphery = Ganj Dareh Neolithic + Onge (ASI) + WSHG + Anatolia Neolithic), and indeed Narasimhan et al. themselves showed this when they presented a *qpAdm* model that was more complex (Ganj Dareh Neolithic + Onge (ASI) + WSHG) than the one used for constraining the AG model comparison (Ganj Dareh Neolithic + Onge (ASI)).

To explore how the parsimony constraint influences results, we allowed four admixture events in the eight-population graph (*Supplementary file 1*). Among 4,000 resulting graphs (one from each *findGraphs* run), 443 were distinct topologically, and 270 had WRs between 2 and 3 SE, that is, fitted the data well. In *Figure 3—source data 6b*, we show four graphs with four admixture events that model the Indus Periphery group as a mixture of three or four sources, with a significant fraction of its ancestry derived from the Hajji Firuz Neolithic or Tepe Hissar Chalcolithic lineages including both Iranian and Anatolian ancestries. The fits of these models are just slightly different (e.g., LL = 11.7 vs 9.3, both WRs = 2.4 SE) from that of the best-fitting model (*Figure 3—source data 6a*), and are similar to that of the simpler published graph. Besides these four illustrative graphs, dozens of topologies with very different models for the Indus Periphery group fit the data approximately equally well, suggesting that there is no useful signal in this type of AG analysis when the parsimony constraint is relaxed (this finding is similar to that in our re-analysis of the dog AG in *Bergström et al., 2020*, where relaxation of the parsimony constraint identified equally well-fitting AGs that were very different with regard to their inferences about population history). These results show that at least with regard to the AG analysis, a key historical conclusion of the study (that the predominant genetic component in the Indus Periphery lineage diverged from the Iranian clade prior to the date of the Ganj Dareh Neolithic group at ca. 10 kya and thus prior to the arrival of West Asian crops and Anatolian genetics in Iran) depends on the parsimony assumption, but the preference for three admixture events instead of four is hard to justify based on archaeological or other arguments.

Why did the *Shinde et al., 2019* AG analysis find support for the IP Iranian-related lineage being the first to split, while our *findGraphs* analysis did not? *Shinde et al., 2019* study sought to carry out a systematic exploration of the AG space in the same spirit as *findGraphs*—one of only a few papers in the literature where there has been an attempt to do so—and thus this qualitative difference in findings is notable. We hypothesize that the inconsistency reflects the fact that the deeply diverging WSHG-related ancestry (*Narasimhan et al., 2019*) present in the IP genetic grouping at a level of ca. 10% was not taken into account explicitly neither in the AG analysis nor in the admixture-corrected $f_4$-symmetry tests also reported in *Shinde et al., 2019*. The difference in qualitative conclusions may also reflect the fact that the Shinde et al. study was distinguishing between fitting models relying on an LL difference threshold of 4 units (based on the AIC). As discussed in Appendix 1, AIC is not applicable to AGs where the number of independent model parameters is topology-dependent even if the numbers of groups and admixture events are fixed, and models compared with AIC should have the same number of parameters. Thus, we believe that the analysis by Shinde et al. was over-optimistic

(as compared to the bootstrap model comparison method we use) about being able to reject models that were in fact plausible using its AG fitting setup.

## 4. *Librado et al., 2021*

In contrast to the other studies revisited in our work, the AG published by *Librado et al., 2021* was inferred automatically using *OrientAGraph*. Models with three (Figure 3b in that study) and zero to five (Ext. Data Figure 5a–d in that study) admixture events were shown. The dataset included 10 populations (9 horse populations and donkey as an outgroup) and was based on 7.4 million polymorphic transversion sites with no missing data at the group level. Unlike all the other AGs we re-evaluate in this study whose fits to the data were evaluated in the published studies using *qpGraph*, the topologies published in *Librado et al., 2021* were not evaluated for statistical goodness-of-fit, and in fact fit the *f*-statistic data so poorly that even simple statistics show they cannot be correct (*Figure 3b*, *Figure 3—source data 7a, c, e,*, *Supplementary file 1*). In this case, the approach of using *findGraphs* to identify alternative topologies with the same number of admixture events that fit the data better is meaningless, as both the published models and the alternative models do not have enough degrees of freedom to accommodate the complexity present in the real data (*Figure 3—source data 7*). In particular, we found that WR of the published model with three admixture events is 23.9 SE (*Figure 3—source data 7a*).

For this reason, we moved to topology searches in more complex model spaces incorporating six to nine admixture events. Temporally plausible models with even a modest fit (WR between 3 and 4 SE) were encountered only among models with eight and nine admixture events (*Figure 3—source data 7j–r*). *Librado et al., 2021* discussed five inferences relying fully or partially on their published AGs reported in that study (*Table 2*). The simplest temporally plausible and best-fitting (WR = 3.4 SE) model we found (eight admixture events, see *Figure 3b* and the first model in *Figure 3—source data 7j*) supports inferences 2 and 4, and is incompatible with inferences 1, 3, and 5 (*Table 2*). We consider this model to be plausible also from the geographical perspective (see Appendix 2 for an interpretation of this topology). We are not arguing here that this AG represents the true history; in fact, it is highly unlikely to be the truth, given how large the space of all possible admixture events is and how much admixture evidently occurred relating all these groups (which makes finding the true model extremely unlikely, see the results on simulated data in *Figure 1* and *Appendix 1—figure 2*). However, our set of 16 temporally plausible and fitting (WR < 4 SE) models with eight or nine admixture events (*Figure 3—source data 7j–r*) is consistent with some features of the published graph being stable: the features (2) that DOM2 and C-PONT are sister groups, and (4) that there was a gene flow from a deep-branching ghost group to NEO-ANA (*Table 2*).

Equally important is our finding that there are plausible models that are inconsistent with other inferences in *Librado et al., 2021*. (*Table 2*). For example, 13 of these 16 models are inconsistent with the suggestion that there was no gene flow connecting the CWC group and the cluster maximized in the Western steppe (DOM2, C-PONT, and TURG) (*Figure 3—source data 7j–r*). In the eight-admixture-event best-fitting AG (*Figure 3b*, the first model in *Figure 3—source data 7j*), CWC actually derives appreciable ancestry from the early domestic horse lineage (DOM2) associated with the Sintashta culture to the exclusion of the more distant Yamnaya-associated TURG and C_PONT horses. This scenario presents a parallel to the one observed in humans, with individuals associated with the CWC receiving admixture from Steppe pastoralists albeit in different proportions: ~75% for humans, versus ~20% in horses. These models specifying a substantial Steppe horse contribution to CWC horses are inconsistent with the inference in *Librado et al., 2021*. that 'Our results reject the commonly held association between horseback riding and the massive expansion of Yamnaya steppe pastoralists into Europe around 3000 BC'. We are not aware of other lines of evidence in the paper (apart from the fitted AG) that support the claim of no Yamnaya horse impact on CWC horses.

Another example of a feature of the published graph that turned out to be unstable is the model for the Tarpan horse. Only 8 of 16 temporally plausible and fitting models (*Figure 3—source data 7j–r*) support the conclusion by Librado et al. that the Tarpan is a mixture of a DOM2-related and a CWC-related lineage. The other eight models suggest that Tarpan is a mixture of a deep lineage and a DOM2-related lineage (*Figure 3b*, the first model in *Figure 3—source data 7j*), echoing a hypothesis that Tarpan may be a hybrid with Przewalski-related horses not represented in the AG (*Librado et al., 2021*). Again, we are not arguing here that our alternative model is right—indeed we are nearly

certain it is wrong in important aspects—but we are merely pointing out that the complexity of the AG space means that qualitatively quite different conclusions are compatible with the statistics fitted in the published paper.

## 5. *Hajdinjak et al., 2021*

The AG inferred by Hajdinjak et al. was constructed manually and incorporated 11 groups and 8 admixture events (Figure 2d in the original study). Most (71.4%) models found with *findGraphs* fit nominally better, and 15.7% fit significantly better than the published model (*Table 1*, *Supplementary file 1*, *Figure 2*), which has a poor absolute fit on this set of sites and groups (WR = 4.8 SE, *Figure 3c*, *Figure 3—source data 8*). The statistics described above and the fact that LL scores on all sites lie outside of the LL distribution on resampled datasets (*Figure 2*) suggest that models in this complexity class are overfitted, but the published topology emerged as fitting relatively poorly. Overfitting arises naturally during manual graph construction as performed in many studies (not only in *Hajdinjak et al., 2021*, but also in *Fu et al., 2016*; *Skoglund et al., 2016*; *Yang et al., 2017*; *Posth et al., 2018*; *McColl et al., 2018*; *Moreno-Mayar et al., 2018*; *Tambets et al., 2018*; *van de Loosdrecht et al., 2018*; *Flegontov et al., 2019*; *Sikora et al., 2019*; *Wang et al., 2019*; *Lipson et al., 2020b*; *Shinde et al., 2019*; *Yang et al., 2020*; *Wang et al., 2021*). The graph grew one group at a time, and each newly added group was mapped on to the pre-existing skeleton graph as unadmixed or as a two-way mixture. Another requirement was that all intermediate graphs have good absolute fits (WR below 3 or 4 SE). When the model-building process is constrained in a particular path and fits of all intermediates are required to be good, unnecessary admixture events are often added along the way, and the resulting graph belongs to a complexity class in which models are overfitted.

*Hajdinjak et al., 2021*'s published graph had three notable features that were interpreted by the authors and used to support some conclusions of the study (*Table 2*), with the following feature considered the most important: there are gene flows from the lineage found in the ~45,000- to 43,000-year-old Bacho Kiro Initial Upper Paleolithic (IUP) individuals to the Ust'-Ishim, Tianyuan, and GoyetQ116-1 lineages. We identified 1,421 topologies fitting nominally or significantly better than the published model and moved on to inspect 50 best-fitting topologies for temporal plausibility (all of them fitting significantly better than the published model). All non-African individuals included in the model are Upper Paleolithic and their dates are not drastically different in relative terms, from ca. 45 to 30 kya (1,000 years before present). Nevertheless, we considered most gene flows from later- to earlier-attested lineages as temporally implausible (for instance, GoyetQ116-1 (~35 kya) → Ust'-Ishim (~44 kya), Sunghir III (34.5 kya) → Tianyuan (40 kya), etc.) since they imply great antiquity of the later-attested lineages and of all lineages derived from them at least partially.

Of the 50 topologies inspected, 32 were considered temporally plausible. Of those topologies, none supported the finding of gene flows from the Bacho Kiro IUP lineage specifically into all three of the Ust'-Ishim, Tianyuan, and GoyetQ116-1 lineages. A total of 17 topologies supported features 2 and 3 but were inconsistent with feature 1; and 14 topologies supported feature 3 only (*Table 2*). Best-fitting representatives of each of these topology classes are shown along with the published model in *Figure 3—source data 8*. Considering topological diversity among models that are temporally plausible, conform to current knowledge about relationships between modern and archaic humans, and fit significantly better than the published model, we conclude that feature 3 is probably robust but other details of the fitted AG in Hajdinjak et al.—for example, gene flows to the Ust'-Ishim, Tianyuan, and Goyet Q116-1 lineages from sources sharing drift exclusively with the Upper Paleolithic Bacho Kiro lineage—should not be interpreted as providing meaningful inferences about population history of Upper Paleolithic modern humans.

A central finding of Hajdinjak et al. is that the Bacho Kiro IUP group shares more alleles with present-day East Asians than with Upper Paleolithic Holocene Europeans despite coming from Europe. Specifically, the study documents significantly positive statistics of the form $D$(an Asian group, Kostenki14; Bacho Kiro IUP, Mbuti). Hajdinjak et al.'s interpretation of this observation is that 'there was at least some continuity between the earliest modern humans in Europe [Bacho Kiro IUP] and later people in Eurasia [East Asians]'. However, a significant $D$- or $f_4$-statistic can have multiple explanations. The statistic $f_4$ (Tianyuan, Kostenki14; Bacho Kiro IUP, Mbuti) is fitted equally well by the published 12-population AG ($Z$-score for the difference between the observed and fitted statistics = 0.64) and by, for example, the AG in *Figure 3c* ($Z$-score = 0.94). Under the latter model that fits the

data significantly better than the published model (p-value = 0.02), the Bacho Kiro IUP and Tianyuan branches are not connected by a gene flow and do not receive gene flows from a third common source, but the common ancestor of Ust'-Ishim and all European Paleolithic lineages receives an 8% gene flow from a divergent modern human lineage splitting deeper than Bacho Kiro IUP and Tianyuan (*Figure 3c*, *Figure 3—source data 8c*). This scenario or some version of it seems archaeologically and geographically plausible and is not disproven by any other line of genetic or non-genetic evidence of which we are aware. It could correspond to a scenario where a primary modern human expansion out of West Asia contributed serially to the major lineages leading to Bacho Kiro, then later East Asians, then Ust'-Ishim, and finally the primary ancestry in later European hunter–gatherers. This has a very different interpretation from the scenario of distinctive shared ancestry between the earliest modern humans in Europe such as Bacho Kiro IUP and later people in East Asia—to the exclusion of later European hunter–gatherers—that is suggested by the Hajdinjak et al. published graph.

We are not claiming that this specific alternative model is correct—indeed, it is almost certainly not the correct one given the topological complexity of the set of all AGs consistent with the data—but the existence of it and many other models that fit the data makes it clear that we do not yet have a unique historical explanation for the excess sharing of alleles that has been documented between some Upper Paleolithic European groups (Bacho Kiro IUP, *Hajdinjak et al., 2021*, and GoyetQ116-1, *Yang et al., 2017* and *Hajdinjak et al., 2021*) and all East Asians.

## 6. *Lipson et al., 2020b*

The AG in the original study was constructed manually (Extended Data Figure 4 in that study) and is very complex (12 groups and 12 admixture events): it exists in a space of ~$10^{44}$ topologies of this complexity. We note that one admixture event was added by *Lipson et al., 2020b* to account for potential modern DNA contamination in ancient Shum Laka individuals, and removing it caused a negligible difference in the fit of the published model (*Supplementary file 1*). Thus, to decrease the complexity of the graph search space, we considered graphs with 12 groups and 11 admixture events. Among 2,000 newly found models, 11.9% fit nominally (but not significantly) better than the published model (*Table 1*, *Supplementary file 1*, *Figure 2*), and absolute fits of 36.7% of novel models are good (WR <3 SE). These metrics, along with the fact that LL scores on all sites lie outside of the LL distribution on resampled datasets (*Figure 2*), suggest that models in this complexity class, including the published model, are overfitted. Of the AGs we re-evaluate in this study, the graph from *Lipson et al., 2020b* shares with the graphs from *Hajdinjak et al., 2021*; *Sikora et al., 2019*; *Wang et al., 2021*, evidence of being overfitted (*Figure 2*).

Below we discuss four prominent features of the AG published in the original study (that were interpreted by the authors and used to support some conclusions of the study) and the extent to which these features consistently replicate across the large number of fitting 12-population graphs with 11 admixture events (*Table 2*). High topological diversity is observed among temporally plausible newly found AGs (see an example in *Figure 4a* and further topologies in *Figure 4—source data 1*). Considering extreme cases, two AGs completely lacked support for three features of the published graph (*Figure 4a*, *Figure 4—source data 1c*), and one graph supported all four features of the published graph fully (*Figure 4—source data 1q*, the second model). There are some graphs where defining two distinct ancestral lineages maximized in West Africans and in Mbuti and Biaka (features 1 and 2, *Table 2*) is essentially impossible since all or nearly all Africans are modeled as a mixture of at least two deep lineages (see alternative graph no. 4 shown in *Figure 4—source data 1d*, the second model). In some graphs there is no single lineage specific to rainforest hunter–gatherers (Biaka, Mbuti, and Shum Laka) since the primary ancestries in these groups form independent deep branches in the African graph (see *Figure 4a* and graph no. 16 shown in *Figure 4—source data 1j*, the second model). The ghost modern and super-archaic gene flows to Africans also had no universal support in the set of alternative graphs we examined (see, for example, *Figure 4a* and *Figure 4—source data 1c*).

Considering the high degree of topological diversity among models that are temporally plausible, conform to known findings about relationships between modern and archaic humans, and fit nominally better than the published model, we conclude that none of the four AG features from the original study are consistently supported by our re-analysis (*Table 2*). This situation may be attributed to (1) overfitting and/or to (2) the lack of information in the dataset (in the combination of groups and SNP sites) and/or to (3) inherent limitations of *f*-statistics, when distinct topologies predict identical

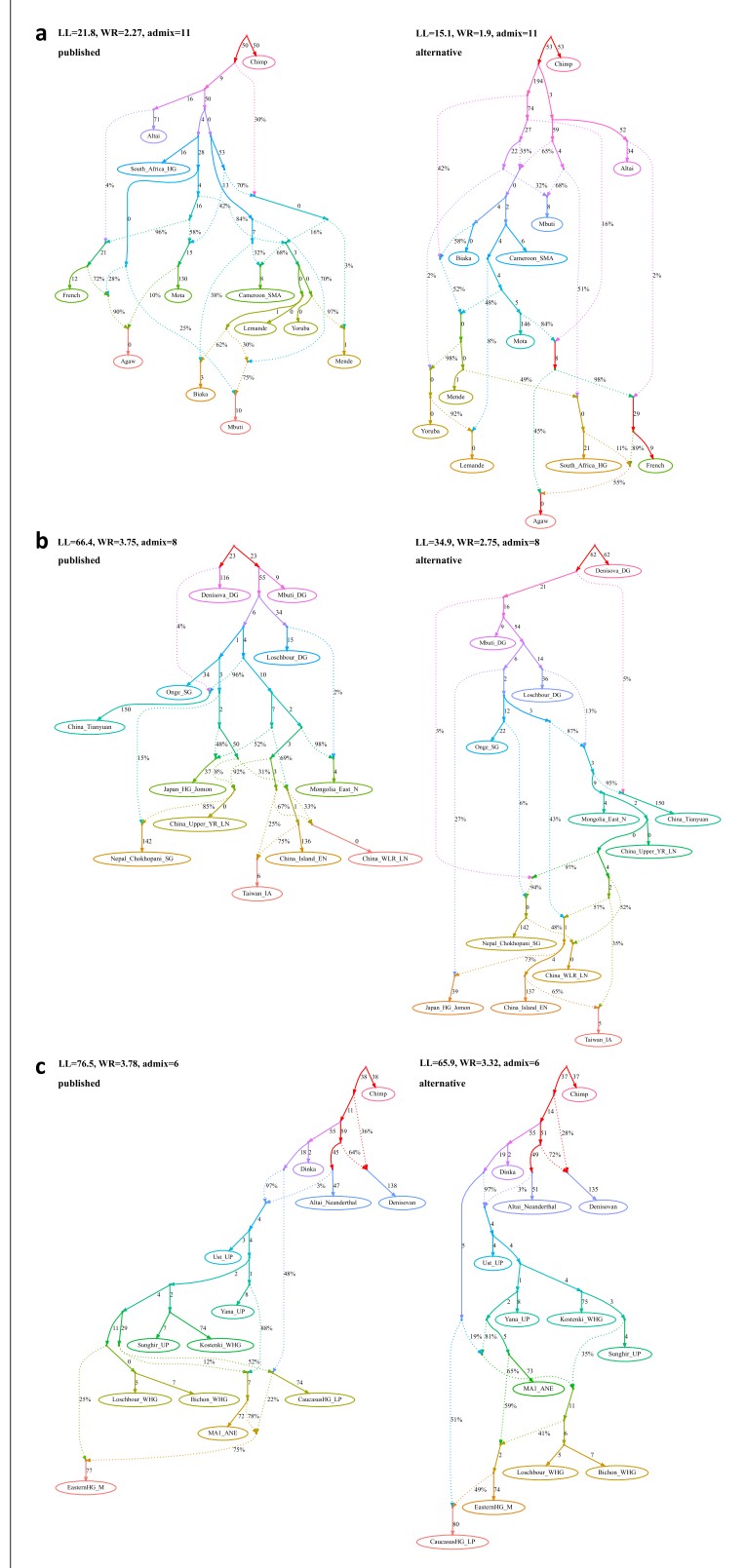

**Figure 4.** Published graphs and selected alternative models from three further studies for which we explored alternative admixture graph (AG) fits. (**a**) The graph published by *Lipson et al., 2020b* (on the left) and a nominally better fitting AG (on the right). In contrast to the published graph, there is no single lineage specific to modern rainforest hunter–gatherers (Biaka and Mbuti) and Shum Laka (Cameroon_SMA). Rather, the primary ancestries

*Figure 4 continued on next page*

*Figure 4 continued*

in each group are separate deep-branching lineages (the deeper lineage they all share is also the source of the majority of ancestry in all anatomically modern humans modeled here). In contrast to the graph in the published paper, there is no West African-maximized ancestry present in mixed form in Biaka, Mbuti, and Shum Laka; archaic admixture is not limited to a subset of Africans but is present in all anatomically modern humans in various proportions; and there is no ghost modern human ancestry in Agaw, Biaka, Lemande, Mbuti, Mende, Mota, Shum Laka, and Yoruba. (**b**) The admixture graph published by *Wang et al., 2021* (on the left) and a significantly better fitting AG meeting the constraints used to inform model building in the published paper (on the right). The finding of Onge-related admixture that is widespread in East Asia suggesting an early peopling via a coastal route is not a feature of this model. (**c**) The admixture graph published by *Sikora et al., 2019* (simplified "Western" graph, on the left) and a nominally better fitting AG (on the right). The striking feature of the AG suggested in the paper whereby Mal'ta (MA1_ANE) derives some ancestry from a CHG-associated lineage is not a feature of this alternative model.

The online version of this article includes the following source data for figure 4:

**Source data 1.** The published admixture graph from *Lipson et al., 2020b* and alternative graphs found with *findGraphs* (12 populations, 11 admixture events) using the updated algorithm for calculating *f*-statistics.

**Source data 2.** The published admixture graph from *Wang et al., 2021* and alternative graphs found with *findGraphs* (12 populations, 8 admixture events) using the updated algorithm for calculating *f*-statistics.

**Source data 3.** The simplified published admixture graph for West Eurasian groups from *Sikora et al., 2019* and alternative graphs found with *findGraphs* (13 populations, 6 admixture events).

**Source data 4.** The simplified published admixture graph for East Eurasian groups from *Sikora et al., 2019* and alternative graphs found with *findGraphs* (14 populations, 6 admixture events).

---

*f*-statistics. Our results highlight the mystery around the highly distinctive genetic ancestry of the Shum Laka individuals themselves, who represent the newly reported data in the *Lipson et al., 2020b* study, and represent a highly important set of genetic datapoints that was not available prior to the study. The ancestral relationships of these four individuals to rainforest hunter–gatherers, and to the primary lineage in present-day West Africans, remains an open question, one whose resolution promises meaningful new insights into modern human population history.

## 7. *Wang et al., 2021*

The AG inferred by *Wang et al., 2021* was constructed manually, and the final graph (Extended Data Figure 6 in the original study) included 12 groups and 8 admixture events. We applied several constraints on the graph space exploration process all of which were shared with the Wang et al. graphs (*Supplementary file 1*). An important feature of the published graphs in *Wang et al., 2021* that was remarked upon in the study is admixture from a source related to Andamanese hunter–gatherers that is almost universal in East Asians (*Table 2*). For example, the abstract states 'Hunter-gatherers from Japan, the Amur River Basin, and people of Neolithic and Iron Age Taiwan and the Tibetan Plateau are linked by a deeply splitting lineage that probably reflects a coastal migration during the Late Pleistocene epoch.' We performed 2,000 *findGraphs* iterations and obtained 1,778 distinct topologies satisfying all the constraints, nearly all of them (1,724) fitting nominally better than the published model, and 12.6% fitting significantly better (*Table 1*, *Supplementary file 1*). The models were ranked by LL scores, and 56 highest-ranking topologies, all of them fitting significantly better than the published one, were assessed for temporal plausibility, and 20 topologies were considered temporally plausible (all of them are shown in *Figure 4—source data 2*). According to these topologies, 0–2 East Asian groups had a fraction of their ancestry derived from a source specifically related to Onge, and 19 topologies included gene flows from the European (Loschbour)-related branch to all 8 East Asian groups (*Figure 4—source data 2*). The inferred topological relationships among East Asians are variable in this group of AGs, and we decided to apply further constraints that guided model ranking and elimination by Wang et al., based on considerations from archaeological evidence, Y chromosome haplogroup divergence patterns, and population split time estimation (see Appendix 2 for details). Applying these three additional constraints, we identified two models (among the 56 subjected to manual inspection) that satisfied all of them. The highest-ranking of those models is shown in *Figure 4b* and *Figure 4—source data 2c* (the second model), and it includes a 13% (deeply) European-related gene flow to the common ancestor of all East Asians, and gene flows from

the Onge-related branch to just two East Asian groups: Nepal Chokhopani and China WLR LN. This model fits the data significantly better than the published model (p-value = 0.028). We do not claim that this is the correct model (indeed we are almost certain that it is not given the high topological diversity of fitting models), but it is not obviously wrong and differs in qualitatively important ways from the published one.

The *Wang et al., 2021* AG provides an illuminating example that helps us to understand the value added by AG construction. The AG construction process in Wang et al. followed a philosophy of not relying entirely on the allele frequency correlation data (not treating the genetic data as independent to explore how much new insight could come from genetic data alone). Instead, the study integrated other lines of genetic evidence as well as linguistic and archaeological insights explicitly into the AG construction process, with the goal of identifying models consistent with multiple lines of evidence. The fact that after this procedure a fitting graph was obtained is not of great interest, as it is essentially always possible to obtain a fit to allele frequency correlation data when enough admixture events are added. The important question is whether any of the emergent features of the graph that were not applied as constraints in the construction process—for example the evidence of ubiquitous Andamanese-related gene flow throughout East Asia suggesting a coastal route expansion that admixed with an interior route expansion proxied by Tianyuan—were stably inferred. Our analysis does not come to this finding consistently among well-fitting and plausible AGs. We conclude that this important feature of the published graph is not supported by *f*-statistic analysis alone (*Table 2*), and indeed we are not aware of a single feature of the *Wang et al., 2021* AG that is stably inferred beyond the constraints applied to build it.

## 8. *Sikora et al., 2019*

Two AGs inferred by *Sikora et al., 2019* were constructed manually based on an SNP set derived from whole-genome shotgun data and incorporated 12 or 13 groups and 10 admixture events (Extended Data Figure 3f in the original study). One graph was focused on West Eurasians, and the other one on East Eurasians, and both included a Neanderthal, a Denisovan, and an African group (Dinka). Although the chimpanzee outgroup was not included in the original graphs, we added it as it drastically constrains the topology search space. In contrast to most other published graphs discussed above, gene flows in the graphs inferred by Sikora et al. do not have equal standing: four low-level gene flows (0–1%) connect the Neanderthal lineage to Upper Paleolithic lineages. We repeated each topology search under two alternative settings: either keeping the number of admixture events at 10 to match the published graphs, or at 6 to match simplified versions of the published graphs lacking these low-level Neanderthal gene flows. We performed that modification to simplify the search space and to alleviate the overfitting problem which becomes severe if 10 gene flows across the graph are allowed (*Supplementary file 1*).

In the case of the "Western" graphs with 6 admixture events, 1,000 topology search iterations were performed, 894 distinct topologies were found, 4 models fit significantly better, and 151 models fit nominally better than the published one (*Table 1*, *Supplementary file 1*). We inspected those 155 topologies and identified 29 topologies (*Figure 4—source data 3*) that are temporally plausible. Sikora et al. came to the following striking conclusion relying on the "Western" AG (*Table 2*): the Mal'ta (MA1_ANE) lineage received a gene flow from the Caucasus hunter–gatherer (CHG) lineage. However, in our *findGraphs* exploration this direction of gene flow (CHG → Mal'ta) was supported by two of the 29 topologies, and the opposite gene flow direction (from the Mal'ta and East European hunter–gatherer lineages to CHG) was supported by the remaining 27 plausible topologies (*Figure 4—source data 3*). The highest-ranking plausible topology (*Figure 4c*) has a fit that is not significantly different from that of the simplified published model with six admixture events (p-value = 0.392). We note that the gene flow direction contradicting the graph by Sikora et al. was supported by published *qpAdm* analyses (*Lazaridis et al., 2016*; *Narasimhan et al., 2019*), and *qpAdm* is not affected by the same model degeneracy issues that are the focus of this study. Considering the topological diversity among models that are temporally plausible, conform to robust findings about relationships between modern and archaic humans, and fit nominally better than the published model, we conclude that the direction of the Mal'ta-CHG gene flow cannot be resolved by AG analysis (*Table 2*).

Some important conclusions based on the "Eastern" graph also do not replicate across all plausible AGs (*Table 2*). In the case of the "Eastern" graphs with 6 admixture events, 4,446 topology search

iterations were performed, and 2,785 distinct topologies were found. Only 3 topologies fit significantly and 13 nominally better than the published one, and 9.8% of topologies fit not significantly worse than the published one (***Table 1***, ***Supplementary file 1***). Of the AGs belonging to these groups, we inspected 116 best-fitting ones and identified 97 AGs that are temporally plausible. The Sikora et al. "Eastern" AG had three distinctive features that were used to support some conclusions of the study (***Table 2***). Only feature 2 was universally supported by all the 97 plausible alternative models fitting significantly better, nominally better, or not significantly worse than the simplified published model, while feature 3 was supported by 83 of 97 plausible models, and feature 1 was supported by 28 of 97 plausible models (***Table 2***). We plotted 14 plausible graphs as examples of topologies supporting all three features, two features, or one feature of the published graph (***Figure 4—source data 4***). We note that all the "Eastern" graphs discussed here, both the published and alternative ones, have relatively poor absolute fits with WR above 4 or 5 SE. Increasing the number of gene flows to 10 allowed us to reach much better absolute fits (with WR as low as 2.42 SE), but that resulted in high topological diversity (on a par with some other case studies discussed above).

## Discussion
### A proposed protocol for using AG fitting in genetic studies

AGs represent a conceptually powerful framework for thinking about demographic history, but, as we demonstrate in this study (see also Appendix 2), the practice of manually constructing a small number of complex models without exploring AG space in an automated way can lead to overconfidence in the validity of these models. An ideal outcome of an AG model exploration exercise would be the identification of a model or a group of topologically very similar models which fit the data well and significantly better than all alternative models with the same number of admixture events; however, this is almost never achieved for graphs with more than eight populations and three admixture events in our experience (Appendix 2), and even this approach can lead to potentially unstable results as relaxing the assumption of parsimony (that fewer admixture events is more likely) can lead to qualitatively quite different equally well-fitting topologies as in our re-analysis of the Bergström et al. and Shinde et al. datasets. Most of the examples of AGs in eight recently published studies we revisited do not fit this ideal pattern, as we were able to identify many topologically different alternative models that could not easily be rejected based on temporal plausibility or other constraints (***Figure 3—source data 4***, ***Figure 3—source data 5***, ***Figure 3—source data 6***, ***Figure 3—source data 7***, ***Figure 3—source data 8***, ***Figure 4—source data 1***, ***Figure 4—source data 2***, ***Figure 4—source data 3***, ***Figure 4—source data 4***). In particular, for all studies except ***Shinde et al., 2019*** (under a strict parsimony assumption however), we identified AGs that were not significantly worse fitting than the published ones, and with topological features that were different in qualitatively important ways. There were also some more encouraging findings of the exercise we performed to re-evaluate published models. For example, at least one of the key inferences about population history relying on AG modeling were stable for all analyzed models for the ***Librado et al., 2021***., Hajdinjak et al., Shinde et al. (under the parsimony assumption), and Sikora et al. (simplified "Eastern" graph) studies (Appendix 2). The existence of some stable features in these graphs helps to point the way toward a protocol that we believe should be applied in all future studies that use AG fitting exercises to support claims about population history.

We propose the following tentative protocol to identify features of fitting AGs that are stable enough to be used to make inferences about population history.

1. For a given combination of populations, carry out an initial scan using *findGraphs* to identify reasonable parameter values for the number of allowed admixture events (graph complexity class). For example, run *findGraphs* allowing between zero and eight admixture events (100 algorithm iterations per graph complexity class), saving one or a few best-fitting AGs after each iteration. The smallest number of admixture events that yields models where the (negative) LL score or the worst *f*-statistic residual is lower than some threshold can then be explored more deeply by running more iterations of *findGraphs*.

2. Run *findGraphs* on the chosen complexity class, where some of the resulting graphs should be inspected manually to determine whether they could in principle be historically plausible models. Implausible models (e.g., models where a very ancient population appears to be admixed between two modern populations) can be filtered out by imposing topological constraints. If no

or only a few graphs remain, *findGraphs* can be run again under these constraints. This can be repeated until one or more graphs with an acceptable LL score or worst residual has been identified. At this stage, apply the bootstrap method to determine whether the best-fitting graph is significantly better than the next best-fitting graph. If it is not, identify a set of graphs which are not clearly worse than the best-fitting graph by performing bootstrap model comparison for many model pairs.

3. Researchers should compare the resulting graphs to each other with the goal of identifying common features. Although *ADMIXTOOLS 2* includes automated tools for cataloguing common topological features (Appendix 1, Sections 1.B.5 and 2.G), we found a manual approach to be valuable as the fitted parameters (especially admixture proportions) are as important for this task as graph topology.

4. Once a set of fitting graphs and stable topological features shared between them is identified, researchers should carry out a *findGraphs* exploration of the space of graphs with one additional admixture event. If inferences are stable even when fitting graphs with one more level of complexity than the graphs with the minimal number of admixture events needed to fit the data, this increases confidence in the inferences. Furthermore, addition of a new population may introduce crucial information to an existing set of populations, which can change the space of fitting topologies in a profound way, as in our re-analysis of the data from *Bergström et al., 2020* (*Figure 3a*, *Figure 3—source data 1*). Thus, it is advisable that the topology optimization procedure is repeated on several alternative population sets, in addition to considering models that allow an additional admixture event beyond the minimum required for parsimony, to explore if inferences about topology change qualitatively.

5. AGs fitted with *f*-statistics do not distinguish between time and population size as the two factors affecting genetic drift. Moreover, many different complex genetic histories for a set of populations can result in the exact same expected *f*-statistics. This provides an opportunity to further constrain a model fitting procedure. Methods that take advantage of information from the site frequency spectra (*momi2*, *fastsimcoal*, *Kamm et al., 2020*, *Excoffier et al., 2013*) or derived site patterns, a special case of site frequency spectra (Legofit, *Rogers, 2019*), can supply alternative information not captured by *f*-statistics (further information can come from methods that fit haplotype divergence patterns such as *MSMC*, *Schiffels and Durbin, 2014* and *SMC++*, *Terhorst et al., 2017*, or inferences based on fitted gene trees such as *RELATE*, *Speidel et al., 2019*, and *ARGweaver*, *Hubisz et al., 2020*; *Hubisz and Siepel, 2020*). These tools are too computationally intensive to explore a large number of models, but the advantages of the different approaches can be combined by first identifying a set of candidate models using *findGraphs*, and then testing these candidate models with other methods. This approach is also expected to help address overfitting since different data types almost always include different variable site sets.

We believe that researchers should only begin to make strong claims about population history with AGs once a protocol such as we propose is applied.

We see the guidelines above as analogous in spirit to the protocols that were introduced in medical genetics at a time of the reproducibility crisis in the field of candidate gene association studies. Many studies looking for risk factors for common, complex diseases resulted in publications with marginally significant p-values without correcting for multiple hypothesis testing that was implicitly performed due to many candidate genes being tested and only those with significant findings being published. Unsurprisingly, most of these claims failed to replicate in follow-up studies in independent sets of samples (*Ioannidis, 2005*; *Border et al., 2019*; *Collins et al., 2012*; *Duncan et al., 2019*). The human medical genetic community addressed this challenge by coming together to support a rigorous set of commonly accepted standards for declaring genome-wide statistical significance, such as the requirement that p-values be corrected for the effective number of independent common variants in the genome and requiring correction for the known confounders of population structure and undocumented relatedness among individuals (*Hirschhorn and Daly, 2005*).

## Conclusions

Sampling AG space is a useful method for modeling population histories, but finding robust and accurate models can be challenging. As we demonstrated by revisiting a handful of published AGs and re-analyzing the datasets used to fit them, *f*-statistics are usually insufficient for identifying uniquely fitting AG models, making it necessary to incorporate other sources of evidence. This provides a challenge to previous approaches for automated model building. We investigated several published

AG models and, in nearly all cases, found many alternative models, some of which are historically and geographically plausible but contradict conclusions that were derived from the published models. To conduct these analyses, we developed a method for automated AG topology optimization that can incorporate external sources of information as topological constraints. This method is developed in the *ADMIXTOOLS 2* framework, which aside from AG modeling, implements many other methods for population history inference based on $f$-statistics.

It is important to recognize that the key concern we have highlighted in this study—the fact that there can often be thousands of different topologies that are equally good fits to the allele frequency correlation patterns relating a set of populations—does not invalidate the use of allele frequency correlation testing in many other contexts in which it has been applied to make inferences about population history. For example, negative $f_3$-statistics ('admixture' $f_3$-statistics) continue to provide unambiguous evidence for a history of mixture in tested populations, and $f_4$-and $D$-symmetry statistics remain powerful ways to evaluate whether a tested pair of populations is consistent with descending from a common ancestral population since separation from the ancestors of two groups used for comparison. The *qpWave* methodology remains a fully valid generalization of $f_4$-statistics, making it possible to test whether a set of populations is consistent with descending from a specified number of ancestral populations (which separated at earlier times from a comparison set of populations). In addition, *Haak et al., 2015* and *Harney et al., 2021* the *qpAdm* extension of *qpWave*—which allows for estimating proportions of mixtures for the tested population under the assumption that we have data from the source populations for the mixture—remains a valid approach, unaffected by the concerns identified here. Instead of relying on a specific model of deep population relationships, *qpAdm* relies on an empirically measured covariance matrix of $f_4$-statistics for the analyzed populations, which is highly constraining with respect to estimation of mixture proportions but can be consistent with a wide range of deep history models. All these methods are implemented in *ADMIXTOOLS 2*.

Finally, approaches that use AGs to adjust for the covariance structure relating a set of populations without insisting that the particular AG model that is proposed is true with can be useful, for example for the purpose of analyzing shared genetic drift patterns of a group of populations that derive from similar mixtures. One example was a study that attempted to test for different source populations for Neolithic migrations into the Balkans after controlling for different proportions of hunter–gatherer admixture (*Mathieson et al., 2018*). Another example was a study that attempted to study shared ancestry between different East African forager populations after controlling for different proportions of deeply divergent source populations (*Lipson et al., 2022*). However, with respect to the inferences about deep history produced by AGs themselves, our results highlight the importance of caution in proposing specific models of population history that relate a set of groups.

## Acknowledgements

We thank Anders Bergström, Esther Brielle, Mateja Hajdinjak, Iosif Lazaridis, Pablo Librado, Mark Lipson, Vagheesh Narasimhan, Ludovic Orlando, Nick Patterson, Mary Prendergast, Jakob Sedig, Kendra Sirak, Pontus Skoglund, and Chuanchao Wang, for suggestions for how to improve specific analyses, and for conversations and critical comments. We thank Matthew Mah, Shop Mallick, Adam Micco, Nadin Rohland, Ron Pinhasi for help in generating additional data from an ancient DNA library from individual I8726 for which 1.24 million SNP capture data was generated and published in *Narasimhan et al., 2019* and for which we report 2.6-fold shotgun data here (*Supplementary file 2*). P.F., P.C., O.F., and U.I. were supported by the Czech Ministry of Education, Youth and Sports (program ERC CZ, project no. LL2103). P.F., P.C., and O.F. were supported by the Czech Science Foundation (project no. 21-27624S). P.F. and P.C. were also supported by the Czech Ministry of Education, Youth and Sports: the "Large Infrastructures for Research, Experimental Development and Innovations" program (project "IT4Innovations National Supercomputing Center" no. LM2015070) and the Inter-Excellence program (project no. LTAUSA18153). R.M. and D.R. were supported by grants from the National Institutes of Health (GM100233 and HG012287), the John Templeton Foundation (grant 61220), and the Allen Discovery Center program, a Paul G Allen Frontiers Group advised program of the Paul G Allen Family Foundation. D.R. and P.F. were supported by private gifts from Jean-Francois Clin. D.R. is an Investigator of the Howard Hughes Medical Institute.

## Additional information

### Funding

| Funder | Grant reference number | Author |
|---|---|---|
| Czech Ministry of Education, Youth and Sports | LL2103 | Pavel Flegontov Piya Changmai Olga Flegontova Ulaş Işıldak |
| Czech Ministry of Education, Youth and Sports | LM2015070 | Pavel Flegontov Piya Changmai |
| Czech Ministry of Education, Youth and Sports | LTAUSA18153 | Pavel Flegontov Piya Changmai |
| National Institutes of Health | GM100233 | Robert Maier David Reich |
| National Institutes of Health | HG012287 | Robert Maier David Reich |
| John Templeton Foundation | grant 61220 | Robert Maier David Reich |
| The Czech Science Foundation | 21-27624S | Pavel Flegontov Piya Changmai Olga Flegontova |

The funders had no role in study design, data collection, and interpretation, or the decision to submit the work for publication.

### Author contributions

Robert Maier, Conceptualization, Resources, Data curation, Software, Formal analysis, Validation, Investigation, Visualization, Methodology, Writing – original draft, Writing – review and editing; Pavel Flegontov, Conceptualization, Resources, Data curation, Formal analysis, Supervision, Funding acquisition, Validation, Investigation, Visualization, Writing – original draft, Project administration, Writing – review and editing, Methodology; Olga Flegontova, Formal analysis; Ulaş Işıldak, Formal analysis, Visualization; Piya Changmai, Formal analysis, Funding acquisition; David Reich, Conceptualization, Resources, Supervision, Funding acquisition, Project administration, Writing – review and editing

### Author ORCIDs

Robert Maier ⬩ http://orcid.org/0000-0002-3044-090X
Pavel Flegontov ⬩ http://orcid.org/0000-0001-9759-4981
Ulaş Işıldak ⬩ http://orcid.org/0000-0001-6497-6254
David Reich ⬩ http://orcid.org/0000-0002-7037-5292

### Decision letter and Author response

Decision letter https://doi.org/10.7554/eLife.85492.sa1
Author response https://doi.org/10.7554/eLife.85492.sa2

## Additional files

### Supplementary files

• Supplementary file 1. Published graphs in the context of automatically found graphs. We compared 22 different graphs from 8 publications to alternative graphs inferred on the same or very similar data; these *findGraphs* runs are highlighted in blue in the 'Iterations' column. In total, 51 *findGraphs* runs are summarized here since in some cases models more complex or less complex than the published one were explored and/or different population compositions were tested (see the 'Topology search constraints and population modifications' column and the footnotes for details). The columns with names in blue show various information on the published graphs or their modified versions and some properties of the published population sets. The columns with names in

magenta show settings used for calculating $f$-statistics and for exploring the AG space, and the number of SNPs used that depends on them. The columns with names in black summarize the outcomes of *findGraphs* runs, that is, the properties of alternative model sets found. **Publication:** Last name of the first author and year of the relevant publication. **Figure in the original publication:** Figure number in the original paper where the AG is presented. **Groups (populations):** The number of populations in each graph. **Singleton pseudo-haploid populations:** The number of populations in the graph composed of a single pseudo-haploid individual. Calculation of negative 'admixture' $f_3$-statistics is impossible for such populations since their heterozygosity cannot be estimated (see the text for details). **No. of negative $f_3$-stats (allsnps: YES):** The number of negative $f_3$-statistics among all possible $f_3$-statistics for a given set of populations when all available sites are used for each statistic. If no negative $f_3$-statistics exist for a set of populations, AG fits are not affected by the '*minac2=2*' setting intended for accurate calculation of $f$-statistics for non-singleton pseudo-haploid groups. **Admixture events:** The number of admixture events in each graph. **Publ. model: log-likelihood (LL):** Log-likelihood score of the published graph fitted to the SNP set shown in the 'SNPs used' column. **Publ. model: LL, median of bootstrap distr.:** Median of the log-likelihood scores of 100 or 500 fits of the published graph using bootstrap resampled SNPs. **Publ. model: worst residual (WR), SE:** The worst $f$-statistic residual of the published graph fitted to the SNP set shown in the 'SNPs used' column, measured in standard errors (SE). **SNPs used:** The number of SNPs (with no missing data at the group level) used for fitting the AG. For all case studies, we tested the original data (SNPs, population composition, and the published graph topology) and obtained model fits very similar to the published ones. However, for the purpose of efficient topology search we adjusted settings for $f_3$-statistic calculation, population composition, or graph complexity as shown here and discussed in the text. **Settings for calculating $f_2$-statistics:** Arguments of the *extract_f2* function used for calculating all possible $f_2$-statistics for a set of groups, which were then used by *findGraphs* for calculating $f_3$-statistics needed for fitting AG models. See Appendix 1, Section 2.A for descriptions of each argument. **Topology search constraints and population modifications:** Constraints applied when generating random starting graphs and/or when searching the topology space. Modifications of the original population composition are also described in this column, where applicable. **Iterations:** The number of *findGraphs* iterations (runs), each started from a random graph of a certain complexity. For each case study, *findGraphs* setups that were considered optimal are highlighted in blue in this column. **Iterations confirming published graph:** The number of iterations (runs) in which the resulting graph was topologically identical to the published graph. In the cases, when the published model was irrelevant since more complex graphs were explored, 'N/A' appears in this and subsequent columns. If less complex models were explored, the published model was still relevant since its version without selected admixture edges was tested. **Distinct alternative topologies found:** The number of distinct newly found topologies. If graph complexity was equal to (or less than) that of the published graph, the published topology (or its simplified version) is not counted here. If graph complexity exceeded that of the published graph, all newly found topologies are counted. If the published topology was recovered by *findGraphs*, the numbers in this column are shown in bold. **Significantly better fitting topologies:** The number of distinct topologies that fit significantly better than the published graph according to the bootstrap model comparison test (two-tailed empirical p-value <0.05). If the number of distinct topologies was very large, a representative sample of models (1/20 to 1/3 of models evenly distributed along the log-likelihood spectrum) was compared to the published one instead. These cases are marked as 'a fraction of models tested' in this column. If model complexity was higher than that of the published model, model comparison was irrelevant and was not performed. **Non-significantly better fitting topologies:** The number of distinct topologies that fit non-significantly (nominally) better than the published graph according to the bootstrap model comparison test (two-tailed empirical p-value ≥0.05). **Non-significantly worse fitting topologies:** The number of distinct topologies that fit non-significantly (nominally) worse than the published graph according to the bootstrap model comparison test (two-tailed empirical p-value ≥0.05). **Significantly worse fitting topologies:** The number of distinct topologies that fit significantly worse than the published graph according to the bootstrap model comparison test (two-tailed empirical p-value <0.05). **Significantly better fitting topologies, %:** The percentage of distinct topologies that fit significantly better than the published graph according to the bootstrap model comparison test (two-tailed empirical p-value <0.05). If the number of distinct topologies was very large, a representative sample of models (1/20 to 1/3 of models evenly distributed along the log-likelihood spectrum) was compared to the published one instead, and the percentages in this and following columns were calculated on this sample. **Non-significantly better fitting topologies, %:** The percentage of distinct topologies that fit non-significantly (nominally) better than the published

graph according to the bootstrap model comparison test (two-tailed empirical p-value ≥0.05). **Non-significantly worse fitting topologies, %:** The percentage of distinct topologies that fit non-significantly (nominally) worse than the published graph according to the bootstrap model comparison test (two-tailed empirical p-value ≥0.05). **Significantly worse fitting topologies, %:** The percentage of distinct topologies that fit significantly worse than the published graph according to the bootstrap model comparison test (two-tailed empirical p-value <0.05). **p-value best alternative vs. publ.:** An empirical two-tailed p-value of a test comparing log-likelihood distributions across bootstrap replicates for two topologies, the highest-ranking newly found topology and the published topology. In some cases, the highest-ranking newly found topology (according to LL) has a fit that is not significantly better than that of the published model, but other newly found models fit significantly better despite having higher LL. P-values below 0.05 are highlighted in green. **Used in *Table 1*:** Here the *findGraphs* runs featured in *Table 1* are marked.

- Supplementary file 2. Statistics for shotgun sequencing of individual I8726.
- Supplementary file 3. Group labels, archaeological and geographic meta-data, and sequencing statistics for the individuals used for admixture graph fitting in *Shinde et al., 2019*. See the 'Comment' column for a list of updates to the dataset composition we performed prior to admixture graph fitting in our study.
- Supplementary file 4. Group labels, archaeological and geographic meta-data, and sequencing statistics for the individuals used for *qpAdm* modelling that revisits *qpAdm* results by *Narasimhan et al., 2019*.
- Supplementary file 5. A summary of selected *qpAdm* results. This is a re-analysis of the data from *Narasimhan et al., 2019*, with a modified group composition that is described in Appendix 2.
- Supplementary file 6. *qpAdm* results for separate individuals from selected target groups (Indus Periphery and others). *qpAdm* p-values are shown for each individual for models of varying complexity, from one- to four-way models.
- Supplementary file 7. Comparison of the original group composition used for admixture graph fitting in *Librado et al., 2021* and the modified groups used in our study.
- MDAR checklist

### Data availability

The new software presented in this manuscript (the ADMIXTOOLS 2 R package) is freely available at https://github.com/uqrmaie1/admixtools (copy archived at *Maier et al., 2022*), along with a detailed manual at https://uqrmaie1.github.io/admixtools/. The ancient human genome newly reported in this manuscript (*Supplementary file 2*) is freely available at the European Nucleotide Archive in the form of an alignment of reads to the hg19 human reference genome (project accession number PRJEB58199). Published software packages re-used in this manuscript are available at: https://bitbucket.org/nygcresearch/treemix/src/master/ (TreeMix, *Pickrell and Pritchard, 2012*) and at https://github.com/DReichLab/AdmixTools (*David Reich Lab, 2023*, ADMIXTOOLS, *Patterson et al., 2012*). Published archaeogenetic datasets re-analyzed in this manuscript were kindly shared by the corresponding authors of the following publications upon our requests: *Bergström et al., 2020*; *Lazaridis et al., 2014*; *Librado et al., 2021*; *Lipson et al., 2020b*; *Shinde et al., 2019*; *Sikora et al., 2019*; *Wang et al., 2021*; *Hajdinjak et al., 2021*. Various statistics for these re-used datasets are summarized in *Supplementary file 1*.

The following dataset was generated:

| Author(s) | Year | Dataset title | Dataset URL | Database and Identifier |
|---|---|---|---|---|
| Maier R, Flegontov P, Flegontova O, Işıldak U, Changmai P, Reich D | 2022 | On the limits of fitting complex models of population history to f-statistics | https://www.ebi.ac.uk/ena/browser/view/PRJEB58199 | European Nucleotide Archive, PRJEB58199 |

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

## Appendix 1

### 1. *ADMIXTOOLS 2*. A new software package and set of algorithmic ideas for fitting allele frequency correlation statistics to genetic data relating populations

We present a new implementation of the popular *ADMIXTOOLS* software (called 'Classic *ADMIXTOOLS*') (*Patterson et al., 2012*; *Haak et al., 2015*). Our implementation (*ADMIXTOOLS 2*) enhances performance by greatly reducing runtime and memory requirements across a wide range of different methods, relative to Classic *ADMIXTOOLS* (*Appendix 1—figure 1a*). Some of these improvements have now been implemented in version 7.0.2 of *ADMIXTOOLS* (https://github.com/DReichLab/AdmixTools/). The present study focuses not on the performance differences between Classic *ADMIXTOOLS* and *ADMIXTOOLS 2*, but on the description of new ideas implemented in one or both of these tools.

### 1.A. Computation and use of $f$-statistics

A key idea that facilitates the performance increases shared by *ADMIXTOOLS 2* and *ADMIXTOOLS* v. 7.0.2 is that any $f$-statistic (which form the basis of almost all *ADMIXTOOLS* programs as well as other toolkits for studying population history such as *popstats*) can be computed from a small number of $f_2$-statistics. For most $f$-statistic-based analyses (for example *qpWave*, *qpAdm*, and *qpGraph*; *Appendix 1—figure 1c*), the time required to compute $f_3$- and $f_4$-statistics algebraically from $f_2$-statistics is trivial compared to the time required to load genotype data and compute $f_2$-statistics. These $f_2$-statistics can be stored and re-used to compute $f_3$- and $f_4$-statistics, thus reducing the size of the input data, runtime, and memory requirements by orders of magnitude (*Appendix 1—figure 1a, d*).

Using precomputed $f_2$-statistics is not always the best solution. In datasets with large amounts of missing data, computing $f_3$- and $f_4$-statistics from precomputed $f_2$-statistics may introduce bias. In this case, it is necessary to compute $f_3$- and $f_4$-statistics directly, using different SNPs for each $f$-statistic (all available SNPs in each population triplet or quadruplet). However, even without the use of precomputed $f_2$-statistics, *ADMIXTOOLS 2* often achieves large performance gains (*Appendix 1—figure 1a*).

The program *qpfstats* in Classic *ADMIXTOOLS* implements an idea which strikes a balance between these two extremes. It increases the accuracy of estimation of $f$-statistics by using a regression approach to jointly estimate the values of all $f_2$-, $f_3$-, and $f_4$-statistics relating a set of populations. Specifically, *qpfstats* searches for values of these statistics that are not only consistent with information from the SNPs that have data in the groups used to compute each particular $f$-statistic, but also satisfy the algebraic relationships expected with other $f$-statistics (thus incorporating information from data at many additional SNPs). See further details on this algorithm at https://github.com/DReichLab/AdmixTools/blob/master/qpfs.pdf, (*Patterson et al., 2012*). This feature is available in *ADMIXTOOLS 2* through the *qpfstats* option in the *extract_f2* function.

Another improvement introduced in *ADMIXTOOLS 2* relates to accurate evaluation of the match between observed and expected $f_3$-statistics when fitting AGs where at least one population is represented by a single individual with genotypes derived by randomly selecting one sequencing read at each variable position ('pseudo-haploid' data). $f$-Statistic computations need to be modified when analyzing pseudo-haploid data, because heterozygosity cannot be computed using comparisons of sequences within the same individual; however, computation of heterozygosity is essential to calculate 'admixture' $f_3$-statistics $f_3$(target; A, B), where negative values provide proof of the mixed nature of the target population. When a target population is represented by multiple individuals, unbiased estimation of admixture $f_3$-statistics can be carried out even for pseudo-haploid data by analyzing positions covered by sequences from at least two individuals and only computing variation rates across individuals. This approach is implemented in Classic *ADMIXTOOLS* with the 'inbreed: YES' option. However, no admixture $f_3$-statistic can be computed with this algorithm if the target population is represented by a single individual (as no variation across individuals within a population can be detected in this case). Classic *ADMIXTOOLS* deals with this case by failing to run if any population in an analysis is represented by a single individual and the 'inbreed: YES' option is turned on. Because the datasets from all the AGs revisited here included at least one population represented by a single individual (*Supplementary file 1*), the 'inbreed: YES' option could not be

used in the original studies (the program failed with this option, by design). Thus, AG fitting in those studies relied on the incorrect algorithm for calculating $f_3$-statistics (except for *Librado et al., 2021*, which used *TreeMix* instead of *qpGraph*) and, as a result, some $f_3$-statistics that are negative and could provide important constraints for AG fitting were evaluated as positive. These concerns are relevant for the *Shinde et al., 2019*, *Lipson et al., 2020b*, and *Wang et al., 2021* studies we revisit below (see *Supplementary file 1* for a list of datasets where negative $f_3$-statistics were encountered). To be able to detect negative $f_3$-statistics and thus take advantage of their power for constraining the space of possibly fitting historical models, in *ADMIXTOOLS 2* we introduced a similar algorithm which makes it possible to compute negative $f_3$-statistics on pseudo-haploid data, at a cost of removing sites with only one chromosome genotyped in any population that is represented by at least two individuals (so that it is possible in theory to compute heterozygosity in these populations). Admixture $f_3$-statistics continue to be incorrectly computed using *ADMIXTOOLS 2* for targets that are singleton populations represented by pseudo-haploid data, as there is no avoiding this particular problem.

**a**

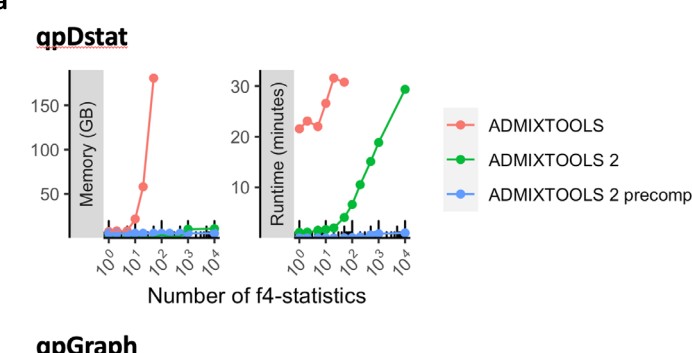

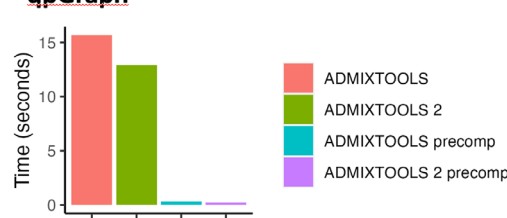

**b**

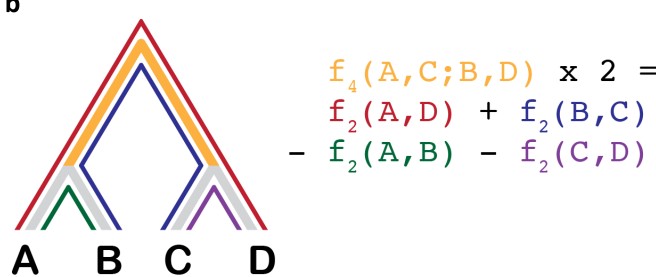

$$f_4(A,C;B,D) \times 2 =$$
$$f_2(A,D) + f_2(B,C)$$
$$- f_2(A,B) - f_2(C,D)$$

**A  B  C  D**

**c**

| Program | f-statistic | Primary use |
|---|---|---|
| qp3pop | $f_3$ | Test for admixture |
| qpDstat | $f_4$ | Test for admixture |
| qpF4ratio | $f_4$ | Estimating admixture proportions |
| qpWave | $f_4$ | Finding how many gene flows connect two sets of populations |
| qpAdm | $f_4$ | Estimating admixture proportions |
| qpGraph | $f_3$ | Fitting AG models of history |

**d**

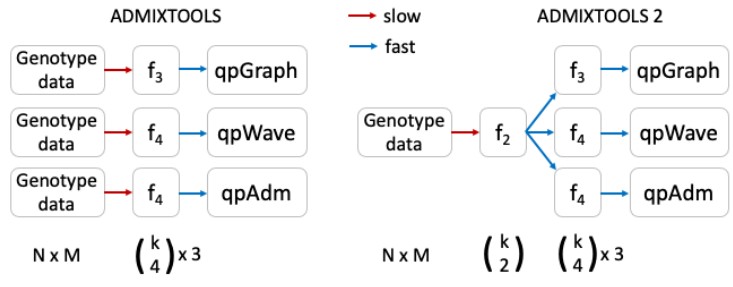

**Appendix 1—figure 1.** Performance comparison of f-statistic computation and AG fitting in Classic ADMIXTOOLS and ADMIXTOOLS 2 and an overview of the major ADMIXTOOLS programs. (**a**) Performance comparison of *f*-statistic computation and AG fitting. Top: Memory usage and runtime for computing *f*-statistics using (1) the *qpDstat* program in *ADMIXTOOLS* v7.0.2 released in 06/2021, (2) the *f4* function in *ADMIXTOOLS 2* without precomputing $f_2$-statistics, and (3) the *f4* function in *ADMIXTOOLS 2* with precomputed $f_2$-statistics. (1) and (2) *Appendix 1—figure 1 continued on next page*

*Appendix 1—figure 1 continued*

give identical results, whereas (3) only gives identical results in the absence of missing data, which limits its usefulness beyond a moderate number of populations. Bottom: Runtime comparison of *qpGraph* with and without precomputed *f*-statistics. (**b**) Illustration of $f_2$- and $f_4$-statistics. $f_2$ measures the amount of drift separating any two populations, while $f_4$ measures the amount of drift shared between two population pairs. Every $f_4$-statistic is a linear combination of four $f_2$-statistics. (**c**) Overview of the major *ADMIXTOOLS* programs, their primary use cases, and their associated *f*-statistics. (**d**) Schematic representation of the computations behind the *ADMIXTOOLS* programs *qpGraph*, *qpWave*, and *qpAdm*. *ADMIXTOOLS 2* separates the computation of $f_2$-statistics from the later steps in the pipeline. Shown below are the number of data points for $N$ individuals, $M$ SNPs, and $k$ populations. The exact number of all possible non-redundant $f_2$-, $f_3$-, and $f_4$-statistics for $k$ populations are $\binom{k}{2}$, $\frac{1}{2}\binom{k}{3}$, and $\frac{1}{3}\binom{k}{4}$. A small number of $f_2$-statistics can be used to obtain a much larger number of $f_3$- and $f_4$-statistics and require much less storage space than the raw genotype data.

## 1.B. AG fitting, model comparison, and interpretation

There are several challenges that arise when modeling the ancestral relationships among populations with AGs, and *ADMIXTOOLS 2* implements solutions to several key problems that were not adequately addressed with previous approaches: (1) Automated AG inference; (2) Estimating confidence intervals for AG parameters; (3) Comparing fits of different AGs; (4) Determining identifiability of AG parameters; and (5) Drawing conclusions from a large number of fitting graphs.

Here, we describe these challenges, how we address them, and how our approaches compare to other approaches, while the section 'Technical presentation of *ADMIXTOOLS 2* in the context of methods based on *f*-statistics' gives detailed descriptions.

### 1.B.1. Automated AG inference

Constructing AGs manually runs the risk of overlooking models that challenge conventional hypotheses. On the other hand, current methods for inferring AGs automatically (*Leppälä et al., 2017*; *Molloy et al., 2021*; *Pickrell and Pritchard, 2012*; *Yan et al., 2021*) do not allow external information to be integrated into the analysis, and often result in models that may fit the genetic data but can be rejected on other grounds. In addition, *TreeMix* (*Pickrell and Pritchard, 2012*), as well as *OrientAGraph* (*Molloy et al., 2021*), an improved version of *TreeMix*, can miss AG topologies that exist on parts of the non-convex likelihood surface that are bypassed by these algorithms for exploring AGs (e.g., topology M7 in Figure 4 of *Molloy et al., 2021*). *MixMapper* (*Lipson et al., 2013*) and *miqoGraph Yan et al., 2021* have a different limitation: exploring topologies with more than one admixture event in the history of any group is not possible. Due to these limitations, many published findings are based on manual proposal of topologies and evaluation of fit, and the great majority of studies using this manual approach (see, e.g., *Reich et al., 2011*; *Reich et al., 2012*; *Lazaridis et al., 2014*; *Fu et al., 2016*; *Skoglund et al., 2016*; *Yang et al., 2017*; *McColl et al., 2018*; *Moreno-Mayar et al., 2018*; *Tambets et al., 2018*; *van de Loosdrecht et al., 2018*; *Flegontov et al., 2019*; *Sikora et al., 2019*; *Wang et al., 2019*; *Lipson et al., 2020b*; *Shinde et al., 2019*; *Yang et al., 2020*; *Hajdinjak et al., 2021*; *Wang et al., 2021*; *Bergström et al., 2022*) rely on the software *qpGraph*. We introduce an approach for finding well-fitting AGs automatically that can integrate external information, and that recovers graph topologies more accurately than *TreeMix* (*Appendix 1—figure 2*). External information can be integrated by specifying a set of constraints that AGs must satisfy. This not only ensures that resulting models are temporally plausible, but also cleanly separates prior assumptions from the independent constraints provided by genetic data. Our strategy implemented in the function '*findGraphs*', differs from *TreeMix/OrientAGraph* in several deep ways, most notably in that it optimizes graphs directly, rather than optimizing trees first and adding admixture events later. This makes it less prone to getting stuck in local optima: our simulation results show that *findGraphs* is more accurate for random graphs (*Appendix 1—figure 2*), and that it can recover specific topologies that pose problems for *TreeMix* and *OrientAGraph*.

### 1.B.2. Estimating confidence intervals for AG parameters

Since our new implementation of *qpGraph* can evaluate models much more rapidly, it becomes feasible to evaluate the same model multiple times on different SNP sets. This allows us to derive bootstrap confidence intervals (*Boos, 2003*) for all parameters estimated by *qpGraph*, including drift lengths, admixture weights, LL scores, and $f_4$-statistic residuals. It should also be noted that the estimated confidence intervals do not take into account uncertainty about the graph topology.

## 1.B.3. Comparing the fits of different AGs

Using the bootstrap method for evaluating a graph multiple times on different SNP sets not only allows us to obtain confidence intervals for single graphs, but also allows us to test whether the fit of one graph is significantly better than the fit of another graph, by obtaining confidence intervals for the LL score difference or difference in the largest $f_4$-statistic residuals (worst residuals, WR) of two graphs. When we apply this approach to a range of datasets, we find that models with modest LL differences are often not distinguishable after accounting for the variability across SNPs, even if one might expect them to be distinguishable based on the magnitude of the likelihood difference (*Appendix 1—figure 3a, b*). Thus, previous methods relying on AIC or BIC (such as *Shinde et al., 2019*; *Flegontov et al., 2019*) that used specified likelihood difference thresholds to reject some models over others, were over-aggressive. These methods are problematic since they generally assume that the models compared have the same effective number of degrees of freedom, but the number of independent parameters estimated in an AG is not simply determined by the number of groups, drift edges, or admixture events, as it also depends on the graph topology in a complex way. A second challenge in comparing different AG models arises when comparing models of different complexity (i.e., with a different number of admixture events). Established methods such as AIC and BIC can also account for different model complexity, but if the number of independent parameters in a model is known.

We implement a method to compare AG models of different complexity by using a new scoring function, which uses different blocks of SNPs for deriving fitted and estimated $f$-statistics. This ensures that our model comparison test does not favor more complex models by allowing them to overfit the data. This cross-validation approach can also be used to rank alternative models of the same complexity and deal with overfitting. We note that the calculation of cross-validated LL scores is not turned on in *findGraphs* by default, and to make our results more comparable to those of the published studies we revisited, we relied on standard LL scores in this study.

To test if our method is well calibrated, we simulated 100,000 SNPs under the same graph in 1,000 replicates. We then created two new topologies by removing one out of two symmetric edges from the first graph (*Appendix 1—figure 3c*). These new incorrect models are symmetrically related to the first graph and can be used to test whether the true difference in LL scores of these two graphs is zero. The uniform distribution of p-values confirms that our method is well calibrated (*Appendix 1—figure 3d*). A caveat is that only one symmetric topology was explored in this way.

## 1.B.4. Determining identifiability of AG parameters

Fitting AGs results in an estimate of the overall model fit, as well as in estimates of branch lengths and admixture weights. However, even with infinite data some of these parameters cannot be estimated, as they are not identifiable from the system of equations that corresponds to the AG. Issues like this have been well described for simple topological features of a graph. For example, the lengths of the two branches connected to the root node cannot be estimated independently. Furthermore, in a graph with $n$ populations and $a$ admixture events, at least one parameter will not be identifiable unless the inequality $\binom{n}{2} >= 2n + 2a - 3$ is satisfied (*Lipson, 2020a*). However, even in graphs that meet this criterion, some parameters are not identifiable, and until the development of *ADMIXTOOLS 2*, there was no method for testing whether any given parameter in an AG is identifiable. We introduce a method for testing which parameters in an AG are identifiable, and which are not, based on the Jacobi matrix of the graph's system of $f_2$ equations. Like our method for deriving confidence intervals for AG parameters, this can improve the interpretability of AG analyses.

Our methods for automated topology inference, for bootstrapping LL scores or worst residuals for comparing model fits, for cross-validation of AGs of different complexities, for estimating confidence intervals and determining identifiability of AG parameters can greatly improve the interpretability of AG analyses. We implement all these methods in *ADMIXTOOLS 2* to assist the user in accurately testing a series of admixture models for ancient populations.

## 1.B.5. Drawing conclusions from a large number of fitted models

As discussed in the Results section, when we apply our methods for finding optimal graphs and comparing AGs to a number of previously published models, we find that there often exists a much larger number of fitting models than has previously been appreciated. In these cases, we are unable to prioritize a single model, or even a small number of models, based on the evidence we

have. However, we are still able to reject the vast majority of all tested models. This suggests that insight can be gained by identifying common features among the well-fitting models. We therefore introduce methods for summarizing collections of well-fitting AGs to determine which features they share. In practice, we find that these methods can aid manual inspection of *findGraphs* results, but the high diversity of well-fitting topologies we see in most case studies and the importance of fitted parameters (especially admixture proportions) for historical interpretation of topologies makes it difficult to reliably automatize the process of interpreting fitted AG models.

## 2. Technical presentation of *ADMIXTOOLS 2* in the context of methods based on *f*-statistics

Much of the content that follows recapitulates theory presented in previous work, notably *Reich et al., 2009*, *Green et al., 2010*, and *Patterson et al., 2012*, but we summarize it here for coherence.

### 2.A. *f*-Statistics

All *ADMIXTOOLS* programs are based on the statistics $f_2$, $f_3$, and $f_4$, for population pairs, triplets, and quadruples, respectively.

$f_2$ quantifies the genetic drift separating two populations $A$ and $B$ . For a single SNP, it is given by $f_2\left(A, B\right) = \frac{1}{M} \sum_j \left(a_j - b_j\right)^2$ , where $a_j$ and $b_j$ are the allele frequencies for SNP $j$ in populations $A$ and $B$. When allele frequencies are estimated using a small number of samples, this estimator of $f_2$ will be biased upwards. An unbiased estimator of $f_2$ is given by

$f_2 = \frac{1}{M} \sum_j \left(a_j - b_j\right)^2 - \frac{a_j\left(1-a_j\right)}{n_{A,j}-1} - \frac{b_j\left(1-b_j\right)}{n_{B,j}-1}$, where $n_{A,j}$ and $n_{B,j}$ are the observed allele counts in populations $A$ and $B$.

$f_3\left(A; B, C\right) = \frac{1}{M} \sum_j \left(a_j - b_j\right)\left(a_j - c_j\right)$ is the covariance of the allele frequency differences between populations $A$ and $B$, and the allele frequency differences between populations $A$ and $C$ (assuming that alleles are coded randomly, so that $a - b$ and $a - c$ are both 0 in expectation). Significantly negative values of $f_3(A; B, C)$ suggest that $A$ is a mixture of sources related to $B$ and $C$ (although the converse does not hold: $A$ might be admixed between $B$ and $C$ even if $f_3$ is positive).

$f_4\left(A, B; C, D\right) = \frac{1}{M} \sum_j \left(a_j - b_j\right)\left(c_j - d_j\right)$ is the covariance of the allele frequency differences between $A$ and $B$, and the allele frequency differences between $C$ and $D$. Significantly positive values of $f_4(A, B; C, D)$ (or equivalently significantly negative values of $f_4(A, B; D, C)$) reveal that $A$ and $B$ do not form a clade with respect to $C$ and $D$, and that some of the drift separating $A$ from $C$ is shared with the drift separating $B$ from $D$.

$f_3$ and $f_4$ can be written as linear combinations of $f_2$-statistics:

$$f_3\left(A; B, C\right) = \frac{1}{2}\left(f_2\left(A, B\right) + f_2\left(A, C\right) - f_2\left(B, C\right)\right) \tag{1}$$

$$f_4\left(A, B; C, D\right) = \frac{1}{2}\left(f_2\left(A, D\right) + f_2\left(B, C\right) - f_2\left(A, C\right) - f_2\left(B, D\right)\right) \tag{2}$$

This implies that all $f_3$- and $f_4$-statistics can be computed from $f_2$-statistics as long as they are defined on the same SNPs.

For revisiting published studies, we used the '*extract_f2*' function with the '*maxmiss*' argument set at 0, which corresponds to the '*useallsnps: NO*' setting in classic *ADMIXTOOLS*. It means that no missing data are allowed (at the level of populations) in the specified set of populations for which pairwise $f_2$-statistics are calculated. For the values of the '*blgsize*', '*adjust_pseudohaploid*', and '*minac2*' arguments we use in our analyses, see *Supplementary file 1*. The '*blgsize*' argument sets the SNP block size in Morgans, and we used either the default value of 0.05 (5 cM), or 4,000,000 bp when a genetic map was not available. Genotypes of pseudo-haploid samples are usually coded as 0 or 2 (i.e., they are, strictly speaking, pseudo-diploid), even though only one allele is observed. The '*adjust_pseudohaploid*' argument ensures that the observed allele count increases only by 1 for each pseudo-haploid sample. If '*TRUE*' (default), samples that do not have any genotypes coded as 1 among the first 1,000 SNPs are automatically identified as pseudo-haploid. This leads to slightly more accurate estimates of $f$-statistics. Setting this parameter to '*FALSE*' treats all samples as diploid.

Another important argument ('*minac2=2*') of the '*extract_f2*' function removes sites with only one chromosome genotyped in any non-singleton population and is needed for unbiased estimation of negative $f_3$-statistics in non-singleton pseudo-haploid populations. In the absence of negative

$f_3$-statistics or pseudo-haploid populations, this argument has no influence on AG LL scores. This algorithm for calculation of $f$-statistics triggered by the '*minac2=2*' argument is described below.

For $f_3(A; B, C)$, we compute the uncorrected numerator for each SNP, $(a - b) \times (a - c)$. We then subtract a bias correction factor at each SNP, $p(1 - p)/(ac - 1)$, which we only need for population $a$ (because the other factors cancel out); $p$ is the allele frequency, and $ac$ is the observed allele count. In pseudo-haploid samples, $(ac - 1)$ would be zero and produce an error in any sites with only one observed allele. With the '*inbreed: NO*' setting in Classic *ADMIXTOOLS*, the smallest non-zero value for $ac$ is 2, so the division by 0 problem is avoided, but the correction factor is slightly smaller than it should be. *ADMIXTOOLS 2* adds only an allele count of 1 for each site in a pseudo-haploid sample (with the default option '*adjust_pseudohaploid = TRUE*'), so there can be cases where $ac = 1$. To imitate what the setting '*inbreed: NO*' in Classic *ADMIXTOOLS* is doing, $ac$ is set to 2 at those sites (or the denominator is set to 1). There is still a small difference between Classic *ADMIXTOOLS* and *ADMIXTOOLS 2* at other sites because each observed site adds two alleles in *ADMIXTOOLS* with the default setting '*inbreed: NO*', but only 1 allele in *ADMIXTOOLS 2* with the default setting '*adjust_pseudohaploid = TRUE*', but for AG fitting that does not matter. One solution to avoid biased correction factors is to only consider sites with $ac$ of at least two, which is what the '*inbreed: YES*' setting in Classic *ADMIXTOOLS* does. The problem with this is that we cannot use populations with a single pseudo-haploid sample, which is often useful, and would only give misleading results if that population is admixed. The new option '*minac2=2*' in *ADMIXTOOLS 2* is different from the '*inbreed: YES*' setting in Classic *ADMIXTOOLS* since it makes an exception for populations consisting of a single pseudo-haploid sample in that it sets $ac$ to 2 at each site (denominator is set to 1) when computing the correction factor for those populations.

## 2.B. Fitting AGs

An AG is a directed acyclic graph specifying the topology of the ancestral relationships among a set of populations. Each node in this graph represents a (present-day or ancient) population. Terminal nodes (also called leaf nodes) represent observed populations, while internal nodes represent unobserved ancestral populations. Modeling all observed populations as leaf nodes confers some robustness to drift specific to single populations and to genotyping errors. The edges connecting the populations are weighted and correspond either to the magnitude of genetic drift that has occurred along that branch (drift edges), or to the admixture proportions (admixture edges, where two edges point to the same node).

The goal of *qpGraph* is to test how well a given graph topology fits the observed $f$-statistics. This is achieved by varying the edge weights until the maximum likelihood fit is obtained. The following section describes the graph fitting in more detail.

First, for $k$ populations, all $\frac{k(k+1)}{2}$ $f_3$-statistics of the form $f_3(O; X_1, X_2)$ are computed, where $O$ is one of the $k$ populations (typically an outgroup), and $X_1$ and $X_2$ are all pairs formed from the other populations (including pairs where $X_1 = X_2$). These $f_3$-statistics can then be used to fit the graph and to compute the likelihood. The likelihood score of a graph is the dot product of the differences between the expected and observed $f_3$-statistics, weighted by the inverse covariance matrix of $f_3$-statistics:

$$L\left(g\right) = -\frac{1}{2}\left(f_{3,\,obs} - f_{3,\,fit}\right)' Q^{-1}\left(f_{3,\,obs} - f_{3,\,fit}\right) \tag{3}$$

Here, $f_{3,\,obs}$ are the observed $f_3$-statistics and $f_{3,\,fit}$ are the fitted $f_3$-statistics. Both are vectors of length $q = \frac{k(k+1)}{2}$ for $k$ populations excluding the outgroup. $Q$ is the $q \times q$ covariance matrix of $f_3$-statistics, where the diagonal entries are the $f_3$-statistic variances, and the off-diagonal entries are the covariances for all pairs of $f_3$-statistics. Just like the variances (the squared standard errors), the covariances are estimated from the jackknife leave-one-block-out $f_3$-statistics.

Finding the edge weights which maximize the likelihood score involves two nested optimization steps. The inner optimization finds the drift weights which maximize the likelihood score while fixing the admixture weights. The outer optimization finds the admixture weights which maximize the likelihood score, while optimizing the drift weights for each set of admixture weights. The inner optimization uses a quadratic programming solver to find the optimal drift weights, while the outer optimization uses a general purpose optimization algorithm to find the optimal admixture weights.

While the gradient function in the outer optimization adjusts the admixture weights, the objective function iterates over the following steps:

1. Optimization of drift weights conditional on admixture weights
2. Estimation of fitted $f_3$-statistics
3. Calculation of the graph likelihood using observed and fitted $f_3$-statistics

These steps are repeated until convergence is reached and the likelihood score can no longer be improved by adjusting the admixture weights.

Step 1 optimizes the drift edge weights, while holding the admixture weights constant. All drift edge weights are required to be non-negative, which makes this a constrained quadratic programming problem (hence *qp*Graph). Additional upper and lower bounds can be specified for individual graph edges.

Step 2 turns the edge weights into fitted $f_3$-statistics. To see how edge weights in an AG translate to $f_3$-statistics, it helps to first consider how they translate into $f_2$-statistics for a pair of populations. Without any admixture events, there is exactly one path $p$ connecting any two populations. The fitted $f_2$-statistic ($f_{2,\,fit}$) is the sum of edge weights $w_e$ along this path $p$ connecting two populations. The fitted $f_2$-statistic is the sum of edge weights $w_e$ along this path:

$$f_{2,\,fit} \;=\; \sum_{e\,\in\,p} w_e$$

In the presence of admixture events, two populations may be connected via multiple paths. Each admixture node that lies between the two populations increases the number of possible paths. The fitted $f_2$-statistic for the two populations now becomes the weighted sum of all these paths, where the weight of each path is given by the product of all estimated admixture proportions $w_a$ along this path ($\prod_{a\,\in\,p} w_a$):

$$f_{2,\,fit} \;=\; \sum_{p\,\in\,P} \prod_{a\,\in\,p} w_a \sum_{e\,\in\,p} w_e \qquad\qquad (4)$$

The fitted $f_2$-statistics are then used to obtain fitted $f_3$-statistics using *Equation 1*.

Step 3 uses the fitted and observed $f_3$-statistics to estimate the likelihood score using *Equation 3*.

Prior to these three steps, initial admixture weights are drawn randomly. To ensure that the end results do not depend on the random initialization, the whole optimization process is repeated multiple times with different random initial values. The original *ADMIXTOOLS* implementation retains only the results from the initial values resulting in the lowest absolute likelihood score. The new *ADMIXTOOLS 2* implementation provides an option to retrieve the results for all random initializations. This can be useful, as large fluctuations between different random initializations can be an indicator of an overparameterized or otherwise poorly fitting model.

## 2.C. Automated AG inference

To find graph topologies that could conceivably have given rise to the observed $f$-statistics, we start with a randomly generated graph with a fixed number of admixture events, apply a number of modifications to this graph, and evaluate each of the resulting graphs. We then pick the best-fitting graph and repeat this procedure until graph modifications no longer lead to improved scores. We use a number of random graph modifications, as well as targeted modifications which are informed by parameters obtained during the fitting of the current graph.

For the targeted modifications, we change the optimization of a single graph from a constrained optimization problem, in which drift edges are constrained to be positive and admixture weights are constrained to be between zero and one, to an unconstrained optimization problem in which both types of parameters can take any real values. Rearranging the nodes adjacent to edges which were estimated to be negative results in an improved fit at a much higher rate than random graph adjustments.

The random modifications include (1) pruning and randomly re-grafting leaf nodes, (2) pruning and randomly re-grafting a set of connected nodes in the graph, (3) swapping the orientation of admixture edges, (4) shifting admixture edges, (5) re-rooting the graph, and (6) combinations of two or more of any of these modifications.

The number of admixture events is not affected by the graph modifications described so far. A significant score improvement can often be achieved by adding a single admixture edge to several random positions in a graph. This is unsurprising since it increases the degrees of freedom of the original graph. However, picking the best fitting graph with one admixture edge added, and testing all graphs that result from removing a single admixture edge from that graph, often results in a graph with the same number of admixture events and a better fit than the original graph. We employ this strategy whenever the regular graph modifications described above do not lead to any further improvements.

We keep track of the search tree of all previously evaluated graphs and their scores in order to not evaluate any graph more than once, and so that backtracking in the search space is possible in cases where no more local improvements can be identified. Nevertheless, multiple iterations with different random starting graphs are usually necessary to find graphs with good fits. The number of iterations needed to approach a global optimum depends on the size of the search space, but the optimal number of iterations is hard to estimate in practice.

For revisiting published studies, we used the following settings of the *findGraphs* algorithm:

- *mutfuns = namedList(spr_leaves, spr_all, swap_leaves, move_admixedge_once, flipadmix_random, place_root_random, mutate_n)*, a list of functions used to modify graphs.
- *numgraphs = 10*, number of alternative graphs produced by randomly applying the mutation functions at the start of each generation.
- *stop_gen = 10,000*, total number of generations after which to stop.
- *stop_gen2=30*, number of generations without LL score improvement after which to stop.
- *plusminus_generations = 10*. If the best score does not improve after *plusminus_generations* generations, another approach to improving the score is attempted: A number of graphs with an additional admixture event is generated and evaluated. The resulting graph with the best score is picked, and new graphs are created by removing any one admixture event (bringing the number back to what it was originally). The graph with the lowest score is then selected. This approach often makes it possible to break out of local optima.
- *opt_worst_residual = FALSE*. Optimize for lowest worst residual instead of best score. '*FALSE*' by default, because the LL score is generally a better indicator of the quality of the model fit, and because optimizing for the lowest worst residual is much slower since $f_4$-statistics need to be computed for each graph.
- *reject_f4z=0*. If this is a number greater than zero, all $f_4$-statistics with |$Z$-score|>*reject_f4z* will be used to constrain the search space of AGs: Any graphs in which $Z$-scores greater than *reject_f4z* are expected to be zero will not be evaluated.
- *diag = 1e−04*. This argument is passed to the *qpgraph* function and determines the regularization term added to the diagonal elements of the covariance matrix of fitted branch lengths (after scaling by the matrix trace). Default is 0.0001.
- *numstart = 10*. This argument is passed to the *qpgraph* function and determines the number of random initializations of starting weights (defaults to 10). Increasing this number will make the optimization slower but reduce the risk of not finding the optimal weights.
- *lsqmode = FALSE*. This argument is passed to the *qpgraph* function. If set to '*FALSE*', the inverse $f_3$-statistic covariance matrix is not discarded by the algorithm.

The arguments '*admix_constraints*' (constraints on the number of admixture events in the history of a given population), '*event_constraints*' (constraints on the branching order of specified lineages), and '*outpop*' (the population assigned as an outgroup) were set according to *Supplementary file 1*. Each *findGraphs* run was initiated by a random graph with a specified number of admixture events. Usually, the same topology constraints were applied at the stage of random graph generation and the topology search stage, for exceptions see *Supplementary file 1*.

## 2.D. Evaluating automated AG inference through simulations

We evaluated the performance of *findGraphs* by simulating genetic data under a large number of different AG models, applying *findGraphs* to each simulated dataset in three independent iterations, and comparing the resulting best graph across three iterations to the simulated graph. We applied *TreeMix* to the same simulated data for comparison. We simulated between 8 and 16 populations per graph, and between 0 and 10 admixture events. For each parameter combination, we simulated 20 different AGs generated by the *random_admixturegraph* function. We counted both the fraction

of random simulated graphs where the best inferred graph was identical to the simulated graph, as well as the fraction of random simulated graphs where the best inferred graph was either identical to the simulated graph or had a better score than the simulated graph. For models with a large number of admixture events the number of possible models is so large that it becomes increasingly likely that there will be some alternative models which fit the data better than the model under which the data were simulated.

We used *msprime v.0.7.4* and the *msprime_sim* wrapper function in *ADMIXTOOLS 2* to simulate data for 100,000 unlinked SNPs and 100 diploid samples per population for each AG. The simulation parameters we chose were aimed at facilitating fast simulations of large numbers of informative SNPs rather than at being as realistic as possible. We therefore expect that the simulation results allow us to make comparisons across groups, but not that they are informative about the rate at which 'true' models can be recovered in empirical data. We simulated under a constant mutation rate of 0.001 per site per generation, a constant haploid effective population size of 1,000, with neighboring nodes in the graph separated by 1,000 or more generations, and all admixture events occurring in discrete pulses of 50%/50% proportions.

To allow for a fair comparison between *findGraphs* and *TreeMix*, we made sure that small differences in the way AGs are modeled in *findGraphs* and in *TreeMix* were accounted for before testing graphs for identical topology. For example, *TreeMix* AGs can have lineages terminating at an admixture node, whereas in *findGraphs* lineages always end at a 'leaf' node with a single ancestor.

More realistic simulations were performed with *msprime v.1.1.1* which allows accurate simulation of recombination and of multi-chromosome diploid genomes relying on the Wright–Fisher model (*Nelson et al., 2020*, *Baumdicker et al., 2022*). We simulated three chromosomes (each 100 Mbp long) in a diploid genome by specifying a flat recombination rate ($2 \times 10^{-8}$ per bp per generation) along the chromosome and a much higher rate at the chromosome boundaries ($\log_e 2$ or ~0.693 per bp per generation, see https://tskit.dev/msprime/docs/stable/ancestry.html#multiple-chromosomes). A flat mutation rate, $1.25 \times 10^{-8}$ per bp per generation (*Scally and Durbin, 2012*), and the binary mutation model were used. To maintain the correct correlation between chromosomes, the discrete time Wright–Fischer model was used for 25 generations into the past, and deeper in the past the standard coalescent simulation algorithm was used (as recommended by *Nelson et al., 2020*).

We simulated AGs of four complexity classes: eight or nine groups sampled at leaves, four or five pulse-like admixture events. All group sizes were identical: 10 diploid individuals with no missing data. Demographic events were separated by date intervals ranging randomly between 1,500 and 8,000 generations, with an upper bound on the tree time depth at 40,000 generations. Effective population sizes were constant along each edge, and were picked randomly from the range of 2,000–40,000 diploid individuals. Admixture proportions for all admixture events varied randomly between 10% and 40%. For subsequent analyses we selected only simulations where pairwise $F_{ST}$ for groups were in the range characteristic for anatomically modern and archaic humans (there was at least one $F_{ST}$ value below 0.15). In this way, 20 random topologies were simulated per graph complexity class, each including also a distant outgroup that facilitates exploration of the topology space. The outgroup diverged at 40,000 generations ago and had a constant diploid population size of 100,000 individuals. Since there was no missing data and all individuals were diploid, we first calculated all possible $f_2$-statistics for 4 Mbp-sized genome blocks (with the '*maxmiss = 0*', '*adjust_pseudohaploid = FALSE*', and '*minac2=FALSE*' settings) and then used them for calculating $f_4$-statistics as linear combinations of $f_2$-statistics or for fitting AGs (with the '*numstart = 100*' and '*diag = 0.0001*' settings).

## 2.E. Comparing the fits of different AGs

We are interested in determining whether one AG fits the data significantly better than another AG, or whether an observed score difference $\Delta = S_1 - S_2$ can be attributed to variability across independent SNPs. We first consider two AGs with the same number of admixture events, where we can ignore the problem of comparing two models with different complexity. As in other bootstrap standard error calculations, we divide the genome into $n$ blocks indexed by $i$, and we draw $b$ sets of $n$ blocks with replacement, indexed by $j$. We fit both graphs $b$ times—once for each bootstrap set of SNP blocks. This results in a set of $b$ score differences $\Delta_j$. The bootstrap confidence interval for the difference in scores is given by the quantiles of the distribution of $\Delta_j$. We also compute an empirical bootstrap p-value, testing the null hypothesis that two different graphs fit the data

equally well. It is computed as $p = max\left(\frac{1}{b}, 2\delta\right)$ (**Boos, 2003**), where $\delta$ is either the fraction of $\Delta_j > 0$, or the fraction of $\Delta_j < 0$, whichever is smaller. The reason for applying bootstrap resampling, as opposed to jackknife resampling in this case, is that the distribution of score differences tends to have a high kurtosis, which can make jackknife estimates inaccurate. Simulating data under the null hypothesis is not straightforward in this case, because it involves finding two non-identical graphs which in expectation fit the data equally well. We decided to simulate under one graph and compare two graphs which are symmetrically related to the simulated graph (**Appendix 1—figure 3**). This confirmed that the p-value follows a uniform distribution under the null hypothesis.

Next, we consider comparisons of two graphs of different complexity. The problem here is that more complex graphs have more degrees of freedom which allow them to overfit the data better, without necessarily being any closer to the truth. To solve this problem, we introduce an out-of-sample likelihood score. The regular likelihood score is given by:

$$L(g) = -\frac{1}{2}\left(f_{3,\,obs} - f_{3,\,fit}\right)' Q^{-1}\left(f_{3,\,obs} - f_{3,\,fit}\right),$$ with $f_{3,\,obs}$ and $f_{3,\,fit}$ defined on the same set of SNPs. The out-of-sample likelihood score is defined in the same way, except that $f_{3,\,obs}$ and $f_{3,\,fit}$ are defined on mutually exclusive sets of SNP blocks, thereby preventing any overfitting. The covariance matrix $Q$ is defined on the same set of SNP blocks as $f_{3,\,fit}$. As described earlier, we use block-bootstrap to fit both graphs multiple times on different SNP blocks. In each bootstrap iteration, we use all SNP blocks which are not used in fitting the graph for estimating $f_{3,\,obs}$.

## 2.F. AG identifiability

An edge in an AG is unidentifiable, if small changes to the weight of this edge (admixture proportions in the case of an admixture edge, drift length in the case of a drift edge) do not necessarily lead to changes in expected $f$-statistics. This is the case if the small change in weight can be offset by small changes in other graph edges, leading to a situation where observed $f$-statistics can be explained by more than one weight estimate for that edge. To find unidentifiable edges, we derive the Jacobi matrix of the graph's system of $f_2$ equations (**Equation 4** applied to each population pair). In principle, whether a parameter is identifiable can depend on the values of all other parameters. However, in practice this is rarely the case, and so we draw values for all parameters from a uniform distribution, which gives us a Jacobi matrix with numeric values. We then determine the rank of the Jacobi matrix, along with the rank of all matrices that result from dropping a single column (a parameter corresponding to a graph edge). For identifiable edges, the rank of the full matrix will be greater than the rank of the reduced matrix, and for unidentifiable edges, the ranks will be the same.

## 2.G. Drawing conclusions from a large number of fitting models

We developed several methods that aim to summarize a collection of graphs which all fit the data similarly well. By highlighting features which are observed repeatedly across graphs, it becomes possible to extract interpretable conclusions from an otherwise hard to interpret collection of possible models. These graph summaries identify features in each graph that can be compared to different graphs describing the same populations. We summarize each graph in several ways:

1. Admixture status of each population
   For each population, we count the total number of admixture events that are encountered along all paths from the chosen leaf to the root.
2. Order of population split events
   For each pair of population pairs, we determine if the most recent split of the first pair has occurred before or after the most recent split of the second pair, or whether the graph does not specify the order in which those splits occurred.
3. Proxy populations
   For each admixed population in a graph, we attempt to identify proxy sources: populations closest to the admixing populations. In contrast to the other approaches to summarizing graphs which are based only on the topology of each graph, this can also rely on information about the estimated graph parameters.
4. Cladality
   For each group of four populations, we test whether the graph implies that any $f_4$-statistic describing the relationship between the four populations is expected to be zero.
5. Node descendants

Each internal node in an AG is an ancestor to a specific set of leaf populations. An AG can be characterized by the sets of leaf populations formed by the internal nodes. Multiple AGs may be compared by counting the number of overlapping sets. This also makes it possible to quantify for each internal node in a single graph, how often a matching internal node can be found across a collection of alternative graphs, which is conceptually similar to bootstrap support values in phylogenetic trees.

While these methods provide some help in comparing features across many graphs, they are not able to reliably answer the question whether the fitting graphs are relatively similar or dissimilar from each other, and whether they are similar to any particular graph. This is in part due to the fact that small topological changes involving populations of interest may be more relevant than similar topological changes involving only populations that are not the focus of the study.

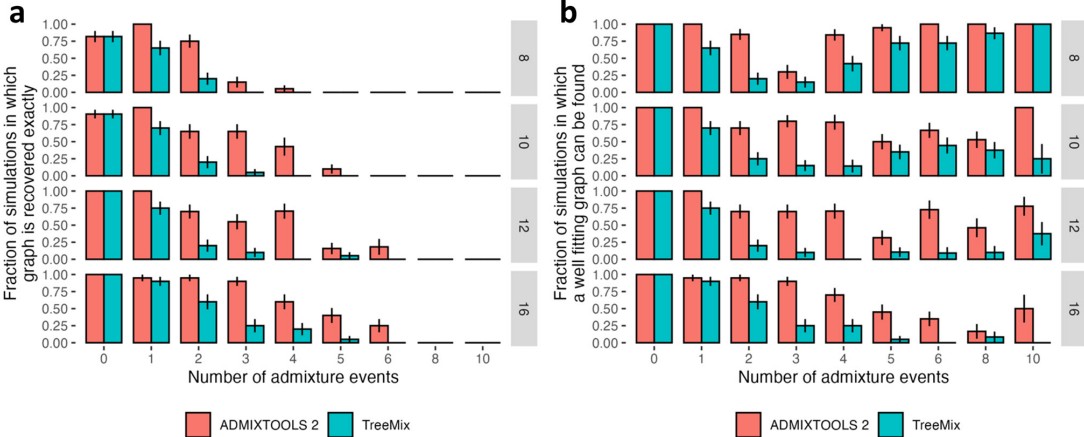

**Appendix 1—figure 2.** Comparison of accuracy of automated search for optimal topology in the *findGraphs* function of *ADMIXTOOLS 2* and in *TreeMix* using simulated graphs with 8, 10, 12, and 16 populations, and 0–10 admixture events. Error bars show standard errors calculated as $SE^2 = p (1 – p) / n$, where $p$ is the fraction on the $y$-axis and $n$ is the number of simulations in each group (typically 20). In the case of *ADMIXTOOLS 2*, we applied *findGraphs* three times on each simulated dataset and picked a result with the best fit score. More details are provided in Methods. (**a**) Fraction of simulations where the simulated graph is recovered exactly. (**b**) Fraction of simulations where the simulated graph is either recovered exactly, or the score is at least as good as the score of the simulated graph, when both graphs are evaluated by *ADMIXTOOLS 2*. More admixture edges greatly increase the search space and make it more difficult to recover the simulated graph, but they do make it easier to find alternative graphs with good fits.

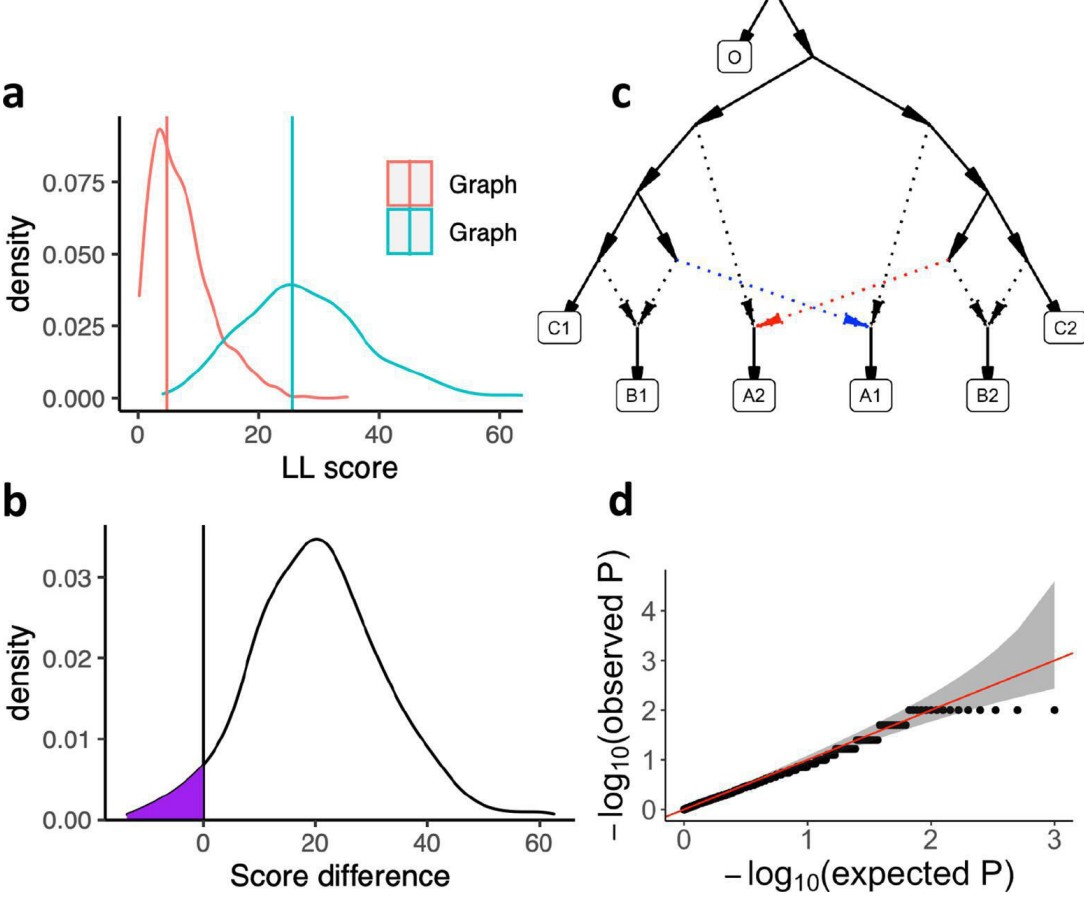

**Appendix 1—figure 3.** Calibrating the bootstrap model comparison approach. (**a**) Bootstrap sampling distributions of the log-likelihood scores for two AGs (shown in *Appendix 1—figure 3—Figure supplement 1*) for the same populations fitted using real data. Vertical lines show the log-likelihood scores computed on all SNP blocks. (**b**) Distribution of differences of the bootstrap log-likelihood scores for both graphs (same data as in **a**). The purple area shows the proportion of resamplings in which the first graph has a higher score than the second graph. The two-sided p-value for the hypothesis of no difference is equivalent to twice that area (or one over the number of bootstrap iterations if all values fall on one side of zero). In this case it is 0.078. (**c**) The AG which was used to evaluate our method for testing the significance of the difference of two graph fits on simulated data. We simulated under the full graph and fitted two graphs that result from deleting either the red admixture edge or the blue admixture edge. These two graphs have the same expected fit score but can have different scores in any one simulation iteration. (**d**) QQ plot of p-values testing for a score difference between the two graphs (on simulated data) under the hypothesis of no difference, confirming that the method is well calibrated.

The online version of this article includes the following figure supplement(s) for appendix 1—figure 3:

**Appendix 1—Figure 3 supplement 1.** The admixture graphs compared in (*Appendix 1—figure 3*).

## Appendix 2

### 1. *Bergström et al., 2020*

The AG for dogs in Figure 1e of *Bergström et al., 2020* was inferred by exhaustively evaluating all graphs with two admixture events and outgroup 'Andean fox' for the six populations that remain in the graph after excluding an Early Neolithic dog from Germany. The only six-population graph with a worst residual (WR) below 3 standard errors (SE) was then chosen as a scaffold onto which the Early Neolithic dog genome from Germany was mapped, allowing for one more admixture event, and a seven-population graph with the lowest LL score was shown as the final model in the paper. Alternative six-population scaffolds were not explored in the original publication, although two six-population graphs with fits very similar to the best one were found. LL was not used as a ranking metric for alternative models in the original study; instead, the number of $f_4$-statistics having residuals above 3 SE was considered. Since no $f_3$-statistics were negative when all sites available for each population triplet were used (the 'useallsnps: YES' option), we did not use the upgraded algorithm for calculating $f_3$-statistics on pseudo-haploid data.

Our *findGraphs* results confirm the published six-population graph in that no graph with lower LL score is identified, but 3 of 14 unique alternative graphs found fit not significantly worse than the published graph (the published graph was also recovered by *findGraphs*) (***Table 1***, ***Supplementary file 1***). When we used *findGraphs* to infer seven-population graphs with three admixture events (again fixing Andean fox as the outgroup), we identified five graphs with LL score nominally better than that of the published graph and one with a score that is slightly lower than that of the published graph but actually significantly better according to our model comparison methodology (this model is very similar to the published graph, ***Figure 3—source data 1***). In the newly found seven-population graph with the best LL score (***Figure 3—source data 1***), the Siberian (Baikal), American, and East Mediterranean dogs are admixed, and the West European, East European (Karelia), and dogs of Southeast Asian origin (New Guinea singing dog) are unadmixed, while the opposite pattern is found in the published graph (***Table 2***). The best-fitting graph does not fit the data significantly better than the published graph (two-tailed empirical p-value = 0.332), but it bears a closer resemblance to the human population history (see the third-best graph found by *findGraphs* on human data from *Bergström et al., 2020* in ***Figure 3—source data 2***) than the published seven-population graph (***Figure 3a***, ***Figure 3—source data 1***).

In this new seven-population model (***Figure 3a***), both American and Siberian dog lineages represent a mixture between groups related to the Asian and East European dog lineages, and robust genetic results suggest that in the time horizon investigated in the original publication (after ca. 10,900 years ago) nearly all Siberian (***Jeong et al., 2019***; ***Sikora et al., 2019***) and all American (***Raghavan et al., 2014***; ***Raghavan et al., 2015***; ***Moreno-Mayar et al., 2018***) human populations were admixed between groups most closely related to Europeans and Asians. According to this model, East Mediterranean dogs are modeled as a mixture of a basal branch (splitting deeper than the divergence of the Asian and European dogs) and West European dogs, again in agreement with current models of genetic history of West Asian human populations who are modeled as a mixture of 'basal Eurasians' and WHG (***Lazaridis et al., 2016***; ***Lipson et al., 2017***). Although greater congruence with human history increases the plausibility of *findGraph*'s newly identified model relative to the published model, to make unbiased comparisons between the history of the two species, model selection should be done strictly independently for each species, and so the genetic data alone does not favor one model more than another. Our results provide a specific alternative hypothesis that differs in qualitatively important ways from the published model and can be tested against new genetic data as it becomes available as well as other lines of genetic analysis of existing data.

To explain why the original paper on the population history of dogs missed the model that *findGraphs* identified that is plausibly a closer match to the true history, we observe that the *Bergström et al., 2020* AG search was exhaustive under the parsimony constraint (no more than twp admixture events for six populations), and thus missed the potentially true topology including three admixture events for these six populations. This case study also illustrates that even in a relatively low complexity context (seven groups and three admixture events) applying manual approaches for finding optimal models is risky. When any new group such as an Early Neolithic dog from Germany is added to the model, it may introduce crucial information into the system, and re-exploring the

whole graph space in an automated way is advisable. In contrast, mapping a newly added group on a simple skeleton graph (even when that skeleton is a uniquely best-fitting model) may yield a topology that is at odds with the true history. As the original Bergström et al. paper noted (Figure 3C of that study), no congruent six-population graph models were found for humans and dogs under the parsimony assumption: the three most congruent graphs for dogs resulted in poorly fitting models for the corresponding human populations (WR above 10 SE), and the three most congruent graphs for humans resulted in poorly fitting models for the corresponding dog populations (WR between 5 and 10 SE). We added a WHG group to the set of human groups from the original publication and using *findGraphs* on the original set of 77 K transversion SNPs we found that the third best-fitting model for humans (*Figure 3—source data 2*) (which is not significantly different in fit from the first one) is nearly identical topologically to the newly found dog graph.

Even though *findGraphs* identified an AG topology that fits the data as well as the seven-population graph in Bergström et al. and is qualitatively quite different with respect to which populations were admixed, the new topology continues to support another of the key inferences of that study: that many of the early divergences among domesticated dog lineages occurred prior to the date of the Karelian dog (~10,900 ya). Thus, both graphs concur in providing strong evidence that the radiation of domesticated dog lineages occurred by the early Holocene, prior to the domestication of other animals. We further emphasize that the *Bergström et al., 2020* graph is the best-case scenario (along with *Lazaridis et al., 2014* discussed below) for published AGs. Most published graphs are far less stable even than this.

## 2. *Lazaridis et al., 2014*

The graph in Figure 3 (from *Lazaridis et al., 2014*) was inferred in the following manner. First, a phylogenetic tree without admixture was constructed which was the best fit for all $f_4$-statistics among the populations 'Mbuti', 'WHG Loschbour' (*Lazaridis et al., 2014*), 'LBK Stuttgart' (*Lazaridis et al., 2014*), 'Onge', and 'Karitiana', with 'Mbuti' fixed as an outgroup. Next, all possible AGs were considered that result from adding a single admixture edge to this tree. After it was found that each of them had a WR > 3 SE, several graphs with two admixture events were considered, and some of them had WR < 3 SE. The 'MA1' genome (*Raghavan et al., 2014*) was added to these graphs in several different ways, and only one of these configurations was found to have WR < 3 SE. This was then used as a skeleton graph onto which a European population (represented by different present-day groups) was added. No fitting graph was found in which present-day Europeans could be modeled as a two-way mixture (adding one admixture event to the graph). After inspecting the non-fitting *f*-statistics of one of these graphs, it was found that modeling modern Europeans as a three-way mixture (adding two admixture events to the graph) is consistent with all *f*-statistics. Six $f_3$-statistics were negative when all sites available for each population triplet were used (the 'useallsnps: YES' option, *Supplementary file 1*), but the upgraded algorithm for calculating $f_3$-statistics on pseudo-haploid data had no effect since the only pseudo-haploid group in the dataset (MA1) was a singleton population, and the algorithm removes sites with only one chromosome genotyped in any <u>non</u>-singleton population. Thus, below we show results generated using the standard algorithm for calculating *f*-statistics.

First, we considered the published skeleton graph onto which a European population was later added (*Lazaridis et al., 2014*). As in the *Bergström et al., 2020* example, the best six-population graph with two admixture events found by *findGraphs* is identical to the published six-population graph, which has an LL score of 3.0 (*Supplementary file 1*). The second-best graph found has an LL score of 31.8. When computing the bootstrap p-value for the difference between these two graphs, we find that in 1.6% of all SNP resamplings the second-best graph has a better score than the published graph, resulting in a two-tailed empirical p-value of 0.032 for a difference in fits between these two graphs. All 14 alternative graphs found by our algorithm fit significantly worse than the published graph (*Supplementary file 1*).

When we add the European population (French) and consider seven-population graphs with four admixture events, we find 40 out of 306 distinct graphs with a score better than that of the published graph (10 of those graphs are shown in *Figure 3—source data 3*). The best-fitting newly found model and two other models fit the data significantly better than the published model (*Supplementary file 1*), but their topology is qualitatively very similar to that of the published graph (*Figure 3—source data 3*). In the best-fitting newly found model, French and Karitiana share some drift to the exclusion of MA1, while in the published model the source of MA1-related ancestry in French is closer to MA1 than to Karitiana.

It is important to point out that not all of the 40 alternative graphs that fit nominally or significantly better than the published one are consistent with the conclusion that modern European populations are admixed between three different ancestral populations (*Figure 3—source data 3*). For example, the fifth alternative graph in *Figure 3—source data 3* that is fitting nominally better than the published model (p-value = 0.464) includes no basal Eurasian ancestry in EEF (LBK Stuttgart), and instead models Onge as having ~50% West Eurasian-related ancestry and MA1 as having ~25% Asian ancestry. According to that graph, the present-day European population was formed by admixture of an MA1-related lineage and a European Neolithic-related lineage, with no West European hunter–gatherer (WHG) contribution. Of course, other lines of evidence make it clear that LBK Stuttgart is a mixture of Anatolian farmer-related ancestry and WHG Loschbour-related ancestry (*Lazaridis et al., 2016*; *Lipson et al., 2017*), thus providing external information in favor of the *Lazaridis et al., 2014* model, and the use of such ancillary information in concert with graph exploration is important in order to obtain more confident inferences about population history taking advantage of AGs.

The second alternative graph in *Figure 3—source data 3* that fits just negligibly worse than the highest-ranking model has another distinctive feature: LBK Stuttgart is modeled as a mixture of a WHG-related and a basal Eurasian lineage, but modern Europeans receive a gene flow not from the LBK-related lineage, but from its basal Eurasian source. Although temporally plausible, this model is much less plausible from the archaeological point of view than the published model, and thus in this case too we can reject it as unlikely based on non-genetic evidence. We note, however, that a large group of newly found models (247 graphs) fits not significantly worse than the published one (*Supplementary file 1*), and those are topologically diverse. Thus, strictly speaking, the AG method on the given dataset cannot be used to prove that the published model is the only one fitting the data.

## 3. *Shinde et al., 2019*

The skeleton AG in the original study (*Shinde et al., 2019*) was constructed manually on the basis of an SNP set derived from the 1240K enrichment panel, and subsequently all possible branching orders (105) within the five-population Iranian farmer-related clade were tested. The published model (Figure 3 in that study) included 9 groups and 3 admixture events, but one group (Belt Cave Mesolithic) had a very high missing data rate, and as a result model fitting relied not just on the merged dataset which included 19,000 polymorphic sites without missing data across groups, but also on a dataset with approximately 470,000 sites that excluded the Belt Cave individual. The topological inferences were consistent for both analyses (Table S3 of that study). Following the approach of the published paper, we repeated *findGraphs* analysis both with and without the Belt Cave individual. Thus, we initially explored the following topology classes: 9 groups with 3 admixture events on ca. 19,000 polymorphic sites and 8 groups with 3 admixture events on ca. 470,000 sites (*Supplementary file 1*). The sample composition of the groups and the SNP dataset matched that in the original study. We summarize results across 4,000 independent iterations of the *findGraphs* algorithm for each topology class.

For the nine-population graph we found 89 models with LL nominally better than that of the published model (*Supplementary file 1*). For the eight-population graph, we found 61 nominally and 4 significantly better fitting models (*Supplementary file 1*), and their topological diversity was high (*Figure 3—source data 4*). We note that the following groups were admixed by default in the graph models compared in the original study: Hajji Firuz Neolithic (labeled 'Chalcolithic' in that study but the dates are Neolithic) and Tepe Hissar Chalcolithic were considered as mixtures of an Anatolian farmer-related lineage and an Iranian farmer-related lineage; Indus Periphery was considered as a mixture of an Andamanese-related lineage representing ancient South Indians (ASI) and an Iranian farmer-related lineage. However, calculation of negative 'admixture' $f_3$-statistics for these target groups is impossible using the original dataset and the original model fitting algorithm for several reasons. First, the Indus Periphery group was represented by a single pseudo-haploid individual (I8726) from the 'Indus Valley cline' for whom the best-quality data were available. But direct calculation of 'admixture' $f_3$-statistics for such a group as a target is impossible since its heterozygosity cannot be estimated. Second, as discussed above, in Classic *ADMIXTOOLS* it is impossible to apply a correction intended for accurate calculation of $f_3$-statistics on pseudo-haploid data (the 'inbreed: YES' option) if there is at least one population composed of one individual only (a singleton population). Third, the original Hajji Firuz Neolithic group composed of five individuals

included a family of three second- or third-degree relatives, and that artificially inflated the drift on the Hajji Firuz branch and made detecting a negative statistic $f_3$(Hajji Firuz Neolithic; X, Y), even if present, highly unlikely. Indeed, no $f_3$-statistic turned out to be nominally negative for the groups on the eight-population graph when statistics were calculated according to the original settings (we used settings equivalent to 'useallsnps: NO' and 'inbreed: NO' in classic *ADMIXTOOLS*, 470,389 polymorphic sites were available). Considering this fact, it is not surprising that our automated topology space search is not well constrained. The original study differed from ours since the constraints were introduced manually, but we wanted our topology search to be automatic and to explore a wider range of the parameter space.

In order to provide power to detect negative $f_3$-statistics useful for constraining the model search, we (1) removed two members of the family from the Hajji Firuz Neolithic group, (2) extended the number of individuals and sites available for the Indus Periphery group by generating new shotgun-sequencing data for a previously published library (*Narasimhan et al., 2019*) derived from individual I8726 (see *Supplementary file 2*) and by adding published data for three other individuals from the Indus Valley cline (from Gonur in Turkmenistan and Shahr-i-Sokhta in Iran; *Narasimhan et al., 2019*; *Shinde et al., 2019*), (3) removed from other groups two individuals based on second to third degree relatedness, and (4) removed two individuals from other groups based on evidence of contamination with modern human DNA. All the changes to the dataset are shown in *Supplementary file 3*. In addition to these dataset adjustments, the new algorithm for calculating $f$-statistics makes it possible to compute negative $f_3$-statistics on pseudo-haploid data, but at a cost of removing sites with only one chromosome genotyped in any non-singleton population (see Appendix 1). We eventually detected significantly negative 'admixture' $f_3$-statistics $f_3$(Tepe Hissar Chalcolithic; Ganj Dareh Neolithic, Anatolia Neolithic), $f_3$(Indus Periphery; Ganj Dareh Neolithic, Onge), and other similar statistics for the same target groups. We also observed a nominally negative (*Z*-score = −0.6) statistic $f_3$(Hajji Firuz Neolithic; Ganj Dareh Neolithic, Anatolia Neolithic), which is suggestive but does not by itself prove admixture in the recent history of the Hajji Firuz Neolithic group. For this updated analysis, 249,009 variable sites without missing data at the group level were available for the eight populations.

We repeated topology search with this set of $f$-statistics providing additional constraints, performing 4,000 runs of the *findGraphs* algorithm. The Mota ancient African individual was set as an outgroup and three admixture events were allowed in the eight-population graph. Among 4,000 resulting graphs (one from each *findGraphs* run), 144 were distinct topologically, and the published model was recovered in 13 runs of 4,000 (*Supplementary file 1*). Only four distinct topologies fitting nominally better than the published one were found, and those had LL scores almost identical to that of the published eight-population model (16.97 and 17.66 vs. 17.85). These four alternative models (*Figure 3—source data 5b*) shared all topologically important features of the published model (*Figure 3—source data 5a*). Five other topologies differed in important ways from the published one and emerged as fitting the data worse, but not significantly worse, than the published one (*Figure 3—source data 5c*): two-tailed empirical p-values reported by our bootstrap model comparison method ranged between 0.060 and 0.112. Three of these topologies included a trifurcation of Iranian farmer-related lineages leading to the Indus Periphery, Hajji Firuz Neolithic, and Ganj Dareh Neolithic groups. The other two topologies included Hajji Firuz Neolithic as an unadmixed Anatolian-related lineage. In both cases, the Indus Periphery group was modeled as receiving a gene flow from either the Onge lineage (a proxy for ASI) or a deep Asian lineage.

The finding that the predominant ancestry component of the Indus Periphery group was the most basal branch in the Iranian farmer clade was a prominent claim of the original study *Shinde et al., 2019*; for example, the abstract stated: 'The Iranian-related ancestry in the IVC derives from a lineage leading to early Iranian farmers, herders, and hunter gatherers before their ancestors separated.' Our finding that the Hajji Firuz Neolithic lineage may be as deep within the Iranian clade as the Indus Periphery lineage or may even diverge from the Anatolian branch shows that this statement cannot be confidently made based on AG analysis alone.

However, the findings we have described up to this point do not invalidate the broader conclusion that the AG modeling in Shinde et al. was used to support; namely (using the phrasing from the abstract) that the genetic data 'contradict… the hypothesis that the shared ancestry between early Iranians and South Asians reflects a large-scale spread of western Iranian farmers east.' This

finding if correct is important, since it implies that the Iranian-related ancestry in the IVC (Indus Valley Civilization genetic grouping, which is the same group as IP), split from the Iranian-related ancestry in the first Iranian plateau farmers before the date of the Hajji Firuz farmers, who at ~8000 years ago are among the earliest people living on the Iranian plateau known to have grown West Asian crops. The ancient DNA record combined with radiocarbon dating evidence suggests that beginning around the time of the Hajji Firuz farmers, both West Asian domesticated plants such as wheat and barley, and Anatolian farmer-related admixture, began spreading eastward across the Iranian plateau. If the Iranian-related ancestry in IP was spread eastward into the Indus Valley across the Iranian plateau as part of the same agriculturally associated expansion—perhaps brought by people speaking Indo-European languages as well as introducing West Asian crops—then we would expect to see at least some of the Iranian-related ancestry in IP being a clade with that in Hajji Firuz relative to Ganj Dareh. The fact that we do not find any models compatible with this scenario is thus a potentially important finding. In summary, there are two reasons the genetic analyses we have reported up to this point continue to support the finding that the Iranian-related ancestry in IP is <u>not</u> a clade with the Iranian-related ancestry in Hajji Firuz (and Tepe Hissar) and thus is unlikely to reflect the same eastward movement of agriculturalists. First, in *findGraphs* analysis, all models specifying IP and Tepe Hissar and/or Hajji Firuz as a clade relative to Ganj Dareh were significantly worse-fitting that the published one. Instead, either the Iranian-related ancestry in IP definitively splits off first (the topology from Shinde et al.), or the branching order of the IP, Ganj Dareh, and the Hajji Firuz/Tepe Hissar lineages cannot be determined, or IP, Ganj Dareh, and Tepe Hissar are a clade relative to Hajji Firuz. In all these fitting topologies, the ~10,000-year-old radiocarbon date of the Ganj Dareh individuals sets a lower bound on the split time between IP and Hajji Firuz/Tepe Hissar, which is pre-agriculturalist. This suggests that the Iranian-related ancestry in IP is not due to an eastward agriculturalist expansion.

But in fact, the AG analysis reported above is not an adequate exploration of the problem. Although absolute fits of the best models found are good (WR = 2.5 SE), the parsimony constraint allowing only three admixture events precluded correct modeling of basal Eurasian ancestry shared by all West Asian groups (*Lazaridis et al., 2016*) or of the Indus Periphery group itself, for which a more complex 3-component admixture model was proposed (*Narasimhan et al., 2019*). Concerned that this oversimplification could be causing our search to miss important classes of models, we explored *qpAdm* models for the Indus Periphery group further, following the 'distal' protocol with 'rotating' outgroups outlined by *Narasimhan et al., 2019* and using the dataset and outgroups ('right' populations) from that study. All sites available for analyses were used, following Narasimhan et al. (the 'useallsnps: YES' option). The combined Indus Periphery group we analyzed included seven individuals from Shahr-i-Sokhta and three individuals from Gonur (three individuals were removed from the *Narasimhan et al., 2019* dataset due to potential contamination with modern human DNA and low coverage). We removed one individual from the Ganj Dareh Neolithic group as potentially contaminated, and one second- or third-degree relative was removed from the Anatolia Neolithic group, see the dataset composition in *Supplementary file 4*. We note that no 'distal' *qpAdm* models were tested for the combined Indus Periphery group by *Narasimhan et al., 2019*, and individuals from this group were modeled one by one (Table S82 from *Narasimhan et al., 2019*), which potentially reduced the sensitivity of the method.

A model 'Indus Periphery = Ganj Dareh Neolithic + Onge (ASI)' was strongly rejected for the Indus Periphery group of 10 individuals with a p-value = $2 \times 10^{-15}$, and a model that was shown to be fitting for all Indus Periphery individuals modeled one by one by Narasimhan et al. (Ganj Dareh Neolithic + Onge (ASI) + West Siberian hunter–gatherers (WSHG)) was rejected for the grouped individuals with a p-value = 0.0044. In contrast, a model 'Indus Periphery = Ganj Dareh Neolithic + Onge (ASI) + WSHG + Anatolia Neolithic' was not rejected based on the p > 0.01 threshold used in Narasimhan et al. (the p-value was marginal but passing at 0.03) and produced plausible admixture proportions for all four sources that are confidently above zero: 53.2 ± 5.3%, 28.7 ± 2.1%, 10.5 ± 1.3%, and 7.7 ± 2.9%, respectively (*Supplementary file 5*). The same 'distal' model albeit with Anatolian Neolithic always in higher proportion was found as one of the simplest models (or the only simplest model) fitting the data for many other groups from Iran and Central Asia explored by *Narasimhan et al., 2019*: Aligrama2_IA (13% Anatolia Neolithic), Barikot_H (21%), BMAC (26%), Bustan_BA_o2 (15%), Butkara_H (24%), Saidu_Sharif_H_o (12%), Shahr_I_Sokhta_BA1 (19%), and

SPGT (23%) (*Supplementary file 5* gives a compendium of 'distal' modeling results by Narasimhan et al.). When we modeled Indus Periphery individuals separately, as in *Narasimhan et al., 2019*, the simplest two-component model 'Ganj Dareh Neolithic + Onge (ASI)' was rejected for 5 of 10 individuals (at least ~315,000 sites were genotyped per individual), including the individual I8726 used for the AG analysis in Shinde et al. (*Supplementary file 6*). The model 'Ganj Dareh Neolithic + Onge (ASI)' was not rejected only for individuals with fewer than 141,000 sites genotyped, suggesting that this result is attributed not to population heterogeneity, but to lack of power.

These *qpAdm* results show that the parsimony assumption that was made when constructing the AG analysis in *Shinde et al., 2019* is contradicted by *f*-statistic evidence, and indeed Narasimhan et al. themselves showed this when they presented a distal *qpAdm* model that was more complex (Ganj Dareh Neolithic + Onge (ASI) + WSHG) than the one used for constraining the AG model comparison (Ganj Dareh Neolithic + Onge (ASI)). Another line of evidence used to support the principal historical conclusion by Shinde et al. was a series of $f_4$-statistic cladality tests following correction of allele frequencies using an admixture model 'target group = Iranian farmer + Anatolia Neolithic + Onge (ASI)', with a great majority of tests supporting the deepest position of the Iranian farmer ancestry component in the Indus Periphery group within the Iranian farmer clade (*Shinde et al., 2019*). However, the model used for allele frequency correction (Ganj Dareh Neolithic + Onge (ASI) + Anatolia Neolithic) was simpler than the 4-component model we found and different from the 3-component model for the Indus Periphery group suggested by *Narasimhan et al., 2019*, which is a weakness of that analysis. A valuable direction for future work would be to repeat this analysis with a 4-component allele frequency correction model (Ganj Dareh Neolithic + Onge (ASI) + WSHG + Anatolia Neolithic), although that is beyond the scope of the present study, which simply aims to revisit the reported analyses and test if they fully support their inferences by ruling out alternative explanations.

To explore how the parsimony constraint influences results, we allowed four admixture events in the eight-population graph (*Supplementary file 1*). Among 4,000 resulting graphs (one from each *findGraphs* run), 443 were distinct topologically, and 270 had WRs between 2 and 3 SE, that is, fitted the data well. We explored 35 topologies with LL scores in a narrow range between 9.3 (the best value) and 13.3. In *Figure 3—source data 6*, we show four graphs with four admixture events that model the Indus Periphery group as a mixture of three or four sources, with a significant fraction of its ancestry derived from the Hajji Firuz Neolithic or Tepe Hissar Chalcolithic lineages including both Iranian and Anatolian ancestries. The fits of these models are just slightly different (e.g., LL = 11.7 vs 9.3, both WRs = 2.4 SE) from that of the best-fitting model (*Figure 3—source data 6*), and similar to that of the published graph. Besides these four illustrative graphs, dozens of topologies with very different models for the Indus Periphery group fit the data approximately equally well, suggesting that there is no useful signal in this type of AG analysis when the parsimony constraint is relaxed (this finding is similar to that in our re-analysis of the dog AG in *Bergström et al., 2020*, where relaxation of the parsimony constraint identified equally well-fitting AGs that were very different with regard to their inferences about population history). These results show that at least with regard to the AG analysis, a key historical conclusion of the study (that the predominant genetic component in the Indus Periphery lineage diverged from the Iranian clade prior to the date of the Ganj Dareh Neolithic group at ca. 10 kya and thus prior to the arrival of West Asian crops and Anatolian genetics in Iran) depends on the parsimony assumption, but the preference for three admixture events instead of four is hard to justify based on archaeological or other arguments.

Why did the *Shinde et al., 2019* AG analysis find support for the IP Iranian-related lineage being the first to split, while our *findGraphs* analysis did not? *Shinde et al., 2019* study sought to carry out a systematic exploration of the AG space in the same spirit as *findGraphs*—one of only a few papers in the literature where there has been an attempt to do so—and thus this qualitative difference in findings is notable. We hypothesize that the inconsistency reflects the fact that the deeply diverging WSHG-related ancestry (*Narasimhan et al., 2019*) present in the IVC (Indus Valley Civilization genetic grouping, which is the same group as Indus Periphery) at a level of ca. 10% was not taken into account explicitly neither in the AG analysis nor in the admixture-corrected $f_4$-symmetry tests also reported in *Shinde et al., 2019*. The difference in qualitative conclusions may also reflect the fact that the Shinde et al. study was distinguishing between fitting models relying on a LL difference threshold of 4 units (based on the AIC). As discussed in Appendix 1, AIC is not applicable to AGs

where the number of independent model parameters is topology dependent even if the numbers of groups and admixture events are fixed, and models compared with AIC should have the same number of parameters. Thus, the analysis by Shinde et al. was over-optimistic about being able to reject models that were in fact plausible using its AG fitting setup.

The archaeological and linguistic implications of the Shinde et al. study are important, and there are several avenues available for further attempting to distinguish historical scenarios using *f*-statistics that are outside the scope of a methodological study like this one. Some of our observations that are most challenging for the conclusions of Shinde et al. are those related to the graphs with four admixture events in *Figure 3—source data 6b* that fit the Iranian farmer-related ancestry in the Indus Periphery group as deriving partially from the Hajji Firuz Neolithic or Tepe Hissar Chalcolithic-related lineages. The *qpAdm* method (*Haak et al., 2015*, *Harney et al., 2021*) is able to use information from distal outgroups (such as WSHG) not included in the AG modeling exercise revisited here. Leveraging this information might be able to obtain constraints that would further test the key historical conclusions from Shinde et al. Non-*f*-statistic-based methods could also be informative. Finally, we emphasize that the $f_4$-statistic cladality tests correcting for the Anatolian farmer- and Onge-related admixture in the Indus Periphery grouping do continue to provide support for the historical conclusion of Shinde et al. (these analyses reject models where the Tepe Hissar or Hajji Firuz groups share genetic drift with the Indus Periphery individuals), with the caveat that they do not correct for the WSHG admixture.

## 4.*Librado et al., 2021*

In contrast to the other studies revisited in our work, the AG published by Librado et al., 2021was inferred automatically using *OrientAGraph*. Models with three (Figure 3b in that study) and zero to five (Ext. Data Fig. 5a-d in that study) admixture events were shown. The dataset included 10 populations (nine horse populations and donkey as an outgroup) and was based on 7.4 million polymorphic transversion sites with no missing data at the group level. We observed that some groups used for the *OrientAGraph* and *qpAdm* analyses were very broad geographically and temporally (see Table S1 in the original study), and thus we tested two alternative group compositions: the original one and a streamlined one. In the latter case we included individuals from one archaeological site and one archaeological period per group: the Botai, C-PONT, DOM2, ELEN, and NEO-ANA groups were modified in this way, and the CWC, LP-SFR, Tarpan, and TURG groups were left with the composition used in the original paper (*Supplementary file 7*). In addition, seven individuals with missing data proportion exceeding 80% were removed from the analysis, affecting the donkey outgroup, DOM2, and NEO-ANA groups (*Supplementary file 7*). Since among all possible $f_3$-statistics for the 10 populations three were negative (using all sites available for each population triplet, 'useallsnps: YES'), we applied the upgraded algorithm for calculating *f*-statistics, which removed sites with only one chromosome genotyped in any non-singleton population, resulting in the following site counts for the original and modified population compositions: 11,092 and 1,767,419 sites, respectively. The very low number of sites available in the former case is due to the fact that all individuals are pseudo-haploid, and that two groups (the donkey outgroup and NEO_ANA) are composed of two individuals, a high-coverage one and a low-coverage one. Thus, just sites genotyped in both donkey individuals and in both NEO_ANA individuals were kept. Considering this problem, we focused on the modified group composition only. We tested a range of model complexities (from 3 to 9 gene flows) and performed 1,000 *findGraphs* topology search runs per model complexity class.

Unlike all the other AGs we re-evaluate in this study whose fits to the data were evaluated in the published studies using *qpGraph*, the topologies published in *Librado et al., 2021* (with three to five admixture events) were not evaluated for statistical goodness-of-fit, and in fact fit the *f*-statistic data so poorly that even simple statistics show they cannot be correct (*Figure 3b*, *Figure 3—source data 7a, c, e*, *Supplementary file 1*). In this case, the approach of using *findGraphs* to identify alternative topologies with the same number of admixture events that fit the data better is meaningless, as both the published models and the alternative models do not have enough degrees of freedom to accommodate the complexity present in the real data; all models are guaranteed to be wrong. In particular, we found that WR of the published model with three admixture events is 23.9 SE (*Figure 3—source data 7a*). In this complexity class *findGraphs* found 22 topologically diverse models that fit significantly better than the published one (*Table 1*, *Supplementary file 1*), but nevertheless have extremely poor absolute fits (from 16.2 to 21.3 SE, see a temporally plausible example in *Figure 3—source data 7b*). In the complexity class with four admixture events, no model fitting better than the

published one was found; however, five alternative models fitting not significantly worse than the published one had lower WR (10 or 12 SE vs. 14.1 SE, *Figure 3—source data 7d*). The WR of the published model with five admixture events was 6.9 SE (*Figure 3—source data 7e*); just two models fitting nominally better and 223 models fitting non-significantly worse than the published model and having similar or higher WR were found (*Table 1*, *Supplementary file 1*). These results suggest that while *OrientAGraph* was often (but not always) able to find the same tentative global likelihood optimum as *findGraphs*, neither three nor five admixture events are enough to explain the data since nearly all the groups are probably admixed.

For this reason, we moved to topology searches in more complex model spaces incorporating six to nine admixture events. Temporally plausible models with even a modest fit (WR between 3 and 4 SE) were encountered only among models with eight and nine admixture events (*Figure 3—source data 7j-r*). In the complexity class with eight admixture events, five such temporally plausible fitting models were found, with WRs ranging from 3.4 to 3.9 SE (all these models are shown in *Figure 3—source data 7j-l*). In the complexity class with 9 admixture events, 11 such models were found, with WRs ranging from 3.4 to 4.0 SE (all these models are shown in *Figure 3—source data 7m-r*).

*Librado et al., 2021* discussed the following inferences relying fully or partially on their published AGs reported in that study (*Table 2*): (1) NEO-ANA-related admixture is absent in DOM2; (2) DOM2 and C-PONT are sister groups (they form a clade); (3) there is no gene flow connecting the CWC group and the cluster associated with Yamnaya horses and horses of the later Sintashta culture whose ancestry is maximized in the Western Steppe (DOM2, C-PONT, TURG); (4) there was gene flow from a deep-branching ghost group to NEO-ANA; and (5) Tarpan is a mixture of a CWC-related and a DOM2-related lineage.

The simplest temporally plausible and best-fitting (WR = 3.4 SE) model we found (modified group composition, eight admixture events; see *Figure 3b* and the second model in *Figure 3—source data 7j*) supports inferences 2 and 4, and is incompatible with inferences 1, 3, and 5 (*Table 2*). This newly found model can be interpreted as follows. There is a trifurcation of three deep lineages: a lineage maximized in Western and Central Europe (up to 100% of ancestry in a Late Paleolithic group from France, LP_SFR), a Western-Steppe-specific lineage (up to 55% in TURG), and a Tarpan-specific lineage (22% in Tarpan). Western and Central European horses, represented by LP-SFR, by the majority ancestry in horses found in the Corded Ware culture context (CWC), and by the majority ancestry in wild Neolithic Anatolian horses (NEO_ANA), contributed about half of the ancestry in the Western Steppe groups TURG, C-PONT, and DOM2. The other half of ancestry in the Western Steppe groups is represented by the Western Steppe-specific lineage. That lineage also contributed about 50% of ancestry in wild horses from the Yana Upper Paleolithic site (ELEN), and the other half of ELEN's ancestry is derived from an even deeper lineage. The Botai group is modeled as a mixture of European horses (69%) and Siberian horses (31% ELEN-related ancestry). In contrast to *Librado et al., 2021*, Tarpan is modeled as a mixture of its specific lineage (22%) and a DOM2-related group (78%), and CWC also received ancestry (21%) from a DOM2-related group. All the populations included in the model except for LP_SFR are admixed, and there is evidence of substantial genetic influence from a lineage that was eventually maximized in the Western Steppe (although it did not necessarily originate there) in the ELEN and Botai groups. We consider this model to be plausible from both temporal and geographical perspectives.

We are not arguing here that our eight-admixture-event model represents the true history; in fact, it is highly unlikely to be entirely true, given how large the space of all possible admixture events is and how much admixture evidently occurred relating all these groups (which makes finding the unique truly fitting model extremely unlikely based on *f*-statistic fitting, see the results on simulated data in *Figure 1* and *Appendix 1—figure 2b*). We have also not attempted in any way to replicate the AG exploration procedure performed in the *Librado et al., 2021*. study; the graph fitting procedure was quite different from ours, based on *OrientAGraph* optimization rather than *findGraphs* optimization, and a Block Jackknife procedure with a different genome block size for determining standard errors (4 Mbp in our protocol and ca. 500 kbp in the *Librado et al., 2021*. study). Regardless of how the graph was obtained, it is valuable for providing readers with guidance about which topological features of the graphs are meaningful and stable, and which are less certain, especially—as in the case of the AG presented in the paper—when some features of the presented model do not fit the data by a wide margin, as evident by the WR of 6.9 in the published model for five admixture events.

Our set of 16 temporally plausible and fitting (WR < 4 SE) models with eight or nine admixture events (*Figure 3—source data 7j–r*) is consistent with some features of the published graph being stable: the features (2) that DOM2 and C-PONT are sister groups, and (4) that there was a gene flow from a deep-branching ghost group to NEO-ANA (*Table 2*).

Equally important, however, is our finding that there are plausible models that are inconsistent with other inferences in *Librado et al., 2021*. (*Table 2*). For example, 13 of these 16 models are inconsistent with the suggestion that there was no gene flow connecting the CWC group and the cluster maximized in the Western steppe (DOM2, C-PONT, and TURG) (*Figure 3—source data 7j–r*). In the eight-admixture-event best-fitting plausible model (*Figure 3b* and the second model in *Figure 3—source data 7j*), CWC actually derives appreciable ancestry from the early domestic horse lineage (DOM2) associated with the Sintashta culture to the exclusion of the more distant Yamnaya-associated TURG and C_PONT horses. This scenario presents a parallel to the one observed in humans, with individuals associated with the CWC receiving admixture from Steppe pastoralists albeit in different proportions: ~75% for humans, versus ~20% in horses. These models specifying a substantial Steppe horse contribution to CWC horses would weaken support for the inference in *Librado et al., 2021*. that 'Our results reject the commonly held association between horseback riding and the massive expansion of Yamnaya steppe pastoralists into Europe around 3000 BC.' We are not aware of other lines of evidence in the paper (apart from the fitted AG) that support the claim of no Yamnaya horse impact on CWC horses.

Another example of a feature of the published graph that turned out to be unstable is the model for the Tarpan horse. Only 8 of 16 temporally plausible and fitting models (*Figure 3—source data 7j–r*) support the conclusion by *Librado et al., 2021*. that the Tarpan is a mixture of a DOM2-related and a CWC-related lineage. The other 8 models suggest that Tarpan is a mixture of a deep lineage and a DOM2-related lineage (*Figure 3b* and the second model in *Figure 3—source data 7j*), echoing a hypothesis that Tarpan may be a hybrid with the Przewalski horse lineage not represented in the AG (*Librado et al., 2021*).

Again, we are not arguing here that our fitting alternative model is right—indeed we are nearly certain it is wrong in important aspects—but we are merely pointing out that the complexity of the AG space means that qualitatively quite different conclusions are compatible with the genetic data. Other aspects of the *Librado et al., 2021*. study, most notably the dramatic geographic expansion of the DOM2 modern domestic horse lineage after 4000 years ago in association with the Sintashta culture which is the most extraordinary finding of *Librado et al., 2021*., are in no way challenged by our results.

## 5. *Hajdinjak et al., 2021*

The AG inferred by Hajdinjak et al. was constructed manually on the basis of an SNP set derived from in-solution enrichment of two SNP panels (1240K and a further million of transversion polymorphisms discovered as polymorphic within one or two sub-Saharan African individuals or among archaic humans) and incorporated 11 groups and 8 admixture events (Figure 2d in the original study). The published graph has no clear outgroup since the deepest branch (Denisovan) is admixed. This property of the graph makes automated graph space exploration difficult. We explored two topology classes: (1) 11 groups with 8 admixture events, the original SNP set, Denisovan assigned as an outgroup only at the stage of generating random starting graphs (gene flows to/from the Denisovan branch were allowed at the topology optimization step); and (2) 12 groups with 8 admixture events, chimpanzee added and the original SNP set changed due to the zero missing rate condition, and chimpanzee assigned as an outgroup at both algorithm stages (*Supplementary file 1*). For both graph complexity classes, two topology search settings were tested: (1) either no additional constraints were applied beyond the outgroup constraints described above, or (2) the Vindija Neanderthal and Mbuti were allowed to have no admixture events in their history, and the Denisovan lineage was allowed to have up to one admixture event in its history (these constraints were in line with the model in the original study and with literature on the genetic history of archaic humans, e.g., *Prüfer et al., 2014*). The composition of the groups matched that in the original study, as did the parameter settings for *qpGraph*, with the exception of 'least squares mode', which was used in the original study, but not in our analysis. 'Least squares mode' computes LL scores without taking into account the *f*-statistic covariance matrix, and we confirmed that changing this parameter does not qualitatively change our results. Since no $f_3$-statistics were negative when all sites available for each population triplet were used (the 'useallsnps: YES' option), we did not use the upgraded algorithm for calculating $f_3$-

statistics on pseudo-haploid data. We summarize results across 2,000–4,000 independent runs of the *findGraphs* algorithm (*Supplementary file 1*).

When chimpanzee was not included into the analysis and no topology constraints were applied, nearly all newly found models turned out to be distinct (3,996 of 4,000), nearly all (96.8%) fit nominally better and 15.9% fit significantly better than the published model (*Supplementary file 1*), and absolute fits of 91.3% of novel models are good (WR < 3 SE). Similar results were obtained when the topology search algorithm was constrained: nearly all (89.5%) of 1,999 newly found models fit nominally better and 26% fit significantly better than the published model (*Supplementary file 1*).

When chimpanzee was set as an outgroup and no topology constraints were applied, the picture remained similar. Nearly all newly found models turned out to be distinct (1,996 of 2,000), and a very large fraction of them (56.8%) fit significantly better than the published model (*Supplementary file 1*); 16.4% of novel models demonstrated WR < 3 SE. Similar results were obtained when the topology search algorithm was constrained: most (71.4%) newly found models fit nominally better and 15.7% fit significantly better than the published model (*Table 1*, *Supplementary file 1*, *Figure 2*), which has a poor absolute fit on this set of sites and groups (WR = 4.8 SE, *Figure 3c*, *Figure 3—source data 8*). The statistics described above and the fact that LL scores on all sites lie outside of the LL distribution on resampled datasets (*Figure 2*) suggest that models in this complexity class are overfitted, but the published topology emerged as fitting relatively poorly.

Overfitting arises naturally during manual graph construction as performed in many studies (not only in *Hajdinjak et al., 2021*, but also in, e.g., *Fu et al., 2016*; *Skoglund et al., 2016*; *Yang et al., 2017*; *Posth et al., 2018*; *McColl et al., 2018*; *Moreno-Mayar et al., 2018*; *Tambets et al., 2018*; *van de Loosdrecht et al., 2018*; *Flegontov et al., 2019*; *Sikora et al., 2019*; *Wang et al., 2019*; *Lipson et al., 2020b*; *Shinde et al., 2019*; *Yang et al., 2020*; *Wang et al., 2021*). The graph grew one group at a time, and each newly added group was mapped on to the pre-existing skeleton graph as unadmixed or as a two-way mixture. This imposed constraints on the model-building process. Another constraint imposed was the requirement that all intermediate graphs have good absolute fits (WR below 3 or 4 SE). When the model-building process is constrained in a particular path and fits of all intermediates are required to be good, unnecessary admixture events are often added along the way, and the resulting graph belongs to a complexity class in which models are overfitted and many alternative models fit equally well. There is no single obviously correct order of adding branches to a growing graph. For example, the Kostenki and Sunghir lineages were included into the initial graph (Fig. S6.1 in the original study) as unadmixed lineages, and their admixture status was not revisited at subsequent steps (unlike that of Tianyuan and Ust'-Ishim), except for adding the archaic gene flow common for non-Africans. For that reason, the published graph differs from many alternative better-fitting and temporally plausible graphs where the Kostenki and Sunghir lineages are modeled as more complex mixtures (*Figure 3—source data 8*).

*Hajdinjak et al., 2021*'s published graph had the following notable features that were interpreted by the authors and used to support some conclusions of the study (*Table 2*): (1) there are gene flows from the lineage found in the ~45,000- to 43,000-year-old Bacho Kiro Initial Upper Paleolithic (IUP) individuals to the Ust'-Ishim, Tianyuan, and GoyetQ116-1 lineages; (2) the ~35,000-year-old Bacho Kiro Cave individual BK1653 belonged to a population that was related, but not identical, to that of the GoyetQ116-1 individual; and (3) the Vestonice16 lineage is a mixture of a Sunghir-related and a BK1653-related lineage. To assess if these features are supported by our re-analysis, we focused on our most constrained *findGraphs* run: with chimpanzee set as an outgroup and with the topology constraints applied at the topology search step. We identified 1,421 topologies fitting nominally or significantly better than the published model and satisfying the constraints and moved on to inspect 50 best-fitting topologies for temporal plausibility (all of them fitting significantly better than the published model). All non-African individuals included in the model are Upper Paleolithic and their dates are not drastically different in relative terms: from ca. 45 kya (thousand years before present) for some Bacho Kiro IUP individuals (*Hajdinjak et al., 2021*) to ca. 30 kya for the Vestonice16 individual (*Fu et al., 2016*). Nevertheless, we considered most gene flows from later- to earlier-attested lineages as temporally implausible (for instance, GoyetQ116-1 (~35 kya) → Ust'-Ishim (~44 kya), GoyetQ116-1 (35 kya) → Bacho Kiro IUP (45–43 kya), Kostenki14 (38 kya) → Ust'-Ishim (44 kya), Sunghir III (34.5 kya) → Tianyuan (40 kya), Vestonice16 (30 kya) → Tianyuan (40 kya)) since they imply great antiquity of the later-attested lineages, for example, >40 kya for

the Vestonice16 lineage, and even greater antiquity for the related lineages such as Sunghir III and Kostenki14.

Of the 50 topologies inspected, 32 were considered temporally plausible. Of those topologies, none supported feature 1 of the published AG (there is no replication of the finding of gene flows from the Bacho Kiro IUP lineage specifically to all three of the Ust'-Ishim, Tianyuan, and GoyetQ116-1 lineages). One topology supported features 2 and 3, and partially supported feature 1 (there was Bacho Kiro → GoyetQ116-1 gene flow, but no Bacho Kiro → Tianyuan and Bacho Kiro → Ust'-Ishim gene flows). A total of 17 topologies supported features 2 and 3 but were inconsistent with feature 1; and 14 topologies supported feature 3 only (*Table 2*). Best-fitting representatives of each of these topology classes are shown along with the published model in *Figure 3—source data 8*. Considering topological diversity among models that are temporally plausible, conform to current knowledge about relationships between modern and archaic humans, and fit significantly better than the published model, we conclude that feature 3 is probably robust but other details of the fitted AG in Hajdinjak et al. (Figure 2d of that study)—for example, gene flows to the Ust'-Ishim, Tianyuan and Goyet Q116-1 lineages from sources sharing drift exclusively with the Upper Paleolithic Bacho Kiro lineage—should not be interpreted as providing meaningful inferences about population history of Upper Paleolithic modern humans. For example, the upper right-hand alternative model plotted in *Figure 3—source data 8c* supports features 2 and 3 but includes no gene flows from the Bacho Kiro IUP lineage.

A central finding of Hajdinjak et al. is that the Bacho Kiro IUP group shares more alleles with present-day East Asians than with Upper Paleolithic Holocene Europeans despite coming from Europe. Specifically, the study documents significantly positive statistics of the form $D$(an Asian group, Kostenki14; Bacho Kiro IUP, Mbuti) (Figure 2b and Extended Data Figure 5 in the original study). For example, $D$(Tianyuan, Kostenki14; Bacho Kiro IUP, Mbuti) is significantly positive ($D$ = 0.0032, SE = 0.0010, $Z$ = 3.2) on the dataset used for testing the 12-population graphs (263,698 sites without missing data across all 12 groups). The same statistic is also significantly positive ($D$ = 0.0029, SE = 0.0006, $Z$ = 4.4) when all 1,312,292 non-missing sites in the population quadruplet are analyzed. Hajdinjak et al.'s interpretation of this observation, using the language from the abstract, is that 'there was at least some continuity between the earliest modern humans in Europe [Bacho Kiro IUP] and later people in Eurasia [East Asians]'.

However, a significant $D$-statistic can have multiple explanations. The statistic $f_4$(Tianyuan, Kostenki14; Bacho Kiro IUP, Mbuti) is fitted equally well by the published 12-population AG (Z-score for the difference between the observed and fitted statistics = 0.64) and by, for example, the lower left-hand graph in *Figure 3—source data 8c* (Z-score = 0.94) reproduced in *Figure 3c*. Under the latter model that fits the data significantly better than the published model (p-value = 0.02), the Bacho Kiro IUP and Tianyuan branches are not connected by a gene flow and do not receive gene flows from a third common source, but the common ancestor of Ust'-Ishim and all European Paleolithic lineages receives an 8% gene flow from a divergent modern human lineage splitting deeper than Bacho Kiro IUP and Tianyuan (*Figure 3c*, *Figure 3—source data 8c*). This scenario or some version of it seems archaeologically and geographically plausible and is not disproven by any other line of genetic or non-genetic evidence of which we are aware. It could correspond to a scenario where a primary modern human expansion out of West Asia contributed serially to the major lineages leading to Bacho Kiro, then later East Asians, then Ust'-Ishim, and finally the primary ancestry in later European hunter–gatherers. This has a very different interpretation from the scenario of distinctive shared ancestry between the earliest modern humans in Europe such as Bacho Kiro IUP and later people in East Asia—to the exclusion of later European hunter–gatherers—that is suggested by the Hajdinjak et al. published graph.

We are not claiming that this specific alternative model is correct—indeed, it is almost certainly not the correct one given the topological complexity of the set of all AGs consistent with the data—but the existence of it and many other models that fit the data makes it clear that we do not yet have a unique historical explanation for the excess sharing of alleles that has been documented between some Upper Paleolithic European groups (Bacho Kiro IUP, *Hajdinjak et al., 2021* GoyetQ116-1, *Yang et al., 2017* and *Hajdinjak et al., 2021*) and all East Asians.

## 6. *Lipson et al., 2020b*

The AG in the original study (*Lipson et al., 2020b*) was constructed manually based on an SNP set derived from the 1240K enrichment panel, and the final model was alternatively tested on the combined Human Origins subpanels 4 and 5 (each ascertained on one African individual) or on sites ascertained as polymorphic in archaic humans. The final published model (Extended Data Figure 4 in that study) is very complex (12 groups and 12 admixture events): it exists in a space of ~$10^{44}$ topologies of this complexity. We note that one admixture event was added by *Lipson et al., 2020b* to account for potential modern DNA contamination in ancient Shum Laka individuals, and removing it caused a negligible difference in the fit of the published model (*Supplementary file 1*). Thus, to decrease the complexity of the graph search space, we considered graphs with 12 groups and 11 admixture events. Twenty-two $f_3$-statistics for these 12 groups turned out to be negative (when the 'useallsnps: YES' setting was used), and thus for exploring this graph complexity class we had to remove sites with only one chromosome genotyped in any non-singleton population (*Supplementary file 1*). The following constraints were applied during the topology search: chimpanzee was assigned as an outgroup at both stages of the process (while generating random starting graphs and while searching the topology space); Altai Neanderthal was required to be unadmixed; and non-Africans (French) were required to have at least one admixture event in their history. The composition of the groups we analyzed matched that in the original study. We summarize results across 2,000 independent iterations of the *findGraphs* algorithm.

All newly found models turned out to be distinct (2,000), and 11.9% fit nominally (but not significantly) better than the published model (*Table 1*, *Supplementary file 1*, *Figure 2*). Absolute fits of 36.7% of novel models are good (WR < 3 SE). Fits of the highest-ranking model and the published model are not significantly different according to the bootstrap model comparison method (p-value = 0.176). These metrics, along with the fact that LL scores on all sites lie outside of the LL distribution on resampled datasets (*Figure 2*), suggest that models in this complexity class, including the published model, are overfitted. Of the AGs we re-evaluate in this study, *Lipson et al., 2020b* shares with *Hajdinjak et al., 2021*, *Sikora et al., 2019*, and *Wang et al., 2021* evidence of being overfitted (*Figure 2*).

We also wanted to check if overfitting would be found in the graph complexity classes corresponding to two simpler intermediate graphs from the original study (*Supplementary file 1*): 7 groups and 4 admixture events (Figure S3.24 in that study) and 10 groups and 8 admixture events (Figure S3.25 in that study). The population composition of the dataset we used for this analysis was slightly different from the dataset used by Lipson et al.: the ancient South African hunter–gatherer group was replaced by a related group (present-day Julʼhoan North), and instead of the Shum Laka ancient group, only one high-coverage individual from the same group (I10871) was used. We summarize results across 2,000 or 10,000 independent *findGraphs* runs for each SNP set, for the small and large graphs, respectively. For 7 groups, we found 201 novel topologies fitting better than the published one, and for 10 groups we found nearly 9,000 such topologies (*Supplementary file 1*). In the latter case, 6.8% of newly found topologies fit significantly better than the published topology. For the more complex graph class with 10 groups and 8 admixture events we also found evidence of overfitting: the LL score of the published graph run on the full data is better than almost all the bootstrap replicates on the same data (it falls below the 5th percentile).

Below we discuss selected prominent features of the AG published in the original study (that were interpreted by the authors and used to support some conclusions of the study) and the extent to which these features consistently replicate across the large number of fitting 12-population graphs with 11 admixture events (*Table 2*): (1) A lineage maximized in present-day West African groups (Lemande, Mende, and Yoruba) also contributed some ancestry to the ancient Shum Laka individual and to present-day Biaka and Mbuti; (2) another ancestry component in Shum Laka is a deep-branching lineage maximized in the rainforest hunter–gatherers Biaka and Mbuti; (3) 'super-archaic' ancestry (i.e., diverging at the modern human/Neanderthal split point or deeper) contributed to Biaka, Mbuti, Shum Laka, Lemande, Mende, and Yoruba; and (4) a ghost modern human lineage (or lineages) contributed to Agaw, Mota, Biaka, Mbuti, Shum Laka, Lemande, Mende, and Yoruba. We identified 232 12-population topologies that fit nominally better than the published one, 34 best-fitting topologies (of 232) were manually assessed for temporal plausibility, and we focus on 30 topologies identified as temporally plausible and including a low-level Neanderthal contribution (≤10%) in non-Africans (French). These 30 topologies are shown along with the published model in *Figure 4—source data 1*.

In this set of alternative models, high topological diversity is observed (see an example in *Figure 4a* and further topologies in *Figure 4—source data 1*). We classified the topologies as follows. If an ancestral lineage defined above (for example, a deep-branching lineage maximized in rainforest hunter–gatherers Biaka and Mbuti) exists in the graph, we compared the sets of populations where it is found in the published model and in the model examined. If there was no more than one population where the ancestry is expected according to the published model but not present, or present but not expected, we considered this feature of the published graph supported by the alternative graph. If no ancestral lineage meets the definition above, the feature of the published graph was considered not supported. In all other cases, partial support for the feature was declared (*Figure 4—source data 1*). Considering extreme cases, two alternative graphs completely lacked support for three features of the published graph (*Figure 4a*, *Figure 4—source data 1c*), and one graph supported all four features of the published graph fully (*Figure 4—source data 1q*, the second model). There are some graphs where defining two distinct ancestral lineages maximized in West Africans and in Mbuti and Biaka (features 1 and 2) is essentially impossible since all or nearly all Africans are modeled as a mixture of at least two deep lineages (see the second model in *Figure 4—source data 1d*). In some graphs, there is no single lineage specific to rainforest hunter–gatherers (Biaka, Mbuti, and Shum Laka) since the primary ancestries in these groups form independent deep branches in the African graph (see *Figure 4a* and the second model in *Figure 4—source data 1j*). The ghost modern and super-archaic gene flows to Africans also had no universal support in the set of alternative graphs we examined (see, e.g, *Figure 4a* and *Figure 4—source data 1c*).

Considering the high degree of topological diversity among models that are temporally plausible, conform to known findings about relationships between modern and archaic humans, and fit nominally better than the published model, we conclude that all the four AG features from the original study are not supported by our re-analysis (*Table 2*). As in the case study above, the published manually constructed model is a representative of a large class of models that are equally well fitting to the limits of our resolution. This situation may be attributed to (1) overfitting and/or to (2) the lack of information in the dataset (in the combination of groups and SNP sites) and/or to (3) inherent limitations of *f*-statistics, when distinct topologies predict identical *f*-statistics.

In reconsidering the findings of *Lipson et al., 2020b* it is important to keep in mind that analysis of allele frequency correlation statistics is not the only type of information that can be used to make inferences about population relationships in deep time. Other methodologies have provided important insights into deep African population history, and the model building in *Lipson et al., 2020b* was guided in an informal way by these other lines of evidence. For example, unknown archaic lineages admixing into some African populations were hypothesized through identification of deeply splitting haplotypes that are too long to have been freely mixing with other haplotypes in present-day populations for all of their history (*Hammer et al., 2011*; *Lachance et al., 2012*; *Speidel et al., 2019*). Similarly, analysis of haplotype divergence times of pairs of populations has been used to provide evidence of an early radiation of modern human lineages maximized today in southern African hunter–gatherers, Mbuti rainforest hunter–gatherers, and the great majority of other present-day populations; and a later split of lineages related to East African hunter–gatherers, West African agriculturalists, and non-Africans, which is a feature of the Lipson et al. model (*Campbell and Tishkoff, 2008*; *Mallick et al., 2016*). Notably, some alternative models we found do not contradict the above-mentioned results and are profoundly different from the published model at the same time (see, e.g., *Figure 4a*). These constraints are not enough, however, to provide evidence for all the topological details of the *Lipson et al., 2020b* AG highlighted in this section, or for other features of the *Lipson et al., 2020b* AG that were not invoked in the previous literature and newly proposed in that study, such as the 'ghost modern' lineage splitting around the same time as the lineages leading to southern African hunter–gatherers and central African rainforest hunter–gatherers and mixing in highest proportion to Ethiopian hunter–gatherers and to a lesser proportion to West Africans, and the 'basal West African' lineage that contributes uniquely to Shum Laka. Many of the models that emerged as good fits in our AG-building exercise as the published one did not share some of these features (*Figure 4—source data 1*).

The high diversity of well-fitting AG models that satisfy known constraints relating diverse African populations highlights the need for further research based on multiple lines of genetic analysis (in addition to allele frequency correlation patterns) to obtain further insights into deep African history.

Our results particularly highlight the mystery around the highly distinctive genetic ancestry of the Shum Laka individuals themselves, who represent the newly reported data in the *Lipson et al., 2020b* study and a highly important set of genetic datapoints that was not available prior to the study. The ancestral relationships of these four individuals to both rainforest hunter–gatherers, and to the primary lineage in present-day West Africans, remains an open question, one whose resolution promises meaningful new insights into human population history.

## 7. *Wang et al., 2021*

The AG inferred by *Wang et al., 2021* was constructed manually on the basis of an SNP set derived from the 1240K enrichment panel. We focused our analysis on the final graph (Extended Data Figure 6 in *Wang et al., 2021*, 12 groups and 8 admixture events) and on two simpler intermediates in the model-building process (Figures SI3-9 and SI3-10a in *Wang et al., 2021*). To simplify the latter two models further, we removed a low-level gene flow (1%) from a WHG-related lineage (Loschbour) to the Mongolia Neolithic group, which resulted in negligible LL differences (0.5 and 2.4 log-units, respectively). Thus, using *findGraphs* we explored the following topology classes: 9 groups with 4 admixture events, 10 groups with 5 admixture events, and 12 groups with 8 admixture events (*Supplementary file 1*). The composition of the groups matched that in the original study. We summarized results across 2,000 independent iterations of the *findGraphs* algorithm for each topology class. In the case of the most extensive population set (12 groups), three $f_3$-statistics turned out to be negative (when the 'useallsnps: YES' setting was used), and thus for exploring this graph complexity class we had to remove sites with only one chromosome genotyped in any non-singleton population (*Supplementary file 1*). For this complexity class, we also applied several constraints on the graph space exploration process all of which were shared with the Wang et al. graphs: the Denisovan genome was assigned as an outgroup in the random starting graphs, but not at the topology search stage; up to one admixture event was allowed in the history of the Denisovan group; no admixture events were allowed in the history of Mbuti, Loschbour, and Onge; and the (Denisovan, (Mbuti, (Loschbour, Onge))) branching order was required.

For each topology class we found hundreds to thousands of topologically unique graphs fitting nominally better than the published models (*Table 1*, *Supplementary file 1*). For both simple topology classes, no model fitting significantly better than the published one was found (*Supplementary file 1*). However, the final published model fits the data significantly worse than 12.6% of newly found models of the same complexity (*Table 1*, *Supplementary file 1*). The fact that many topologically diverse models had good absolute fits (65%, 55%, and 15% of distinct newly found graphs with 9, 10, and 12 groups, respectively, had WR < 3 SE) suggests that AG models in these complexity classes are overfitted. Further evidence of overfitting comes from the poor fits of the published model on bootstrap-resampled datasets as compared to their fits on all sites (*Figure 2*).

An important feature of the published graphs in *Wang et al., 2021* that was remarked upon in the study is admixture from a source related to Andamanese hunter–gatherers that is almost universal in East Asians, occurring in the Jomon, Tibetan, Upper Yellow River Late Neolithic, West Liao River Late Neolithic, Taiwan Iron Age, and China Island Early Neolithic (Liangdao) groups (*Table 2*). For example, the abstract states 'Hunter-gatherers from Japan, the Amur River Basin, and people of Neolithic and Iron Age Taiwan and the Tibetan Plateau are linked by a deeply splitting lineage that probably reflects a coastal migration during the Late Pleistocene epoch.' We performed 2,000 *findGraphs* iterations and obtained 1,778 distinct topologies satisfying all the constraints, nearly all of them (1,724) fitting nominally better than the published model, and 12.6% fitting significantly better (*Supplementary file 1*). The models were ranked by LL, and 56 highest-ranking topologies, all of them fitting significantly better than the published one, were assessed for temporal plausibility (models with gene flows from a later group to Tianyuan dated to 40 kya were removed), and 20 topologies were considered temporally plausible (all of them are shown in *Figure 4—source data 2*). According to these topologies, 0–2 East Asian groups had a fraction of their ancestry derived from a source specifically related to Onge, and 19 topologies included gene flows from the European (Loschbour)-related branch to all 8 East Asian groups (*Figure 4—source data 2*). The inferred topological relationships among East Asians are variable in this group of 20 models, and we decided to apply further constraints that guided model ranking and elimination by Wang et al., based on considerations from archaeological evidence, Y chromosome haplogroup divergence patterns, and population split time estimation.

The constraints that are not based on correlation of allele frequencies across populations that Wang et al. applied and that we applied in our re-examination are as follows. First, combined evidence from archaeology, linguistics, and genetics (a closely shared Y chromosome haplogroup) suggests that the present-day Tibetan Plateau population harbors a substantial proportion of ancestry from a large-scale migration from the Neolithic farming groups from the Upper and Middle Yellow River (*Chen et al., 2015*; *Lu et al., 2016*; *Zhang et al., 2019*). These arguments and radiocarbon dates favor the following branching order of predominant ancestry components: (Mongolia East N, (China Upper YR LN, Nepal Chokhopani)). Second, evidence from archaeology, linguistics and genetics suggests that the expansion of Austronesian speakers and the peopling of Taiwan was from southeast coastal China to Taiwan and Southeast Asia, but not from Taiwan to mainland China (*Bellwood, 2011*; *Gray and Jordan, 2000*; *Ko et al., 2014*). These arguments make a China Island EN → Taiwan IA gene flow direction plausible and make the opposite direction of flow less likely. Third, in the original study MSMC cross-coalescence rates were computed for a few pairs of present-day proxies for the ancient groups, and it was argued that they impose constraints on the graph topology. The inferred coalescence date for the Tibetan and Ulchi groups was slightly younger than the Tibetan-Ami and Tibetan-Atayal dates (see Fig. SI3-1 in the original study), suggesting that the Nepal Chokhopani and Mongolia East N group may share ancestral source populations more recently than these two groups and Taiwan IA. We note that it was not clear in the original paper if the difference in coalescence dates is statistically significant, the finding was clearer in MSMC than in MSMC2 analysis, and there was no attempt to calculate expected cross-coalescence profiles using these methods for models incorporating many gene flows. Nevertheless, we applied this constraint as well in an attempt to understand whether, if we used a constraint system similar to that in Wang et al., we would obtain results that agreed with respect to the finding of Onge-related admixture ubiquitous among East Asian groups.

Applying these three additional constraints, we identified two models (among the 56 ones subjected to manual inspection) that satisfied all of them. The highest-ranking of those models is shown in *Figure 4b* and *Figure 4—source data 2c* (the second model), and it includes a 13% (deeply) European-related gene flow to the common ancestor of all East Asians, and gene flows from the Onge-related branch to just two East Asian groups: Nepal Chokhopani and China WLR LN. This model fits the data significantly better than the published model (p-value = 0.028). We do not claim that this is the correct model (indeed we are almost certain that it is not given the high topological diversity of fitting models), but it is not obviously wrong and differs in qualitatively important ways from the published one.

The *Wang et al., 2021* AG provides an illuminating example that helps us to understand the value added by AG construction. The AG construction process in Wang et al. followed a philosophy of not relying entirely on the allele frequency correlation data (not treating the genetic data as independent to explore how much new insight could come from genetic data alone). Instead, the study integrated other lines of genetic evidence as well as linguistic and archaeological insights explicitly into the AG construction process, with the goal of identifying models consistent with multiple lines of evidence. The fact that after this procedure a fitting graph was obtained is not of great interest, as it is essentially always possible to obtain a fit to allele frequency correlation data when enough admixture events are added. The important question is whether any of the emergent features of the graph that were not applied as constraints in the construction process—for example the evidence of ubiquitous Andamanese-related gene flow throughout East Asia suggesting a coastal route expansion that admixed with an interior route expansion proxied by Tianyuan—were stably inferred. Our analysis does not come to this finding consistently among well-fitting and plausible AGs. We conclude that an important feature of the published graph, that is variable levels of Andamanese-related ancestry found in all East Asians except for Siberians (Mongolia Neolithic) and the Upper Paleolithic Tianyuan (Figure 2 in *Wang et al., 2021*), is not supported by *f*-statistic analysis alone (*Table 2*), and indeed we are not aware of a single feature of the *Wang et al., 2021* AG that is stably inferred beyond the constraints applied to build it.

## 8. *Sikora et al., 2019*

Two AGs inferred by *Sikora et al., 2019* were constructed manually based on an SNP set derived from whole-genome shotgun data and incorporated 12 or 13 groups and 10 admixture events (Extended Data Figure 3f in the original study). One graph was focused on West Eurasians, and the

other one on East Eurasians, and both included a Neanderthal, a Denisovan, and an African group (Dinka). Although the chimpanzee outgroup was not included in the original graphs, we added it as it drastically constrains the topology search space. The following additional constraints were applied at the *findGraphs* model optimization stage: the Neanderthal and African groups were unadmixed and the Denisovan group had no more than one admixture event in its history. These three constraints match the features of the published graph. We also repeated topology searches without constraining the admixture status of the Neanderthal, Denisovan, and Dinka. Since no $f_3$-statistics were negative when all sites available for each population triplet were used (the 'useallsnps: YES' option), we did not apply the algorithm that allows unbiased calculation of $f_3$-statistics on pseudo-haploid data at the expense of loss of analyzed SNPs.

In contrast to most other published graphs discussed above, gene flows in the graphs inferred by Sikora et al. do not have equal standing: four low-level gene flows (0–1%) connect the Neanderthal lineage to Upper Paleolithic lineages (Kostenki, Sunghir, Yana, Ust'-Ishim in the "Western" graph and Sunghir, Yana, Mal'ta, Ust'-Ishim in the "Eastern" graph). We repeated each topology search under two alternative settings: either keeping the number of admixture events at 10 to match the published graphs, or at 6 to match simplified versions of the published graphs lacking these low-level Neanderthal gene flows. We performed that modification to simplify the search space and to alleviate the overfitting problem which becomes severe if 10 gene flows across the graph are allowed (*Supplementary file 1*). Here, we compare LL and WR for the original published models and their simplified versions: the "Western" graph including chimpanzee (LL = 65.7, WR = 3.32 SE) vs. its simplified version (LL = 76.5, WR = 3.78 SE) and the "Eastern" graph including chimpanzee (LL = 85.3, WR = 3.11 SE) vs. its simplified version (LL = 102.4, WR = 4.16 SE). In both cases, we found no statistically significant differences in model fits (relying on the bootstrap model comparison method). In summary, topology search was repeated under 8 settings: for the "Western" or "Eastern" graphs, with no constrains on the admixture status or with the constraints specified above, and with 10 or 6 gene flows (*Supplementary file 1*). Below we focus on results for constrained models with 6 admixture events. In contrast, *Figure 2* and *Table 1* show results for constrained "Western" graphs with 10 admixture events.

In the case of the constrained "Western" graphs with 6 admixture events, 1,000 *findGraphs* runs were performed, 894 distinct topologies were found, 4 models fit significantly better, and 151 models fit nominally better than the published one (*Table 1*, *Supplementary file 1*). We inspected those 155 topologies and identified 29 topologies (*Figure 4—source data 3*) that are temporally plausible and include no non-canonical gene flows from archaic groups such as Denisovan or a ghost archaic group to non-Africans. Sikora et al. came to the following striking conclusion relying on the "Western" AG (*Table 2*): the Mal'ta (MA1_ANE) lineage received a gene flow from the Caucasus hunter–gatherer (CaucasusHG_LP or CHG) lineage. However, in our *findGraphs* exploration this direction of gene flow (CHG → Mal'ta) was supported by two of the 29 topologies, and the opposite gene flow direction (from the Mal'ta and East European hunter–gatherer lineages to CHG) was supported by the remaining 27 plausible topologies (*Figure 4—source data 3*). The highest-ranking plausible topology (*Figure 4c*) has a fit that is not significantly different from that of the simplified published model (p-value = 0.392). We note that the gene flow direction contradicting the graph by Sikora et al. was supported by a published *qpAdm* analyses (*Lazaridis et al., 2016*; *Narasimhan et al., 2019*), and *qpAdm* is not affected by the same model degeneracy issues that are the focus of this study. Considering the topological diversity among models that are temporally plausible, conform to robust findings about relationships between modern and archaic humans, and fit nominally better than the published model, we conclude that the direction of the Mal'ta-CHG gene flow cannot be resolved by AG analysis (*Table 2*).

Some important conclusions based on the "Eastern" graph also do not replicate across all plausible AGs (*Table 2*). In the case of the constrained "Eastern" graphs with 6 admixture events, 4,446 topology search iterations were performed, and 2,785 distinct topologies were found. Only 3 topologies fit significantly and 13 nominally better than the published one (p-value for the highest-ranking newly found model vs. the simplified published model = 0.112), and 9.8% of topologies fit not significantly worse than the published one (*Table 1*, *Supplementary file 1*). Of the topologies belonging to these groups, we inspected 116 best-fitting ones and identified 97 topologies that are temporally plausible and include no gene flows from archaic groups such as Denisovan or ghost archaic to non-Africans that are qualitatively different from the gene flows that are currently widely

accepted. The Sikora et al. "Eastern" AG had the following distinctive features that were used to support some conclusions of the study (*Table 2*): (1) the Mal'ta (MA1_ANE) and Yana (Yana_UP) lineages receive a gene flow from a common East Asian-associated source diverging before the ones contributing to the Devil's Cave (DevilsCave_N), Kolyma (Kolyma_M), USR1 (Alaska_LP), and Clovis (Clovis_LP) lineages; (2) European-related ancestry in the Kolyma, USR1, and Clovis lineages is closer to Mal'ta than to Yana; (3) the Devil's Cave lineage received no European-related gene flows, and Kolyma has less European-related ancestry than ancient Americans (USR1 and Clovis). Only feature 2 was universally supported by all the 97 plausible alternative models fitting significantly better, nominally better, or not significantly worse than the simplified published model, while feature 3 was supported by 83 of 97 plausible models, and feature 1 was supported by 28 of 97 plausible models (*Table 2*). We plotted 14 plausible graphs as examples of topologies supporting all three features, two features, or one feature of the published graph (*Figure 4—source data 4*). We note that all the "Eastern" graphs discussed here, both the published and alternative ones, have relatively poor absolute fits with WR above 4 or 5 SE. Increasing the number of gene flows to 10 allowed us to reach much better absolute fits (with WR as low as 2.42 SE), but that resulted in high topological diversity (on a par with some other case studies discussed above). In the case of the constrained "Eastern" graphs with 10 admixture events, 1,000 *findGraphs* runs were performed, and 1000 distinct topologies were found. Of these topologies, 13.2% fit significantly better, 30% nominally better, and 17.6% non-significantly worse than the published model (p-value for the highest-ranking newly found model vs. the published model <0.002) (*Supplementary file 1*).

