## [Editor Report]

This is a rigorous and critical analysis of the performance of a popular suite of methods for inferring population history, accompanied by improvements. Should be of broad interest to anyone interested in human history.

---

## [Decision Letter]

**Decision letter after peer review:**

[Editors’ note: the authors submitted for reconsideration following the decision after peer review. What follows is the decision letter after the first round of review.]

Thank you for submitting the paper "On the limits of fitting complex models of population history to genetic data" for consideration by *eLife*. Your article has been reviewed by 4 peer reviewers, including Magnus Nordborg as the Reviewing Editor and Reviewer #1, and the evaluation has been overseen by a Senior Editor. The following individuals involved in review of your submission have agreed to reveal their identity: Carsten Wiuf (Reviewer #2); David Balding (Reviewer #3).

Comments to the Authors:

We are sorry to say that, after consultation with editors and reviewers, we have decided that this work cannot be published in *eLife* in its current form. We all agree that the work is excellent, and the writing is good in a local sense, but the manuscript is also overly long and its current presentation makes not very accessible even to those who are genuinely interested in this area (see also individual comments below). While the manuscript contains interesting and important points about population genomics inference and human history, these are buried in what is best described as a mixture between a lab blog and a user manual for ADMIXTURETOOLS. It is thus unclear what the target audience is.

If you were willing to substantially streamline the presentation, so that it is more in the form of a conventional (computational) biology paper, we would be willing to reconsider a new submission for *eLife*.

*Reviewer #1 (Recommendations for the authors):*

Research on human population history has long used models of population splitting with subsequent admixture to make sense of polymorphism data (mostly from humans, but also from commensal species). In this paper, Maier et al use introduce algorithmic improvements that allow them to search the space of such "admixture graphs" more exhaustively and use them to re-evaluate a large number of published results in order to investigate whether the corresponding studies really found the "best" models. They find that they generally did not, and conclude that greater caution is in order when interpreting these kinds of data.

The work is encyclopedic (the single-spaced manuscript is over 50 pages - a monograph, really) and well written. The conclusions are well supported and illustrated by many examples.

The main problem I have with the paper is that reading it feels a bit like entering a cladistics meeting where people debate things like the significance of a particular scale on the middle toe of the Lesser Two-headed Tree Wrangler. While the overall point is clear, the details (and there are many) will be obscure unless you are intimately familiar with the specific examples. This means that the majority of the contents of the paper will/must be skipped by those not directly working on human population history. Which is a pity, as the overall conclusions are generally relevant.

Furthermore, just as would have been the case at a cladistics meeting, this paper is narrowly focused on a particular analysis framework. Technical terms like "f-statistic" are thrown around without introduction. In the few remaining corners of population genetics that do not work on human genetics, there are those who do not think in terms of discrete populations, and who hold heretical notions about isolation-by-distance. The question of whether *any* admixture graph is a good model for the data is not asked here - and it arguably should be given the generality of the title. As the paper makes very clear, there are indeed limits to what we can infer about history from genetic data. However, beyond a vague "here be monsters", the paper offers a general audience little guidance about which of the large numbers of claims in illustrious journals are actually likely to be true.

I think this is an excellent and important paper written for a narrow audience. If you want to reach a larger audience, a very different paper is in order.

I also think you do not address the very broad question in the title – this would again be a different paper.

*Reviewer #2 (Recommendations for the authors):*

The authors aim to correct inference problems with existing software by changing and implementing new features of Admixtools. The conclusions are supported by the results. The program will likely be much used.

The authors deserve credit for acknowledging that there might be many admixture graphs fitting a given data set equally well and that this fact has not been acknowledged in any/many previous study(ies), incl the authors own :-

Also they deserve credit for a quite substantial re-analysis of already published data.

The novelty and selling point seem to be a new version of Admixturetools (together with re-analysis). However, several of the "new ideas" implemented are already in Admixturetools 7.0.2, so the novelty is quite limited. While I acknowledge computational speed-ups, clever data manipulations, proper testing and so forth are important and important to publish, it does not seem to have the novelty-level required for *eLife*.

*Reviewer #3 (Recommendations for the authors):*

The authors briefly describe improvements to the popular ADMIXTOOLS software, now released as ADMIXTOOLS2. These include computational speedups that allow computationally-intensive resampling-based methods such as bootstrap to help assess model fit, even for large datasets, as well as a more extensive automated search over model space (exhaustive search is still infeasible, so constraints based on judgments of plausibility are still required).

Using these improved tools, the authors extensively re-analyse 8 published datasets (human, dog and horse), and find many conclusions of the original authors about demographic history to be not well supported. This is because there exist other plausible models, often many of them, that fit the data better, sometimes significantly so, that does not support the original authors' conclusions based on their preferred model. Often there are also even more alternative models that fit the data worse than the published model, but not significantly worse and so they cannot be excluded.

I'm a statistical methods person not closely familiar with any of the studies re-analysed here, so I don't have deep insights about the revised analyses proposed here. However, the authors seem to me to have conducted a careful and well-justified model fitting exercise with important conclusions and implications for future analyses.

The authors conclude with recommendations for future analyses of genetic data to make inferences about population history. They hope to change the field in a similar way to the step change in standards for statistical significance in association studies that occurred around 2008. The guidelines proposed are no doubt open to improvement over time, but I agree that this study is likely to have a big impact on the field of demographic history inference leading to better-justified conclusions from future analyses. The problem addressed is much more complex than genetic association, and conclusions based on limited exploration of a vast model space will always open to further improvement.

It's a very good paper, I don't have much to say at a general level except that it is very long and perhaps trying to do too much. The introduction of ADMIXTOOLS2 and the general discussion about principles of model fitting both seem inadequate. In contrast, the discussion of the re-analyses of datasets is very thorough. Perhaps two papers would have been better?

In a generally well-written paper I found the first sentence of the abstract confusing "… the only information needed to capture the patterns of allele frequency correlation among populations". Why is "capturing.… correlation" your primary focus? I would restate with an objective that is comprehensible to a general reader.

Abstract L17: "Our results suggest that strong claims.…. be made when all well-fitting and temporally plausible models share common topological features" This doesn't follow from the results, you are asserting it as a principle. The "all.… models" is perhaps a strong requirement when there are many of them.

L42 " finding fitting".

L163 " the time required to process.. trivial compared to the time required to compute.…" I found this confusing because its unclear to me what "process" means here.

L291 " Parameters with extremely wide confidence intervals can thus be immediately shown to be poorly determined." Vacuous statement – the paper is already very long! Delete.

L300 define "worst-residual" (maybe refer here to an explanation in methods).

L303 "… methods relying on AIC or BIC.… were over-aggressive". The merits of resampling based versus likelihood based model comparison is an extremely important topic, it's beyond the scope of the present paper to discuss fully here but you should be able to cite some authorities to support this claim.

L342 "to test the null hypothesis that the true difference in log-likelihood.… is zero" The log-likelihood is a function of the data so is not appropriate in a null hypothesis, which should be a statement about models/parameters and not data.

L375 "… without either fixing one of them or forcing the lengths to be evenly distributed." not strictly true and there must be a better way to say what you mean: we can only estimate 1 parameter, not both.

L514 "Lower scores of the fits obtained.… indicate overfitting to the full data set." This is one of the first of many references to overfitting that I found unsatisfactory as I did not notice a discussion of its implications for your analyses. Overfitting is a broad and ubiquitous phenomenon, why is it important here? I feel you should remove these (not needed to justify your main conclusions) or explain better. Are you diagnosing a mechanism that has led other authors astray? Also the specific wording "overfitting to the full data set" seems odd, is there any other kind of overfitting?

L614 "Near East" Near to what? East of what? Please use standard terminology that is meaningful to a general reader, not this outdated colonial-era relic. Similarly for "Middle East" (L687): that term is widely used but not appropriate in a science paper, use "West Asia" or "Eastern Mediterranean" something more specific. Also "Levantine" – is that a standard term in the dog world? It seems not from a quick search. Replace with some more meaningful to a general reader.

L662 "pseudo-diploid" has not been explained. I looked it up and the meaning I found doesn't make sense here.

L1119 "cannot be right" is not an appropriate way to summarise statistical evidence.

L1505 explain(or replace) "high degeneracy".

L1677 "time and population size as the two sources of genetic drift" they are factors affecting rather than sources.

*Reviewer #4: (Recommendations for the authors):*

Overview:

The article is composed of a review of the existing method for the admixture graph estimation and the update of the ADMIXTURETOOLS, and the application of the package to real-life examples. The Results section is hard to follow, and it contains substantial parts which would better fit the Introduction section and the User Manual. Figures are in their draft versions.

Plusses:

The existing ADMIXTURETOOLS method was improved by useful features (confidence intervals and identifiability of admixture graph parameters and the searching the space of all admixture graphs), and heuristics significantly sped it up for f-statistics computations.

Minuses:

L139: what are the philosophical differences between two versions of ADMIXTURETOOLS.

L169: the bias should be demonstrated more clearly.

L175: what is the regression approach to estimate f-statistics.

L186: "pseudo-diploid and pseudo-haploid" – these computational details are not properly explained if they are essential.

L190: "unbiased" – analysis of bias based on the number of samples is required.

L194: "inbreed: YES" option – user manual detail. This detail does not mean much for users who have not worked with ADMIXTURETOOLS.

L201: "incorrect algorithm for calculating" – should be explained more. It would be more important than to mention user manual details.

Figure1 should be rearranged in columns; there are a lot of places for it.

Figure1a: low subfigure x-axis should be reorganized.

Figure1b: if the formula for the F4 is exact, then what is the source of mentioned bias, and where is the heuristics.

Figure1c: what is the difference between all F3/4. Is there any conventional definition, and what is the difference between mentioned definitions? This should be organized as a table, nut a subfigure.

Figure1d: Why are formulas for F3 not presented.

Figure2: Dash dots for the y-axis levels should be presented to compare bars.

Figure3: (a,b) – what does it mean? The figure is in the draft stage.

1. No basic explanation for unprepared readers, what are f2/f3/f4, and what is the practical difference between them.

2. The article contains many references to features from the ADMIXTURETOOLS by names, without explanation. Authors should provide the text to a broad audience of *eLife*, but not for the users of ADMIXTURETOOLS.

Suggestions:

1. I strongly recommend revising the manuscript stricture. For example, the result section is a blend of "literature revie,"&"draft result"&"methods"&"program manual".

2. The text should be significantly reduced, especially when it is hard to read and seems to be in its draft version.

3. Type of the article should be defined more clearer: whether it is a research article? New method? Or a review manuscript of the existing methods.

4. Be more accurate with figures. In the current version, figures are not properly arranged and resemble draft versions.

5. Introduce the abbreviation for "admixture graph" as, for instance, AG.

---

## [Author Response]

[Editors’ note: the authors resubmitted a revised version of the paper for consideration. What follows is the authors’ response to the first round of review.]

Comments to the Authors:We are sorry to say that, after consultation with editors and reviewers, we have decided that this work cannot be published in eLife in its current form. We all agree that the work is excellent, and the writing is good in a local sense, but the manuscript is also overly long and its current presentation makes not very accessible even to those who are genuinely interested in this area (see also individual comments below). While the manuscript contains interesting and important points about population genomics inference and human history, these are buried in what is best described as a mixture between a lab blog and a user manual for ADMIXTURETOOLS. It is thus unclear what the target audience is.

We are glad the reviewers found the work to be of high quality, and we appreciate the critical comments about the presentation; we believe our revision is very much easier to read. The primary audience of our revision is the large community of researchers analyzing genome-wide data including ancient DNA data to learn about population history of diverse species (not just humans, but also other species as exemplified by the dog and horse datasets we analyze). For this community, Admixture Graph fitting has become a tool whose deployment has almost become de rigueur. Our systematic re-examination of 8 published studies highlights the pitfalls that can arise in applying this tool.

If you were willing to substantially streamline the presentation, so that it is more in the form of a conventional (computational) biology paper, we would be willing to reconsider a new submission for eLife.

We streamlined the presentation as follows. We moved all methodological details on *findGraphs* and *ADMIXTOOLS 2* to Appendix 1 (including the original Figures 1 to 3), and shortened the descriptions of case studies substantially, moving a large fraction of that text to Appendix 2. As a result of these changes, the length of the main text (Introduction – Methods, including figure and table legends) was reduced by about half, from 27,452 to 14,410 words.

Reviewer #1 (Recommendations for the authors):Research on human population history has long used models of population splitting with subsequent admixture to make sense of polymorphism data (mostly from humans, but also from commensal species). In this paper, Maier et al use introduce algorithmic improvements that allow them to search the space of such "admixture graphs" more exhaustively and use them to re-evaluate a large number of published results in order to investigate whether the corresponding studies really found the "best" models. They find that they generally did not, and conclude that greater caution is in order when interpreting these kinds of data.The work is encyclopedic (the single-spaced manuscript is over 50 pages - a monograph, really) and well written. The conclusions are well supported and illustrated by many examples.The main problem I have with the paper is that reading it feels a bit like entering a cladistics meeting where people debate things like the significance of a particular scale on the middle toe of the Lesser Two-headed Tree Wrangler. While the overall point is clear, the details (and there are many) will be obscure unless you are intimately familiar with the specific examples. This means that the majority of the contents of the paper will/must be skipped by those not directly working on human population history. Which is a pity, as the overall conclusions are generally relevant.

We modified the main text to remove as many details as possible. All methodological details on the *ADMIXTOOLS 2* algorithms were moved to Appendix 1, which is divided into two parts. Part 1 includes more accessible summaries of methodological problems and the solutions we propose, while part 2 presents further technical details. Appendix 2 presents all the technical details relevant for understanding the case studies from the literature, while brief summaries of the 8 case studies were kept in the main text. This revision cut the overall length by about half and makes the manuscript more accessible.

Furthermore, just as would have been the case at a cladistics meeting, this paper is narrowly focused on a particular analysis framework. Technical terms like "f-statistic" are thrown around without introduction.

We added a brief definition of *f*-statistics in the Introduction (lines 49-59): “AGs are fitted to *f*-statistics (Reich et al. 2009, Patterson et al. 2012, Peter 2016, Soraggi and Wiuf 2019). For convenience, below we use a concise definition of *f*-statistics by Lipson (2020): “The most general definition is that of the *f*_4_-statistic *f*_4_*(A, B; C, D)*, which measures the average correlation in allele frequency differences between (a) populations *A* and *B* and (b) populations *C* and *D* i.e., *(p*_*A*_
*– p*_*B*_*) × (p*_*C*_
*– p*_*D*_*)*, for allele frequencies *p*, typically averaged over many biallelic single-nucleotide polymorphisms. This *f*_4_-statistic is the same as the *D*-statistic up to a normalization factor.” The other *f*-statistics (*f*_2_ and *f*_3_) can be defined as special cases of *f*_4_-statistics: *f*_2_*(A, B) = f*_4_*(A, B; A, B)* and *f*_3_*(A; B, C) = f*_4_*(A, B; A, C)*. *f*_4_-statistics can be written as linear combinations of *f*_3_- or *f*_2_-statistics, and *f*_3_-statistics can be written as linear combinations of *f*_4_- and *f*_2_-statistics. *f*_2_-, *f*_3_-, and *f*_4_-statistics have straightforward interpretations in terms of drift edges along the tree, see Figure 2 in Patterson et al. (2012) and Figure A1b.”

In the few remaining corners of population genetics that do not work on human genetics, there are those who do not think in terms of discrete populations, and who hold heretical notions about isolation-by-distance. The question of whether *any* admixture graph is a good model for the data is not asked here - and it arguably should be given the generality of the title.

We agree that it is important to ask whether any admixture graph with pulse-like gene flows is a good model for real-life data. However, our paper is complex and long enough, and we believe that expanding its scope even further to address this issue is not a good idea. Instead, we focus on a different but equally important problem for the usefulness of these methods, which is that even if gene flows are pulse-like, are fits of admixture graph data to models reliable?

In our revision we demonstrate on simulated data (see the newly added section “Topological diversity of well-fitting models and effects of parsimony constraints on simulated data”), as well as on real data, that the admixture graph fitting approach is deeply problematic even if populations evolve as distinct units, with pulse-like admixture events. We changed the title to the following to be more specific to the type of data we are analyzing: “On the limits of fitting complex models of population history to *f*-statistics”.

As the paper makes very clear, there are indeed limits to what we can infer about history from genetic data. However, beyond a vague "here be monsters", the paper offers a general audience little guidance about which of the large numbers of claims in illustrious journals are actually likely to be true.

With respect, we believe that we are as clear as possible in providing guidance to readers about which claims in previous publications are reliable.

Our paper provides evidence that novel claims about human history based on manually constructed admixture graphs are unreliable when modeling more than six populations and two or three admixture events. In particular, we report a systematic survey of 8 published papers, and in 7 cases find topologically very different admixture graphs that fit the data as well or in some cases even significantly better than published graphs. We wrote the following on lines 277-288 to provide the reader with some guidance about how to think about generalizability of this re-analysis of published findings:

“The studies were selected according to the criterion that an AG model inferred in the study is used as primary evidence for at least one statement about population history in the main text of the study. In other words, the AG method was used in the original studies to support new conclusions about population history, and not simply to show that there is a model that exists that does not contradict results of other genetic analyses, an approach that is a valid use of AGs and has been taken in some studies (e.g., Seguin-Orlando et al. 2014, Narasimhan et al. 2019, Wang et al. 2019). There are many published studies that could have been included in our re-evaluation exercise as they meet our key criterion (e.g., Yang et al. 2017, McColl et al. 2018, Posth et al. 2018, Flegontov et al. 2019, Calhoff et al. 2021, Kutanan et al. 2021, Bergström et al. 2022, Lipson et al. 2022, and Vallini et al. 2022). However, critical re-evaluation of each published graph is an intensive process, and the sample of studies we revisited is diverse enough to identify some general patterns.”

Moreover, the newly added results section devoted to simulated data makes it clear that the key problem that we identified (high topological diversity among well-fitting models) is applicable to virtually any admixture graph fitted to *f*-statistics, whether inferred manually or automatically.

I think this is an excellent and important paper written for a narrow audience. If you want to reach a larger audience, a very different paper is in order.

Following consultation with the editors, we streamlined the presentation as follows. We moved all methodological details on *findGraphs* and *ADMIXTOOLS 2* to Appendix 1 (including Figures 1 to 3), and shortened descriptions of case studies substantially, moving a large fraction of that text to Appendix 2. As a result of these changes, the length of the main text (Introduction – Methods, including figure and table legends) was reduced by about a half, from 27,452 to 14,410 words.

I also think you do not address the very broad question in the title – this would again be a different paper.

As discussed in the section “A Proposed Protocol for Using AG Fitting in Genetic Studies”, in the Conclusions, and elsewhere in the manuscript, we are not attempting to solve all conceptual problems relevant to the approach of using admixture graphs to model population history, although we do discuss a number of these issues. The problems that we do attempt to resolve are listed in Appendix 1, section 1 (see the subsection titles). We changed the title to be more specific to the focus of our paper: “On the limits of fitting complex models of population history to *f*-statistics”.

Reviewer #2 (Recommendations for the authors):The authors deserve credit for acknowledging that there might be many admixture graphs fitting a given data set equally well and that this fact has not been acknowledged in any/many previous study(ies), incl the authors own :-Also they deserve credit for a quite substantial re-analysis of already published data.The novelty and selling point seem to be a new version of Admixturetools (together with re-analysis). However, several of the "new ideas" implemented are already in Admixturetools 7.0.2, so the novelty is quite limited.

We disagree – these methodological improvements are being described for the first time in this paper, and thus the present paper serves as the original reference for them. The improvements to Classic *ADMIXTOOLS* (implemented in Admixtools 7.0.2) were released by Nick Patterson on the github page of Classic *ADMIXTOOLS* and by Robert Maier on the github page of *ADMIXTOOLS 2* in parallel, once it became clear within our laboratory that these were meaningful advances that would improve the performance of both software packages (we are all in the same laboratory).

While I acknowledge computational speed-ups, clever data manipulations, proper testing and so forth are important and important to publish, it does not seem to have the novelty-level required for eLife.

We believe the most novel part of our manuscript lies in the application of the new methodology to revisiting admixture graphs central to several high-profile published studies, and we have refocused our manuscript around this content. Since the approach of fitting admixture graph models of population history is used widely in human and animal genetics for making claims about demography, a cautionary study like ours is in order. Our paper also presents results important for understanding, for example, deep human population history in Africa and East Asia, or population history of dogs and horses. In the revised manuscript, these new findings are highlighted in the greatly shortened Results section, where critical evaluation of each published study occupies up to a page and often substantially less.

Reviewer #3 (Recommendations for the authors):It's a very good paper, I don't have much to say at a general level except that it is very long and perhaps trying to do too much. The introduction of ADMIXTOOLS2 and the general discussion about principles of model fitting both seem inadequate. In contrast, the discussion of the re-analyses of datasets is very thorough. Perhaps two papers would have been better?

Upon consultation with the editors, we streamlined the presentation as follows. We moved all methodological details on *findGraphs* and *ADMIXTOOLS 2* to Appendix 1 (including Figures 1 to 3), and shortened descriptions of case studies substantially, moving a large fraction of that text to Appendix 2. As a result of these changes, the length of the main text (Introduction – Methods, including figure and table legends) was reduced by about a half, from 27,452 to 14,410 words.

In a generally well-written paper I found the first sentence of the abstract confusing "… the only information needed to capture the patterns of allele frequency correlation among populations". Why is "capturing.… correlation" your primary focus? I would restate with an objective that is comprehensible to a general reader.

We unpacked the first sentence as follows: “Our understanding of population history in deep time has been assisted by fitting admixture graphs (“AGs”) to data: models that specify the ordering of population splits and mixtures, which along with the amount of genetic drift on each lineage and the proportions of mixture, is the only information needed to predict the patterns of allele frequency correlation among populations.”

Abstract L17: "Our results suggest that strong claims.…. be made when all well-fitting and temporally plausible models share common topological features" This doesn't follow from the results, you are asserting it as a principle. The "all.… models" is perhaps a strong requirement when there are many of them.

We write on lines 399-405: “The fraction of graphs with scores better than the score of the published graph should not be overinterpreted, as it is influenced by the *findGraphs* algorithm, which does not guarantee ergodic sampling from the space of well-fitting AGs. In particular, it is possible that despite *findGraph*’s strategies for efficiently identifying classes of well-fitting AGs (see Appendix 1, sections 1.B.1 and 2.C), it has a bias toward missing particular classes of graph topologies. However, even one alternative graph which is not significantly worse-fitting than the published graph suggests that we are not able to identify a single best-fitting model.”

Since the sets of graphs found by *findGraphs* are not guaranteed to sample evenly the space of well-fitting models, we believe that even one AG contradicting a statement about history is enough to say that statement lacks support by the AG approach.

L42 " finding fitting".

The sentence (now on line 61) reads as follows: “Previously published methods for finding fitting AGs (mainly *qpGraph* (Patterson et al. 2012) and *TreeMix* (Pickrell and Pritchard 2012, Molloy et al. 2021)) were not well-equipped to handle the large range of equally well-fitting models for three reasons…” We believe there is nothing wrong or unclear in this sentence.

L163 " the time required to process.. trivial compared to the time required to compute.…" I found this confusing because its unclear to me what "process" means here.

We have rephrased this as follows: “For most *f*-statistic-based analyses (for example *qpWave*, *qpAdm*, and *qpGraph*; Figure A1c), the time required to compute *f*_3_- and *f*_4_-statistics algebraically from *f*_2_-statistics is trivial compared to the time required to load genotype data and compute *f*_2_-statistics. These *f*_2_-statistics can be stored and re-used to compute *f*_3_- and *f*_4_-statistics, thus reducing the size of the input data, runtime, and memory requirements by orders of magnitude (Figure A1a, 1d).”

L291 " Parameters with extremely wide confidence intervals can thus be immediately shown to be poorly determined." Vacuous statement – the paper is already very long! Delete.

We have removed this.

L300 define "worst-residual" (maybe refer here to an explanation in methods)

We mean “worst *f*_4_-statistic residuals (WR)”, as mentioned on line 213 and elsewhere. We have tried to clarify this in our revision.

L303 "… methods relying on AIC or BIC.… were over-aggressive". The merits of resampling based versus likelihood based model comparison is an extremely important topic, it's beyond the scope of the present paper to discuss fully here but you should be able to cite some authorities to support this claim.

We decided to avoid a lengthy discussion about the ability of model likelihood to take the variability across SNPs into account. As mentioned at several points in the main text and Appendices, the bootstrapping approach to model comparison is superior to AIC and BIC since these methods have another problem: it is often not clear what the effective number of degrees of freedom is in the two models being compared since in the case of AGs it depends not only on the number of graph edges, but also on graph topology.

L342 "to test the null hypothesis that the true difference in log-likelihood.… is zero" The log-likelihood is a function of the data so is not appropriate in a null hypothesis, which should be a statement about models/parameters and not data.

We have rephrased the sentence as: “These new incorrect models are symmetrically related to the first graph and can be used to test whether the true difference in LL scores of these two graphs is zero.”

L375 "… without either fixing one of them or forcing the lengths to be evenly distributed." not strictly true and there must be a better way to say what you mean: we can only estimate 1 parameter, not both.

The statement above was removed to avoid confusion.

L514 "Lower scores of the fits obtained.… indicate overfitting to the full data set." This is one of the first of many references to overfitting that I found unsatisfactory as I did not notice a discussion of its implications for your analyses. Overfitting is a broad and ubiquitous phenomenon, why is it important here? I feel you should remove these (not needed to justify your main conclusions) or explain better. Are you diagnosing a mechanism that has led other authors astray? Also the specific wording "overfitting to the full data set" seems odd, is there any other kind of overfitting?

We mention evidence of overfitting (log-likelihood (LL) on all sites outside of the LL confidence interval on bootstrap replicates) to highlight that some published models are too-good fits to the data to be statistically believable, due to fitting more parameters to the data that the data are able to support. We also discuss a mechanism that can lead to construction of such overly complex models: building graphs by adding one group at a time AND the requirement that model WR should be below 3 SE or another threshold AT ALL STEPS of this process. This is commented on in the paper on lines 731-740:

“Overfitting arises naturally during manual graph construction as performed in many studies (not only in Hajdinjak et al. (2021), but also in Fu et al. 2016, Skoglund et al. 2016, Yang et al. 2017, Posth et al. 2018, McColl et al. 2018, Moreno-Mayar et al. 2018, Tambets et al. 2018, van de Loosdrecht et al. 2018, Flegontov et al. 2019, Sikora et al. 2019, Wang et al. 2019, Lipson et al. 2020, Shinde et al. 2019, Yang et al. 2020, and Wang et al. 2021). The graph grew one group at a time, and each newly added group was mapped on to the pre-existing skeleton graph as unadmixed or as a 2-way mixture. Another requirement was that all intermediate graphs have good absolute fits (WR below 3 or 4 SE). When the model-building process is constrained in a particular path and fits of all intermediates are required to be good, unnecessary admixture events are often added along the way, and the resulting graph belongs to a complexity class in which models are overfitted.”

L614 "Near East" Near to what? East of what? Please use standard terminology that is meaningful to a general reader, not this outdated colonial-era relic. Similarly for "Middle East" (L687): that term is widely used but not appropriate in a science paper, use "West Asia" or "Eastern Mediterranean" something more specific. Also "Levantine" – is that a standard term in the dog world? It seems not from a quick search. Replace with some more meaningful to a general reader.

We removed the words “to the full data set”, as suggested.

L662 "pseudo-diploid" has not been explained. I looked it up and the meaning I found doesn't make sense here.

We replaced “Near East” and “Middle East” by “West Asia”; and we replaced “Levantine” by “East Mediterranean”, as suggested.

L1119 "cannot be right" is not an appropriate way to summarise statistical evidence.

We have updated the definition of “pseudo-haploid” data and removed the term “pseudo-diploid” to avoid confusion. See lines 44-45, Appendix 1: ‘genotypes derived by randomly selecting one sequencing read at each variable position (“pseudo-haploid” data)’.

L1505 explain (or replace) "high degeneracy".

The sentence was rephrased.

L1677 "time and population size as the two sources of genetic drift" they are factors affecting rather than sources.

“High degeneracy” was replaced by “high topological diversity”.

Reviewer #4: (Recommendations for the authors):Overview:The article is composed of a review of the existing method for the admixture graph estimation and the update of the ADMIXTURETOOLS, and the application of the package to real-life examples. The Results section is hard to follow, and it contains substantial parts which would better fit the Introduction section and the User Manual. Figures are in their draft versions.

Upon a consultation with the editors, we streamlined the presentation as follows: we moved all methodological details on *findGraphs* and *ADMIXTOOLS 2* to Appendix 1 (including Figures 1 to 3), and shortened descriptions of case studies substantially, moving a large fraction of that text to Appendix 2. As a result of these changes, the length of the main text (Introduction – Methods, including figure and table legends) was reduced by about a half, from 27,452 to 14,410 words.

Plusses:The existing ADMIXTURETOOLS method was improved by useful features (confidence intervals and identifiability of admixture graph parameters and the searching the space of all admixture graphs), and heuristics significantly sped it up for f-statistics computations.Minuses:L139: what are the philosophical differences between two versions of ADMIXTURETOOLS.

This statement was removed for clarity. A conceptual difference between the two packages is the possibility of calculating various statistics (such as admixture graph log-likelihoods (LLs) or *qpAdm* p-values) across bootstrap-resampled replicates of the dataset (implemented in *ADMIXTOOLS 2*), and the introduction of *findGraphs*.

L169: the bias should be demonstrated more clearly.

To be consistent with methodology in the original publications and to avoid biases associated with missing data, in this study we removed missing data prior to calculation of *f*-statistics and fitting of admixture graphs. We believe that the discussion of biased *f*-statistics associated with missing data is important but would add additional detail in what is already a long paper, which is not necessary for our exposition. A discussion of best practices for calculating *f*-statistics in the presence of missing data is presented on the *ADMIXTOOLS 2* github page: https://uqrmaie1.github.io/admixtools/articles/fstats.html#bias-due-to-missing-data

L175: what is the regression approach to estimate f-statistics.

We have rephrased the sentences for clarity: “The program *qpfstats* in Classic *ADMIXTOOLS* implements an idea which strikes a balance between these two extremes. It increases the accuracy of estimation of *f*-statistics by using a regression approach to jointly estimate the values of all *f*_2_-, *f*_3_- and *f*_4_-statistics relating a set of populations. Specifically, qpfstats searches for values of these statistics that are not only consistent with information from the SNPs that have data in the groups used to compute each particular *f*-statistic, but also satisfy the algebraic relationships expected with other *f*-statistics (thus incorporating information from data at many additional SNPs). See further details on this algorithm at https://github.com/DReichLab/AdmixTools/blob/master/qpfs.pdf.”

L186: "pseudo-diploid and pseudo-haploid" – these computational details are not properly explained if they are essential.

We have updated the definition of “pseudo-haploid” data and removed the term “pseudo-diploid” to avoid confusion. See lines 44-45, Appendix 1: ‘genotypes derived by randomly selecting one sequencing read at each variable position (“pseudo-haploid” data)’.

L190: "unbiased" – analysis of bias based on the number of samples is required.

The corrected algorithm for calculating negative *f*_3_-statistics on non-singleton pseudo-haploid populations is not new (it is triggered by the “inbreed: YES” setting in Classic *ADMIXTOOLS*) and rather trivial, and for this reason we believe that an analysis of its performance is beyond the scope of our already-long manuscript. However, this problem was overlooked in Classic *ADMIXTOOLS qpGraph*, where the basic algorithm (that is OK for diploid, but not for pseudo-haploid data) is applied if at least one singleton pseudo-haploid group is present in the dataset. The corrected algorithm for calculating *f*_3_-statistics is discussed in our manuscript extensively on lines 42-78 and 328-354 in Appendix 1 and on lines 809-850 in Appendix 2.

L194: "inbreed: YES" option – user manual detail. This detail does not mean much for users who have not worked with ADMIXTURETOOLS.

In the updated manuscript, *ADMIXTOOLS* option names and similar details are mentioned in Appendices only, not in the main text. We believe that these names of settings in Classic *ADMIXTOOLS* or *ADMIXTOOLS 2* are convenient pointers referring to algorithms that would otherwise be referred to by long names, and therefore we keep these pointers in the Appendices. We also believe that providing some “user manual details” is important for helping readers to understand which settings are equivalent in Classic *ADMIXTOOLS* and *ADMIXTOOLS 2*. Options bear different names in these programs, and the manual for Classic *ADMIXTOOLS* is very short.

L201: "incorrect algorithm for calculating" – should be explained more. It would be more important than to mention user manual details.

The problem we discovered in calculation of *f*_3_-statistics of the form *f*_3_(pseudo-haploid group; *A, B*) by the *qpGraph* software and the new algorithm for calculation of *f*_3_-statistics on pseudo-haploid data is described in detail in Appendix 1 on lines 42-78 and 328-354, in Appendix 2 on lines 809-850, and is referred to in the main text on lines 595-599. We believe this issue is discussed sufficiently in the revised paper.

Figure1 should be rearranged in columns; there are a lot of places for it.

We moved the former Figure 1 to the Appendix and feel that it is a good visual presentation.

Figure1a: low subfigure x-axis should be reorganized.

We do not understand how the x-axis in this figure panel should be reorganized; the figure seems clear to us.

Figure1b: if the formula for the F4 is exact, then what is the source of mentioned bias, and where is the heuristics.

The formula expressing an *f*_4_-statistic as a sum of *f*_2_-statistics is only exact when using the same SNPs for all population pairs. In the presence of missing data, the formula no longer holds, and this can be a source of bias. This issue is well-known in and is described extensively on the *ADMIXTOOLS 2* manual pages:

https://uqrmaie1.github.io/admixtools/articles/fstats.html#bias-due-to-missing-data

Figure1c: what is the difference between all F3/4. Is there any conventional definition, and what is the difference between mentioned definitions?

We added a definition of *f*-statistics in the Introduction (lines 49-59): “AGs are fitted to *f*-statistics (Reich et al. 2009, Patterson et al. 2012, Peter 2016, Soraggi and Wiuf 2019). For convenience, below we use a concise definition of *f*-statistics by Lipson (2020): “The most general definition is that of the *f*_4_-statistic *f*_4_*(A, B; C, D)*, which measures the average correlation in allele frequency differences between (a) populations *A* and *B* and (b) populations *C* and *D* i.e., *(p*_*A*_
*– p*_*B*_*) × (p*_*C*_
*– p*_*D*_*)*, for allele frequencies *p*, typically averaged over many biallelic single-nucleotide polymorphisms. This *f*_4_-statistic is the same as the *D*-statistic up to a normalization factor.” The other *f*-statistics (*f*_*2*_ and *f*_3_) can be defined as special cases of *f*_4_-statistics: *f*_2_*(A, B) = f*_4_*(A, B; A, B)* and *f*_3_*(A; B, C) = f*_4_*(A, B; A, C)*. *f*_4_-statistics can be written as linear combinations of *f*_3_- or *f*_2_-statistics, and *f*_3_-statistics can be written as linear combinations of *f*_4_- and *f*_2_-statistics. *f*_2_-, *f*_3_-, and *f*_4_-statistics have straightforward interpretations in terms of drift edges along the tree, see Figure 2 in Patterson et al. (2012) and Figure A1b.”

This should be organized as a table, nut a subfigure.

If the editors allow, we would prefer to keep panel **c** as a sub-figure, since we would like to avoid another display item which would be referred to only once.

Figure1d: Why are formulas for F3 not presented.

The formulas shown at the bottom of panel d stand for the number of data points that *ADMIXTOOLS* and *ADMIXTOOLS 2* keep in memory, see the legend:

“Shown below are the number of data points for *N* individuals, *M* SNPs, and *k* populations.”

See also below in the legend:

“The exact number of all possible non-redundant *f*_2_, *f*_3_, and *f*_4_-statistics for *k* populations are (k2), 12(k3), and  13(k4).”

Figure2: Dash dots for the y-axis levels should be presented to compare bars.

We moved the former Figure 2 to the Appendix and feel that it is a good visual presentation.

Figure3: (a,b) – what does it mean? The figure is in the draft stage.

Please see the legend for this figure (now Figure A3):

“**a** Bootstrap sampling distributions of the log-likelihood scores for two AGs (shown in Figure S1) of the same populations fitted using real data. Vertical lines show the log-likelihood scores computed on all SNP blocks.

**b** Distribution of differences of the bootstrap log-likelihood scores for both graphs (same data as in a). The purple area shows the proportion of resamplings in which the first graph has a higher score than the second graph. The two-sided p-value for the hypothesis of no difference is equivalent to twice that area (or one over the number of bootstrap iterations if all values fall on one side of zero). In this case it is 0.078.”

1. No basic explanation for unprepared readers, what are f2/f3/f4, and what is the practical difference between them.

We added a definition of f-statistics in the Introduction (lines 49-57):

“AGs are fitted to f-statistics (Reich et al. 2009, Patterson et al. 2012, Peter 2016, Soraggi and Wiuf 2019). For convenience, below we use a concise definition of f-statistics by Lipson (2020): “The most general definition is that of the *f*_4_-statistic *f*_4_*(A, B; C, D)*, which measures the average correlation in allele frequency differences between (a) populations *A* and *B* and (b) populations *C* and *D* i.e., *(p*_*A*_
*– p*_*B*_*) × (p*_*C*_
*– p*_*D*_*)*, for allele frequencies *p*, typically averaged over many biallelic single-nucleotide polymorphisms. This *f*_4_-statistic is the same as the *D*-statistic up to a normalization factor.” The other *f*-statistics (*f*_2_ and *f*_3_) can be defined as special cases of *f*_4_-statistics: *f*_2_*(A, B) = f*_4_*(A, B; A, B)* and *f*_3_*(A; B, C)* = f_4_*(A, B; A, C)*. *f*_4_-statistics can be written as linear combinations of *f*_3_- or *f*_*2*_-statistics, and *f*_3_-statistics can be written as linear combinations of *f*_4_- and *f*_2_-statistics.”

We also refer to the easily understandable topological interpretation of these statistics in terms of drift edges as featured in Patterson et al. 2012 and in our paper in Figure A1b:

“*f*_2_*-*, *f*_3_*-*, and *f*_4_-statistics have straightforward interpretations in terms of drift edges along the tree, see Figure 2 in Patterson et al. (2012) and Figure A1b”.

Practical usage of various *f*-statistics is summarized in Figure A1c.

2. The article contains many references to features from the ADMIXTURETOOLS by names, without explanation. Authors should provide the text to a broad audience of eLife, but not for the users of ADMIXTURETOOLS.

In the updated manuscript, *ADMIXTOOLS* option names and similar details are no longer mentioned in the main text, only in Appendices. We believe that these names of settings in Classic *ADMIXTOOLS* or *ADMIXTOOLS 2* are convenient pointers referring to algorithms that would otherwise be referred to by long names, and therefore we chose to keep these them in the Appendices.

Suggestions:1. I strongly recommend revising the manuscript stricture. For example, the result section is a blend of "literature revie,"&"draft result"&"methods"&"program manual".2. The text should be significantly reduced, especially when it is hard to read and seems to be in its draft version.

Upon a consultation with the editors, we streamlined the presentation as follows. We moved all methodological details on *findGraphs* and *ADMIXTOOLS 2* to Appendix 1 (including Figures 1 to 3), and shortened descriptions of case studies substantially, moving a large fraction of that text to Appendix 2. As a result of these changes, the length of the main text (Introduction – Methods, including figure and table legends) was reduced by about a half, from 27,452 to 14,410 words.

3. Type of the article should be defined more clearer: whether it is a research article? New method? Or a review manuscript of the existing methods.

We hope that the new structure of the manuscript makes it more transparent that we used our newly developed methodology (now presented in Appendix 1, with the support of the editors) to present a critical re-assessment of published findings.

4. Be more accurate with figures. In the current version, figures are not properly arranged and resemble draft versions.

To address this concern, we have moved former Figures 1 to 3 to Appendix 1. The remaining main-text figures 1 and 2 are effectively presented, in our view. Presenting graphs in Figure 3 in the form of small panels would compromise their readability greatly, and we kept them as they are. Following reviewer feedback, we replotted the graphs in Figure 3 in the “classic” Graphviz style to make them more readable, and all graphs in the supplementary figures were replotted in the same style.

5. Introduce the abbreviation for "admixture graph" as, for instance, AG.

We introduced the abbreviation “AG”, as requested.